



# On the understanding of tropospheric fast photochemistry: airborne observations of peroxy radicals during the EMeRGe-Europe campaign

Midhun George[1], Maria Dolores Andrés Hernández[1], Vladyslav Nenakhov[1*], Yangzhuoran Liu[1], John Philip Burrows[1]; Birger Bohn[2]; Eric Förster[3], Florian Obersteiner[3], Andreas Zahn[3]; Theresa Harlaß[4], Helmut Ziereis[4], Hans Schlager[4]; Benjamin Schreiner[5], Flora Kluge[5], Katja Bigge[5], and Klaus Pfeilsticker[5]

[1] Institute of Environmental Physics, University of Bremen, Germany
[2] Institute of Energy and Climate Research, IEK-8: Troposphere, Forschungszentrum Jülich GmbH, Jülich, Germany
[3] Institute of Meteorology and Climate Research, Karlsruhe Institute of Technology, Karlsruhe, Germany
[4] Deutsches Zentrum für Luft- und Raumfahrt (DLR), Institut für Physik der Atmosphäre, Oberpfaffenhofen, Germany
[5] Institute of Environmental Physics, University of Heidelberg, Heidelberg, Germany
[*] now at Flight Experiments, DLR Oberpfaffenhoffen, Germany

Correspondence to M. George (midhun@iup.physik.uni-bremen.de) and M. D. Andrés Hernández (lola@iup.physik.uni-bremen.de).

**Abstract.** In this study, airborne measurements of the sum of hydroperoxyl ($HO_2$) and organic peroxy ($RO_2$) radicals that react with NO to produce $NO_2$, i.e. $RO_2^*$, coupled with actinometry and other key trace gases measurements, have been used to test the current understanding of the fast photochemistry in the outflow of major population centres (MPCs). All measurements were made during the airborne campaign of the EMeRGe (**E**ffect of **M**egacities on the transport and transformation of pollutants on the **R**egional to **G**lobal scal**e**s) project in Europe on-board the **H**igh **A**ltitude **Lo**ng range research aircraft (HALO). The on-board measurements of $RO_2^*$ were made using the in-situ instrument **Pe**roxy **R**adical **C**hemical **E**nhancement and **A**bsorption **S**pectrometer (PeRCEAS). $RO_2^*$ mixing ratios up to 120 pptv were observed in air masses of different origins and composition under different local actinometrical conditions during seven HALO research flights in July 2017 over Europe.

The range and variability of the $RO_2^*$ measurements agree reasonably well with radical production rates estimated using photolysis frequencies and $RO_2^*$ precursor concentrations measured on-board. $RO_2^*$ is primarily produced following the photolysis of ozone ($O_3$), formaldehyde (HCHO), glyoxal (CHOCHO), and nitrous acid (HONO) in the airmasses investigated. The suitability of **p**hotostationary **s**teady-**s**tate (PSS) assumptions to estimate the mixing ratios and the variability of $RO_2^*$ during the airborne observations is investigated. The PSS assumption is robust enough to calculate $RO_2^*$ mixing rations for most conditions encountered in air masses measured. The similarities and discrepancies between measured and calculated $RO_2^*$ mixing ratios are analysed stepwise. The parameters, which predominantly control the $RO_2^*$ mixing ratios under different chemical and physical regimes, are identified during the analysis. The dominant removal processes of $RO_2^*$ in the airmasses measured up to 2000 m are the loss of OH and RO through the reaction with $NO_x$ during the radical interconversion. Above 2000 m, $HO_2 - HO_2$ and $HO_2 - RO_2$ reactions dominate $RO_2^*$ loss reactions. $RO_2^*$ calculations underestimated (< 20 %) the measurements by the analytical expression inside the pollution plumes probed. The underestimation is attributed to the limitations of the PSS analysis to take into account the production of $RO_2^*$ through oxidation and photolysis of the OVOCs not measured during EMeRGe.

## 1. Introduction

Hydroperoxyl ($HO_2$) and organic peroxy ($RO_2$, where R stands for any organic group) radicals are reactive species that play a key role in the chemistry of the troposphere. In combination with the hydroxyl (OH) radical, $HO_2$ and $RO_2$ take part in rapid





chemical processes that control the lifetime of many key trace constituents in the troposphere. Examples of key tropospheric
processes involving $HO_2$ and $RO_2$ are as follows:
• the catalytic cycles, which produce and destroy ozone ($O_3$)
• the generation of key inorganic acids, which are precursors of aerosol (e.g. sulphuric acid, $H_2SO_4$) and important
chemical constituents (e.g. nitric acid, $HNO_3$) in both summer and winter smog
• the generation of organic acids; the production of hygroscopic hydrogen peroxide ($H_2O_2$) and organic peroxides
(ROOH), which enter aerosol and cloud droplets
• the generation of organic peroxy nitrates ($RO_2NO_2$), peroxyacetyl nitrate ($CH_3COO_2NO_2$, PAN) and other summer
smog constituents.
The abundance of $HO_2$ and $RO_2$ in the free troposphere has a non-linear and complex dependency on photochemistry, initiated
by solar actinic radiation, and on the concentration of the precursors, such as carbon monoxide (CO), **v**olatile **o**rganic **c**ompounds
(VOCs), and peroxides. It also strongly depends on the amounts of nitrogen monoxide (NO) and nitrogen dioxide ($NO_2$) due to
the gas-phase reactions of NO and $NO_2$ with the OH and organic oxy (RO) radicals formed during the radical interconversion.
The main production and loss processes of $HO_2$ and $RO_2$ in the troposphere are summarised as follows:
a) Production processes of $HO_2$ and $RO_2$ through photolysis and oxidation by OH formed through photolysis
$O_3 + h\nu \ (\lambda < 320 \ nm) \rightarrow O(^1D) + O_2$ (R1)
$O(^1D) + H_2O \rightarrow 2OH$ (R2a)
$O(^1D) + N_2 \rightarrow O(^3P) + N_2$ (R2b)
$O(^1D) + O_2 \rightarrow O(^3P) + O_2$ (R2c)
$HONO + h\nu \ (\lambda \leq 400 \ nm) \rightarrow OH + NO$ (R3)
$H_2O_2 + h\nu \rightarrow 2OH$ (R4)
$OH + O_3 \rightarrow HO_2 + O_2$ (R5)
$OH + CO + O_2 \rightarrow HO_2 + CO_2$ (R6)
$(*)OH + CH_4 + O_2 \rightarrow CH_3O_2 + H_2O$ (R7)
$(**)HCHO + h\nu \ (\lambda < 340 \ nm) + 2O_2 \rightarrow 2HO_2 + CO$ (R8)
$(*)(**)CH_3CHO + h\nu \ (\lambda < 340 \ nm) + 2O_2 \rightarrow CH_3O_2 + HO_2 + CO$ (R9)
$(**) \ CH_3C(O)CH_3 + h\nu \ (\lambda < 340 \ nm) + 2O_2 \rightarrow 2 \ CH_3O_2 + CO$ (R10)
$(**)CHOCHO + h\nu + 2O_2 \xrightarrow{M} 2HO_2 + 2CO$ (R11)
(*) The $CH_3$ produced from the oxidation of $CH_4$ or the photolysis of VOCs further reacts with $O_2$ to form $CH_3O_2$. The net reaction is written since the formation
of $CH_3O_2$ is much faster than the $CH_3$ formation due to the high amount of $O_2$ present in the atmosphere.
(**) H and CHO formed through the VOC photolysis further react with $O_2$ to form $HO_2$. The net reaction is written since the formation of $HO_2$ is much faster
than the H and CHO formation due to the high amount of $O_2$ present in the atmosphere.




VOCs + OH → OH + $HO_2$ + $RO_2$ and other oxidation products      (R12)
alkenes + $O_3$ → OH + $RO_2$ + other oxidation products      (R13)

b)   Loss processes of $HO_2$ and $RO_2$
$HO_2$ + $HO_2$ → $H_2O_2$ + $O_2$      (R14)
$HO_2$ + $RO_2$ → ROOH + $O_2$      (R15)
$RO_2$ + $RO_2$ → ROOH + $R_{C-1}CHO$ + $O_2$      (R16a)
OH + $HO_2$ → $H_2O$ + $O_2$      (R17)
OH + OH $\xrightarrow{M}$ $H_2O_2$      (R18)
OH + NO $\xrightarrow{M}$ HONO      (R19)
OH + $NO_2$ $\xrightarrow{M}$ $HNO_3$      (R20)
OH + HONO → $H_2O$ + $NO_2$      (R21)
RO + NO $\xrightarrow{M}$ RONO      (R22)
In addition, $HO_2$ and $RO_2$ undergo radical interconversion processes through the following reactions:
$RO_2$ + $RO_2$ → RO + RO + $O_2$      (R16b)
$HO_2$ + NO → OH + $NO_2$      (R23)
$HO_2$ + $O_3$ → OH + $2O_2$      (R24)
$RO_2$ + NO → RO + $NO_2$      (R25)
RO + $O_2$ → $R_{H-1}O$ + $HO_2$      (R26)
R23 is one of the most important reactions in the troposphere as it leads to $O_3$ formation through R27 and R28.
Provided that there is sufficient insolation to ensure rapid photochemical processing and all species involved are known, the sum
of $HO_2$ and $RO_2$ that react with NO to produce $NO_2$ can be estimated from a **p**hotochemical **s**teady-**s**tate (PSS) assumption in
which production and loss mechanisms are equally important. The $HO_2$ + $RO_2$ concentrations and mixing ratios can be estimated
using the PSS assumption for $NO_2$ by considering the following reactions:
$HO_2$ + NO → OH + $NO_2$      (R23)
$RO_2$ + NO + $O_2$ → $R_{H-1}O$ + $NO_2$ + $HO_2$      (R25 + R26)
$NO_2$ + hν (λ < 400 nm) → NO + O      (R27)



| 98 | $O + O_2 \overset{M}{\to} O_3$ | (R28) |
| 99 | $NO + O_3 \to NO_2 + O_2$ | (R29) |

Assuming a PSS for $NO_2$, this leads to Eq. 1
$$[HO_2 + RO_2]_{PSS} = \frac{k_{NO+O_3}}{k_{NO+(HO_2+RO_2)}} \left( \frac{j_{NO_2}[NO_2]}{k_{NO+O_3}[NO]} - [O_3] \right) \qquad \text{(Eq.1)}$$

where $j_{NO_2}$ is the photolysis frequency of $NO_2$; $k_{NO+O_3}$ ($1.9 \times 10^{-14}$ $cm^3$ molecules$^{-1}$ s$^{-1}$) is the rate coefficient of the reaction of
NO with $O_3$ and $k_{NO+(HO_2+RO_2)}$ is the weighted average rate coefficient assumed for the reactions of peroxy radicals with NO.
The comparison of $[HO_2 + RO_2]_{PSS}$ calculated using Eq.1 with ground-based (e.g. Ridley et al., 1992; Cantrell et al., 1997;
Carpenter et al., 1998; Volz-Thomas et al., 2003), and airborne measurements, has shown in the past different degree of
agreement. The underestimations and overestimations found in air masses with different chemical compositions are not well
understood. In the case of airborne measurements, the PSS calculation generally overestimates that measured peroxy radicals
(Cantrell et al., 2003a, 2003b). The differences observed could not be attributed to systematic changes in NO, altitude, water
vapour and temperature, although these variables are often correlated.
Ground-based (Mihelcic et al., 2003; Kanaya et al., 2007, 2012; Elshorbany et al., 2012; Lu et al., 2012, 2013; Tan et al., 2017,
2018; Whalley et al., 2018, 2021; Lew et al., 2020) and airborne (Crawford et al., 1999; Tan et al., 2001; Cantrell et al., 2003b)
measurements have also been compared with model simulations of $HO_2$ and $RO_2$. The discrepancies encountered depend upon
the chemical composition of the air mass and the chemical mechanisms and constraints used in the model simulations. Tan et al.,
2019 and Whalley et al., 2021 reported experimental radical budget calculations based on the published reaction rate coefficients
of fundamental reactions (R1 to R26) controlling OH, $HO_2$ and $RO_2$ in the lower troposphere and ground-based measurements of
all relevant reactants and photolysis frequencies. In this study, a similar approach has been used to calculate the amount of
peroxy radicals in the air masses measured on-board of the **H**igh **A**ltitude **Lo**ng range (HALO) research aircraft over Europe
during the first campaign of the EMeRGe (**E**ffect of **Me**gacities on the transport and transformation of pollutants on the Regional
to Global scal**e**s) project. The available on-board measurements of $RO_2^*$ are defined as the total sum of OH, RO and peroxy
radicals reacting with NO to produce $NO_2$ (i.e., $RO_2^* = OH + \sum RO + HO_2 + \sum RO_2$, where $RO_2$ are the organic peroxy radicals
reacting with NO to produce $NO_2$). Since the amount of OH and RO is much smaller, $RO_2^*$ to a good approximation is the sum
of $HO_2$ and those $RO_2$ radicals that react with NO to produce $NO_2$. For the calculation, $RO_2^*$ is assumed to be in PSS, and an
analytical expression is developed with a manageable degree of complexity to estimate the concentration and mixing ratios of
$RO_2^*$. The simultaneous on-board measurements of trace gases and photolysis frequencies are used to constrain the estimate of
the $RO_2^*$ concentration.
In contrast to other experimental deployments, the concentrations and/or mixing ratios of the majority of the key species
involved in reactions R1 to R26 were continuously measured on-board HALO during the EMeRGe campaign. This minimises
the number of assumptions required for the calculations of $RO_2^*$. Consequently, this data set provides an excellent opportunity to
gain a deeper insight into the source and sink reactions of $RO_2^*$ and the applicability of the PSS assumption for the different
pollution regimes and related weather conditions in the free troposphere.

## 2. EMeRGe field campaign in Europe

The overarching objective of the EMeRGe project is to test and improve the current understanding of the photochemical and heterogeneous processing of pollution outflows from **m**ajor **p**opulation **c**entres (MPCs) and their impact on the atmosphere. Two **i**ntensive **o**bservational **p**eriods (IOP) were carried out to investigate selected European and Asian MPC outflows. The European IOP took place from 10 to 28 July 2017 (http://www.iup.uni-bremen.de/emerge/home/home.html). An extensive set of in-situ and remote-sensing airborne measurements of trace gases and aerosol particles were made on-board the HALO aircraft (see www.halo-spp.de) along flight tracks in the lower layers of the troposphere from northwest Europe to the Mediterranean region.

HALO carried out a total of 53 flight hours distributed over seven flights to investigate the chemical composition of the outflows from the target MPCs: London, Paris, Benelux/ Ruhr metropolitan area, Po Valley, and urban agglomerations such as Rome, Madrid, and Barcelona. The flight tracks are shown in Fig. 1. All HALO flights started from the HALO base at the DLR in Oberpfaffenhofen, southwest of Munich, Germany. To achieve the scientific goals, 60 % of the flights were carried out below 3000 m. Vertical profiles of trace constituents were typically made in three stable flight levels upwind and downwind of the target MPCs. The flights are named E-EU-FN, where E stands for EMeRGe, EU for Europe, and FN is the two-digit flight number. More details about the EMeRGe IOP in Europe and the set of instruments deployed on-board the HALO aircraft are described elsewhere (Andrés Hernández et al., 2021).

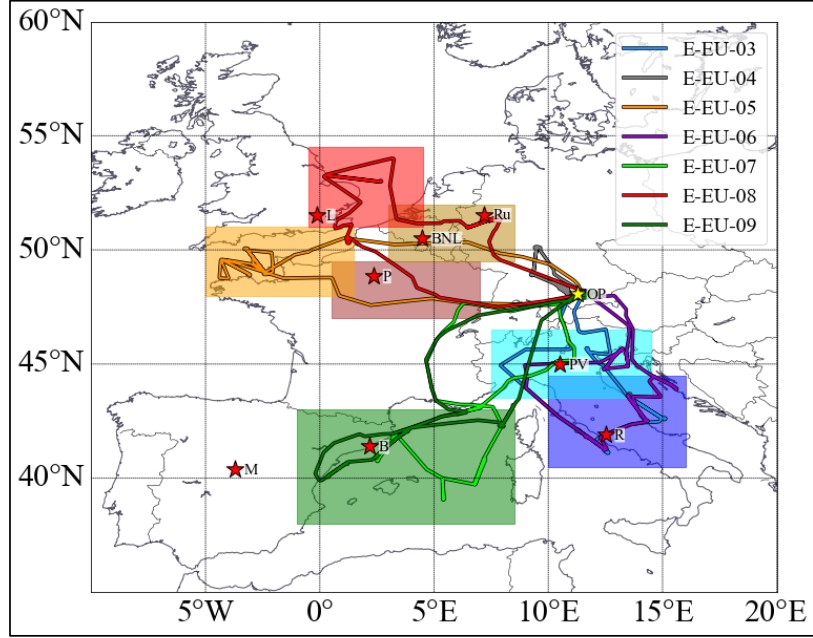

Figure 1: The research flight tracks made by HALO during the EMeRGe-Europe campaign on 11, 13, 17, 20, 24, 26 and 28 July 2017 (E-EU-03 to E-EU-09, respectively, colour coded). MPC target areas are colour coded by shading, and the targeted locations/regions are marked with red stars, M: Madrid, B: Barcelona, P: Paris, L: London; BNL: BeNeLux; Ru: Ruhr area; PV: Po Valley, R: Rome. The location of the HALO base at the DLR in Oberpfaffenhofen, Germany (OP) is indicated by a yellow star.

## 3. PeRCEAS and other instruments on-board

The $RO_2^*$ measurements on-board the HALO research aircraft during EMeRGe were made using the **Pe**roxy **R**adical **C**hemical **E**nhancement and **A**bsorption **S**pectrometer (PeRCEAS). PeRCEAS combines the **Pe**roxy **R**adical **C**hemical **A**mplification



(PeRCA) and **C**avity **R**ing-**D**own **S**pectroscopy (CRDS) techniques in a dual-channel instrument. Each channel has a separate
chemical reactor and detector, which operate alternatively in both background and amplification modes, i.e. without and with the
addition of CO, to account for the rapid background variations during airborne measurements. In the amplification mode, the
sum of the $NO_2$ produced from ambient $RO_2^*$ through the chain reaction, the ambient $NO_2$, the $NO_2$ produced from the ambient
$O_3 – NO$ reagent gas reaction and the $NO_2$ produced in the inlet from any other sources (e.g. thermal decomposition of PAN) is
measured. In the background mode, the sum of the ambient $NO_2$, the $NO_2$ produced from the ambient $O_3 – NO$ reagent gas
reaction and $NO_2$ produced in the inlet from any other sources is measured. The $RO_2^*$ is retrieved by dividing the difference in
$NO_2$ concentration ($\Delta NO_2$) between amplification and background mode by the conversion efficiency of $RO_2^*$ to $NO_2$, which is
referred to as eCL (**e**ffective **c**hain **l**ength). The PeRCEAS instrument and its specifications have been described in detail
elsewhere (Horstjann et al., 2014, George et al., 2020).
The two chemical reactors for sampling the ambient air are part of the **DU**al channel **A**irborne peroxy radical Chemical
Amplifi**er** (DUALER) inlet installed inside a pylon located on the outside of the HALO fuselage. During the EMeRGe campaign
in Europe, a reagent gas mixing ratio of 30 ppmv NO ($[NO] = 1.46 \times 10^{14}$ molecules cm$^{-3}$ at 296 K) and of 9 % CO ($[CO] = 4.4$
$\times 10^{17}$ molecules cm$^{-3}$ at 296 K) were added to the sample flow for the chemical conversion of $RO_2^*$ to $NO_2$. The DUALER inlet
was operated at an internal pressure of 200 mbar to achieve stable chemical conversion. The average eCL under these operational
conditions was determined to be 50 ± 8 from laboratory calibrations, where the error is the standard deviation estimated from the
reproducibility of the experimental determinations. The $HO_2$ and $RO_2$ detection sensitivity depends on the reagent gas NO
concentration due to the rate coefficient of reaction R22 being larger than that for R19. For the measurement conditions used
during the IOP in Europe, the ratio $\alpha = eCL_{CH_3O_2}/eCL_{HO_2}$ is 65% (George et al., 2020).
Although the DUALER pressure is kept constant below the ambient pressure, variations in dynamical pressure > 10 mbar during
the flight can change the residence time and induce turbulences inside the inlet (Kartal et al., 2010; George et al., 2020). These
may lead to different physical losses of radicals before amplification and affect the eCL. In the measurements presented in this
study, variations in dynamical pressure of this magnitude were only encountered during flight level changes of the aircraft. When
used during the analysis, these data sets are either excluded or flagged (P_flag). The effect of the ambient air humidity on eCL
(Mihele and Hastie, 1998; Mihele et al., 1999; Reichert et al., 2003) has been accounted for by a calibration procedure reported
in George et al. (2020).
In addition to the measurement of $RO_2^*$ from PeRCEAS, other in-situ and remote-sensing measurements and basic aircraft data
from HALO are used in this study. Details of the corresponding instruments are summarised in Table 1. Concerning the data
obtained by the remote sensing instruments, the miniDOAS retrieves the Slant Column Density (SCD) of the target gas and a
scaling gas ($O_4$) towards the horizon at the flight altitude. From this, mixing ratios of the targeted gas within the line of sight is
estimated using RT modelling (Stutz et al., 2017; Hüneke et al., 2017; Kluge et al., 2020; Rotermund et al., 2021). The HAIDI
instrument retrieves SCDs below the aircraft. The SCDs from HAIDI are then converted to mixing ratios using the
corresponding geometric Air Mass Factor (AMF) under a well-mixed $NO_2$ layer assumption. As a result of this assumption, the
calculated mixing ratios for HAIDI target gases are lower limits and close to the actual values while flying within and close to a
well-mixed boundary layer. Despite the differences in sampling volume and temporal and spatial resolution in the in-situ and
remote sensing measurement techniques, the concentration of common and related species obtained are in reasonable agreement
(Schumann, 2020).





Table 1: List of the airborne measurements and instrumentation used in this study. PeRCA: Peroxy Radical Chemical Amplification;
CRDS: Cavity Ring-Down Spectroscopy; PTR-MS: Proton-Transfer-Reaction Mass Spectrometer; AT-BS: Adsorption Tube and Bag air
Sampler; TD-GC-MS: Thermal Desorption Gas Chromatography and Mass Spectrometry; DOAS: Differential Optical Absorption
Spectrometry; Univ: University; KIT: Karlsruher Institut für Technologie; DLR: Deutsches Zentrum für Luft- und Raumfahrt; IPA: Institut für
Physik der Atmosphäre; FZ: Forschungszentrum; FX: Flugexperimente.

| **Trace gas-in situ measurements** | | | | |
|---|---|---|---|---|
| Species/parameters | Acronym | Institution | Technique/Instrument | Reference |
| $RO_2^* = HO_2 + \sum RO_2$ | PeRCEAS | Univ. Bremen | PeRCA + CRDS | George et al., 2020 |
| OVOC | HKMS | KIT Karlsruhe | PTR-MS | Brito and Zahn, 2011 |
| $O_3$ | FAIRO | KIT Karlsruhe | UV-Photometry/ Chemiluminescence | Zahn et al., 2012 |
| $O_3$, CO | AMTEX | DLR-IPA | UV-Photometry/ VUV-Fluorimetry | Gerbig et al., 1996 |
| NO, $NO_y$ | AENEAS | DLR-IPA | Chemiluminescence/ Gold converter | Ziereis et al., 2004 |
| $CO_2$, $CH_4$ | CATS | DLR-IPA | CRDS | Chen et al., 2010 |
| **Trace gas- remote sensing measurements** | | | | |
| Species/parameters | Acronym | Institution | Technique/Instrument | Reference |
| $NO_2$, HONO, $CH_2O$, $C_2H_2O_2$, $C_3H_4O_2$ | miniDOAS | Univ. Heidelberg | DOAS / UV-nIR; 2D optical spectrometer | Hüneke et al., 2017 |
| $NO_2$ | HAIDI | Univ. Heidelberg | DOAS / 3x2D-imaging spectrometers | General et al., 2014 |
| **Other parameters** | | | | |
| Species/parameters | Acronym | Institution | Technique/Instrument | Reference |
| Spectral actinic flux density (up/down) Photolysis frequencies | HALO-SR | FZ Jülich | CCD spectro- radiometry | Bohn and Lohse, 2017 |
| Basic aircraft data | BAHAMAS | DLR-FX | various | Mallaun et al., 2015 |

**4. Results and discussion**
**4.1. Airborne $RO_2^*$ measurements during EMeRGe in Europe**
$RO_2^*$ mixing ratios up to 120 pptv were measured during the campaign, as shown in Fig. 2. Typically, the highest $RO_2^*$ mixing
ratios were observed below 3000 m over Southern Europe. This is attributed to the higher insolation and temperatures favouring
the rapid production of $RO_2^*$ from the photochemical oxidations of CO and VOCs.



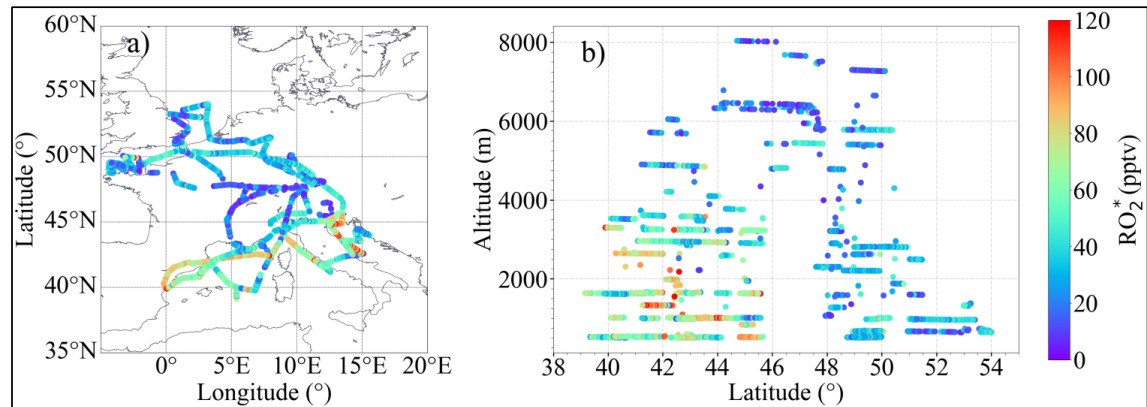

Figure 2: $RO_2^*$ measured during EMeRGe-Europe: a) as a function of longitude and latitude, b) as a function of latitude and
altitude.
The origin and thus the composition of the air sampled during the seven flights over Europe were different and heterogeneous.
Typically, the air masses measured were influenced by emissions from MPCs and their surroundings, and sometimes by biomass
burning transported over short or long distances. The concentration and mixing ratio of $RO_2^*$ depends on the insolation and the
chemical composition of the air masses probed, particularly on the abundance of $RO_2^*$ precursors. Provided that insolation
conditions and a sufficient number of key participating precursors are comparable, the air mass origin is irrelevant for calculating
$RO_2^*$ concentrations and mixing ratios. This is because the $RO_2^*$ concentration is controlled by fast chemical and photochemical
processes. Thus, the $RO_2^*$ variability and production rates provide insight into the photochemical activity of the air masses
probed. Changes in $RO_2^*$ as a function of latitude and altitude, as shown in Fig. 2, confirm the heterogeneity of the
photochemical activity in the air masses probed. Figure 3 shows the $RO_2^*$ vertical profiles averaged for the EMeRGe flights over
Europe in 500 m altitude bins. The error bars are standard errors (i.e. $\pm 1\sigma$ standard deviation of each bin). The vertical profiles
may be biased as the higher altitudes have fewer measurements than those below 3000 m, as mentioned in section 2. The vertical
profiles are a composite from averaging different flights and are shown to summarise the variability in the composition of the air
masses measured during the campaign.





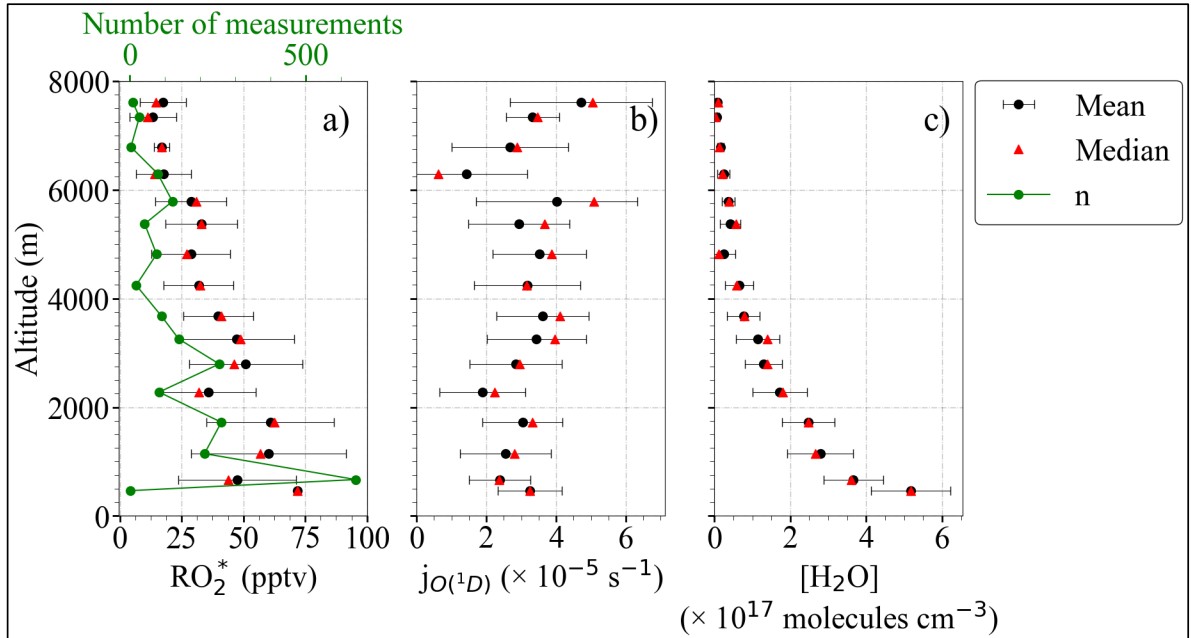

Figure 3: Composite average vertical profiles of a) $RO_2^*$, b) $j_{O(^1D)}$ and c) $[H_2O]$ observations. The measurements are binned over 500 m altitude. The error bars are the standard errors (i.e. $\pm 1\sigma$ standard deviation of each bin). Median values (red triangles) and the number of individual measurements, n, for each bin (in green) are additionally plotted.

Differences between mean and median values indicate less $RO_2^*$ variability in the air masses probed above 3000 m. Most of the EMeRGe measurements below 2000 m were carried out in the outflow of MPCs, which are expected to contain significant amounts of $RO_2^*$ precursors. HALO flew at the lowest altitudes over the English Channel, the Mediterranean and the North Sea. The $H_2O$ concentration in the air masses decreases steadily with altitude as expected. The higher relative variability in $H_2O$ observed at 3000 m and the increase at 5000 m is associated with measurements under stormy conditions, often over the Alps.

### 4.2. $RO_2^*$ production rates

The total production rate of OH and $RO_2^*$ ($P_{OH+HO_2+RO_2}$) can be estimated from the reactions R1 to R13 as follows:

$$P_{OH+HO_2+RO_2} = 2jO_D^1[O_3] \frac{k_{O_D^1+H_2O}[H_2O]}{k_{O_D^1+H_2O}[H_2O]+k_{O_D^1+O_2}[O_2]+k_{O_D^1+N_2}[N_2]} + j_{HONO}[HONO] + 2j_{H_2O_2}[H_2O_2] + 2\sum_i j_i[OVOC_i] +$$

$$\sum_j k_{OH+VOC_j}[OH][VOC_j] + \sum k_{O_3+alkenes_k}[O_3][alkenes_k] \qquad (Eq.\ 2)$$

In this work, Eq. 2 has been applied to the measurements taken within the EMeRGe campaign in Europe. There were no $H_2O_2$ measurements available for EMeRGe IOP. However, from the results reported by Tan et al. (2001), the OH production from the $H_2O_2$ photolysis become significant at low $NO_x$ conditions. Since the $[NO_x] > 8 \times 10^{12}$ molecules cm$^{-3}$ for 60 % of the measurements during the IOP, as a first approximation, the production of OH from $H_2O_2$ photolysis is assumed to be negligible for the dataset considered in this study. Similarly, the VOC photolysis was assumed to dominate the $RO_2^*$ production over the oxidation by OH and ozonolysis of alkenes. The most abundant and reactive oxygenated volatile organic compounds (OVOCs)





measured have been taken as a surrogate for the sum of VOCs. These assumptions lead to Eq. 3, which estimates the $RO_2^*$
production rate ($P_{RO_2^*}$) as:
$P_{RO_2^*} = 2j_{O(^1D)}[O_3] \dfrac{k_{O_D^1 + H_2O}[H_2O]}{k_{O_D^1 + H_2O}[H_2O] + k_{O_D^1 + O_2}[O_2] + k_{O_D^1 + N_2}[N_2]} + j_{HONO}[HONO] + 2j_{HCHO}[HCHO] + 2j_{CH_3CHO}[CH_3CHO] +$
$2j_{CH_3C(O)CH_3}[CH_3C(O)CH_3] + 2j_{CHOCHO}[CHOCHO]$ (Eq.3)
Eq. 3 yields the rate of production of $RO_2^*$ molecules. The production rate can be expressed in units of mixing ratio of $RO_2^*$ by
dividing with the air concentration at each altitude, estimated from the pressure and temperature measurements. Figure 4 shows
the composite averaged vertical profile of all measured $RO_2^*$ mixing ratios colour-coded with the estimated $P_{RO_2^*}$. Small circles
show the 1-minute measurements binned for $P_{RO_2^*}$ up to 0.8 pptv s$^{-1}$ in 0.1 pptv s$^{-1}$ intervals. The production rates above 0.8 pptv
s$^{-1}$ are binned to the 0.8 pptv s$^{-1}$ bin. Larger circles in the figure result from further binning the small circles over 500 m altitude
steps. The error bars are the standard errors for each altitude bin. For the sake of representativeness and comparability, the
number of measurements in each altitude bin is shown in Fig. 4b. Higher $RO_2^*$ mixing ratios observed below 4000 m are
typically associated with $P_{RO_2^*} \geq 0.4$ pptv s$^{-1}$. Above 4000 m both $P_{RO_2^*}$ and $RO_2^*$ start to decrease with altitude, as expected. This
is related to the decrease in $H_2O$ and other radical precursor concentrations with altitude, as detailed in Fig. 5 and Fig. 6. In
previous airborne campaigns at various parts of the world, $RO_2^*$ vertical distributions showed a local maximum between 1500
and 4000 m, as reported by Tan et al. (2001), Cantrell et al. (2003a, 2003b), and Andrés-Hernández et al. (2009). In the present
work, this local maximum is more evident for measurements with $P_{RO_2^*} \geq 0.5$ pptv s$^{-1}$.

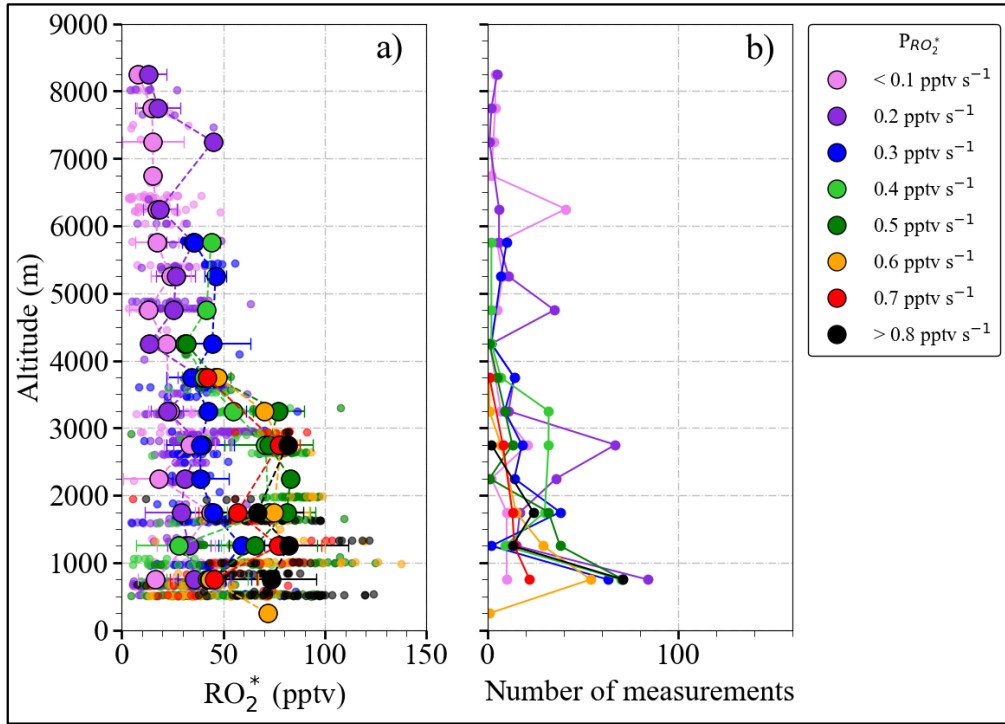


Figure 4: a) Composite averaged vertical distribution of measured $RO_2^*$ colour-coded according to the value of $P_{RO_2^*}$, b) the
number of measurements in each altitude bin. Small circles are 1-minute individual measurements binned with $P_{RO_2^*}$ values in
0.1 ppts$^{-1}$ intervals. Larger circles result from a further binning over 500 m altitude steps. All the production rates below 0.1 pptv
s$^{-1}$ and above 0.8 pptv s$^{-1}$ are binned to 0.1 pptv s$^{-1}$ and 0.8 pptv s$^{-1}$, respectively.





Figure 5 shows the fractional contribution of the production rate from each radical precursor reaction included in Eq. 3 as a
function of altitude. The data are classified into three groups according to the rate of change of production of the $RO_2^*$ mixing
ratio $P_{RO_2^*} < 0.07$ pptv s$^{-1}$ (5a), $0.07 < P_{RO_2^*} < 0.8$ pptv s$^{-1}$ (5b), and $P_{RO_2^*} > 0.8$ pptv s$^{-1}$ (5c) to show the lowest, most common,
and highest ranges, respectively, encountered during the IOP. For 89 % of the measurements, $0.07 < P_{RO_2^*} < 0.8$ pptv s$^{-1}$ applies,
while the rest of the data are equally distributed in the other two $P_{RO_2^*}$ ranges. The data in each group are always binned over 500
m when available.
Typically, the high amount of $H_2O$ in the air masses probed results in the $O_3$ photolysis and subsequent reaction of $O^1D$ with
$H_2O$ (R1-R2a) and is the highest $RO_2^*$ radical production rate ($\geqslant 50$ %) below 4000 m. As the amount of $H_2O$ reduces with
altitude, the relative contribution from $O_3$ photolysis decreases. Above 4000 m, HCHO, HONO, and CHOCHO photolysis
contributions range between 20 % to 40 %, 2.5 % to 30 %, and 5 % to 25 %, respectively. The HCHO contribution increases up
to 80% during measurements above 6000 m. The contributions of $CH_3CHO$ and $CH_3C(O)CH_3$ photolysis are, in contrast,
practically negligible ($< 5$ %).
The vertical changes of the precursor mixing ratios and photolysis frequencies used to calculate $P_{RO_2^*}$ in Fig. 5 are shown in Fig.
6a to 6f. $P_{RO_2^*} < 0.07$ pptv s$^{-1}$ is associated with measurements under cloudy conditions, towards sunset where the photolysis
frequencies are low, or at altitudes above 5000 m in air masses with a low amount of $RO_2^*$ precursors. $P_{RO_2^*} > 0.8$ pptv s$^{-1}$ are
found for air masses, measured below 2000 m in the outflow of MPCs over the sea, for conditions having sufficient insolation
($j_{O(^1D)} > 3 \times 10^{-5}$ S$^{-1}$) and a high content of $RO_2^*$ precursors (HCHO $> 1000$ pptv and HONO $>100$ pptv). The increase in the
photolysis frequencies as a function of altitude is concurrent with decreases in precursor concentrations. As a result, the $P_{RO_2^*}$ do
not significantly vary with altitude in the air masses investigated.
In previous airborne campaigns, Tan et al. (2001) and Cantrell et al. (2003b) reported a reduction of the fractional contribution of
the reaction of $O(^1D)$ with $H_2O$ as the $P_{RO_2^*}$ value decreases. At very low $P_{RO_2^*}$ values ($< 0.03$ pptv s$^{-1}$), the sum of all other
production terms exceeded the fraction from the $O(^1D) + H_2O$ term. For these conditions, $H_2O_2$ and VOCs photolysis dominated
the $P_{RO_2^*}$. In the case of the EMeRGe data set in Europe, only 6 % of $P_{RO_2^*}$ are below 0.06 pptv s$^{-1}$.





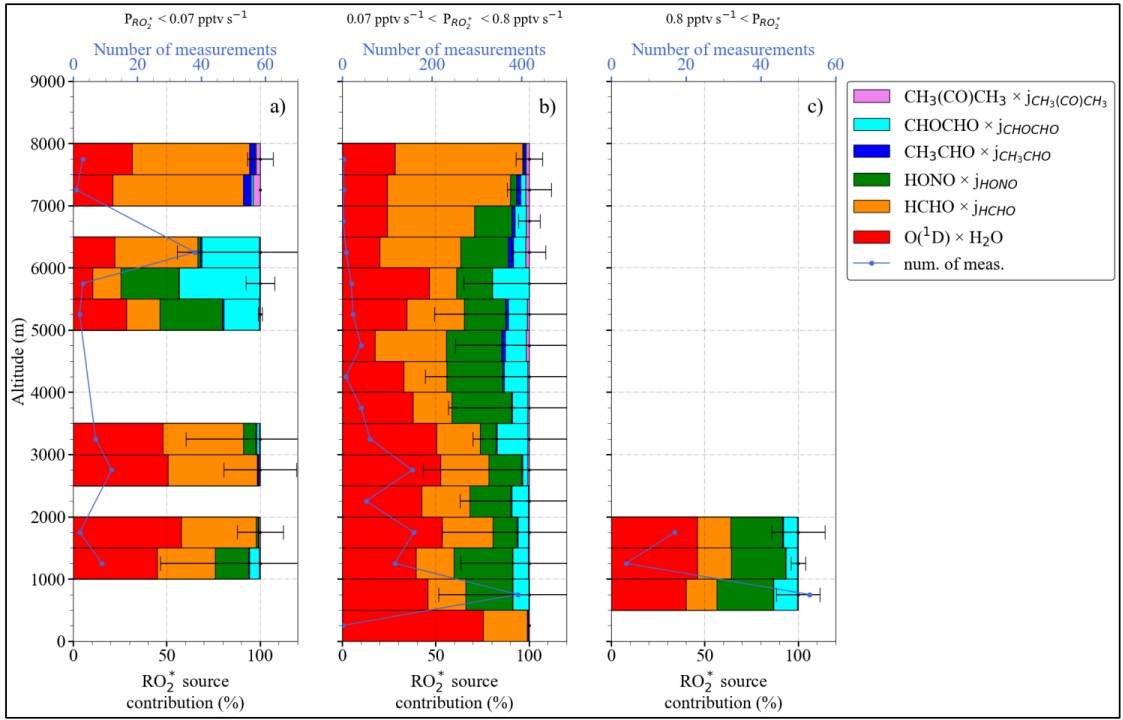


Figure 5: Total $P_{RO_2^*}$ and fractional precursor contributions estimated using Eq. 5 as a function of altitude, for: a) $P_{RO_2^*} < 0.07$ pptv s$^{-1}$, b) 0.07 pptv s$^{-1}$ < $P_{RO_2^*}$ < 0.8 pptv s$^{-1}$, and c) $P_{RO_2^*}$ > 0.8 pptv s$^{-1}$. Note the different scales in the number of measurements.






Figure 6: Vertical distribution and variation of a) to c) precursor mixing ratios; d) to f) photolysis frequencies for the $P_{RO_2^*}$ bins
as in Fig. 5. Note the different scales in the $H_2O$ concentration





### 4.3. PSS estimation of $RO_2^*$ mixing ratios


Under most ambient conditions in the troposphere, the $RO_2^*$, which to a first approximation is the sum of $HO_2$ and $RO_2$, are
short-lived, and the chemical lifetime of $RO_2^*$ is much shorter than the chemical transport time into and out of an air mass being
probed. Consequently, pseudo-steady-state conditions prevail, and the radical production and loss rates are balanced:

$P_{HO_2+RO_2} = L_{HO_2+RO_2}$ (Eq.4)
If the interconversion reactions between OH, RO, $HO_2$ and $RO_2$ (R5 to R7, R12, R16b, and R23 to R26) occur without losses,
then the radical number concentrations are calculated by solving Eq. 4. If the $RO_2^* - RO_2^*$ reactions are assumed to be the
dominant radical loss processes, Eq. 4 leads to Eq. 5.

$$2j_{O(^1D)}[O_3]\frac{k_{O_D^1+H_2O}[H_2O]}{k_{O_D^1+H_2O}[H_2O] + k_{O_D^1+O_2}[O_2] + k_{O_D^1+N_2}[N_2]} + j_{HONO}[HONO] + 2j_{HCHO}[HCHO] + 2j_{CH_3CHO}[CH_3CHO]$$
$$+ 2j_{CH_3C(O)CH_3}[CH_3C(O)CH_3] + 2j_{CHOCHO}[CHOCHO] = k_{RO_2^*+RO_2^*}[RO_2^*]^2$$

(Eq. 5)
where $k_{RO_2^*+RO_2^*}$ is the effective $RO_2^*$ self-reaction rate coefficient, which is defined as the weighted average rate coefficient
between $HO_2 - HO_2$, $HO_2 - RO_2$ and $RO_2 - RO_2$ reactions.
Consequently, the $RO_2^*$ concentrations are expected to correlate with the square root of the $P_{RO_2^*}$.
Figure 7 shows the relationship between the measured $[RO_2^*]$ and the estimated $\sqrt[2]{P_{RO_2^*}}$. Generally, both $[RO_2^*]$ and $\sqrt[2]{P_{RO_2^*}}$
increase with the photolysis frequency of $O_3$ ($j_{O(^1D)}$). The $[RO_2^*] < 0.5 \times 10^{12}$ molecules cm$^{-3}$ and $\sqrt[2]{P_{RO_2^*}} < 1000$ with $j_{O(^1D)} > 5$
$\times 10^{-5}$ belong to the measurements made above 6000 m, where the amount of $RO_2^*$ precursors is low. The relatively weak
correlation observed between $[RO_2^*]$ and $\sqrt[2]{P_{RO_2^*}}$ indicates the presence of other radical loss processes and/or missing production
terms in the $P_{RO_2^*}$ calculation. Apart from this, the spread in the diagram confirms that the effective $RO_2^*$ self-reaction rate
$k_{RO_2^*+RO_2^*}[RO_2^*]^2$ varies widely in the air masses probed due to the effect of changes in $HO_2$ and $\sum RO_2$ concentrations in the
individual loss reaction rate coefficients. As mentioned in section 4.1, photochemical processing was expected to be enhanced
over Southern Europe due to the prevailing high insolation and temperatures during the measurements. This is also reflected in
the higher $P_{RO_2^*}$ and $[RO_2^*]$ observed in Southern Europe as compared to those in Northern Europe (Fig. 7b).

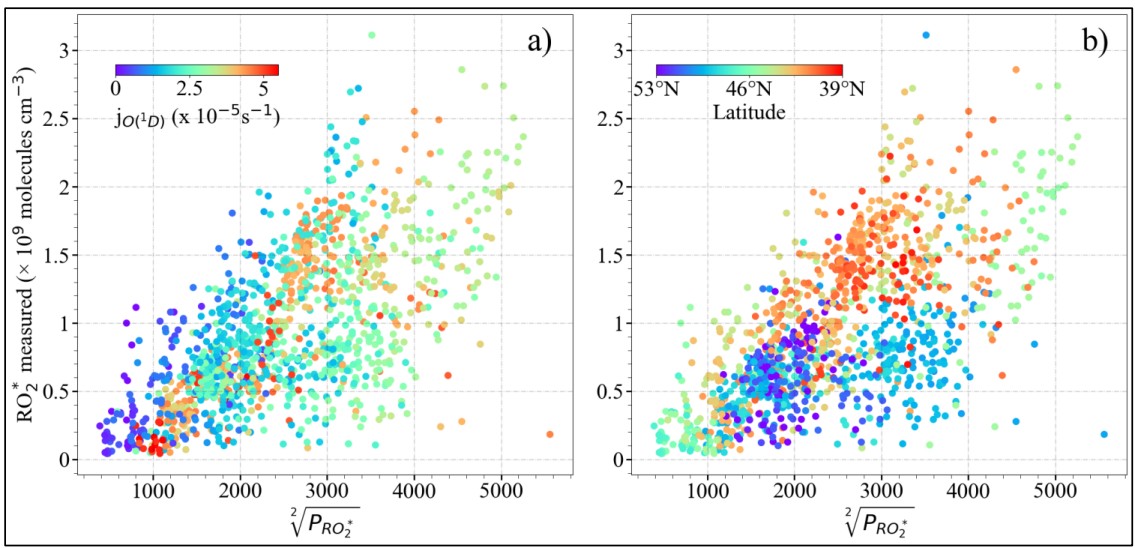


Figure 7: Plot of the measured $[RO_2^*]$ versus estimated $\sqrt[2]{P_{RO_2^*}}$ colour-coded for values of a) $j_{O(^1D)}$ and b) latitude.
The correlation between $[RO_2^*]$ and $\sqrt[2]{P_{RO_2^*}}$ improves when the measurements south and north of 47°N are separately analysed
(Fig. 8). For a given $[RO_2^*]$, the $P_{RO_2^*}$ calculated is higher for the measurements north of 47°N than south of 47°N. The lowest
$[RO_2^*]$ to $\sqrt[2]{P_{RO_2^*}}$ ratios are associated with higher $NO_x$ (NO + $NO_2$), especially north of 47°N, indicating the urban character and
higher content in $RO_2^*$ precursors of the air probed (Fig. 8d). Note that these results are only valid for the data set acquired over
Europe during EMeRGe and do not yield a relationship between $[RO_2^*]$ and $\sqrt[2]{P_{RO_2^*}}$, which is generally applicable for these two
latitude windows.

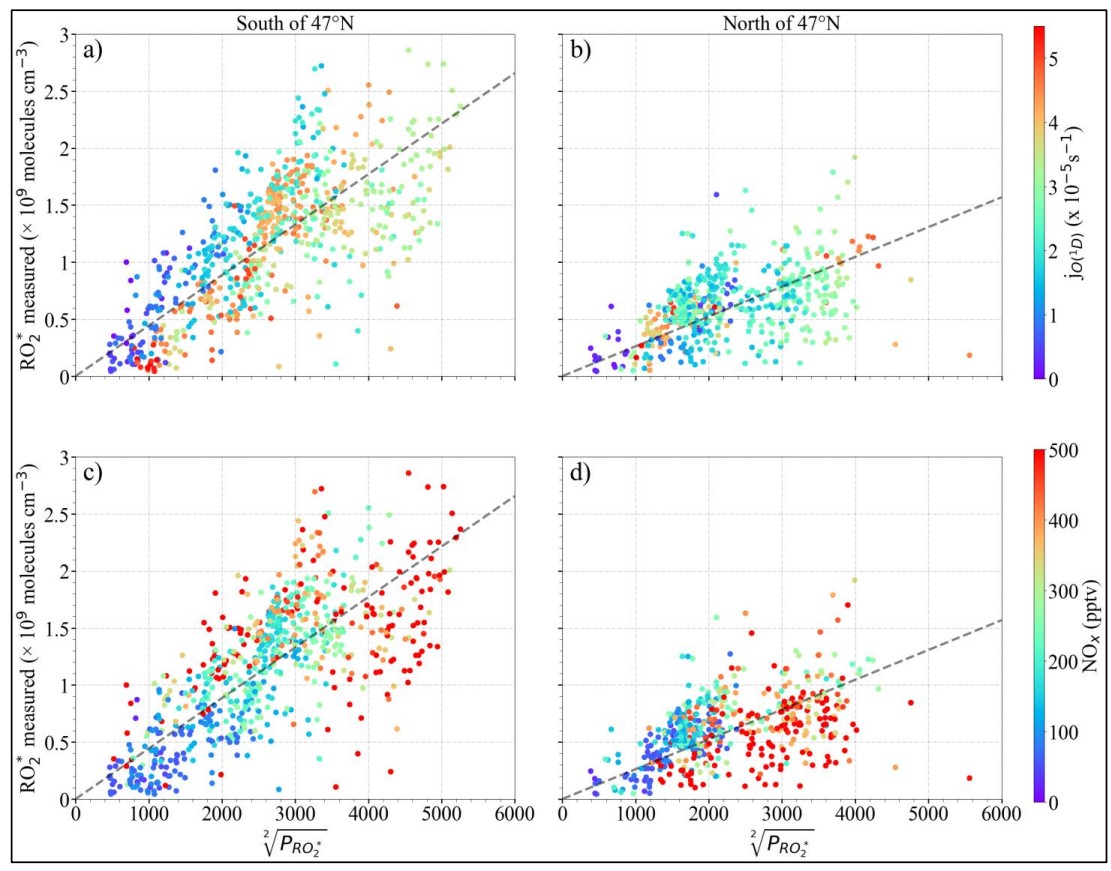

Figure 8: Plots of the measured $[RO_2^*]$ vs $\sqrt[2]{P_{RO_2^*}}$ for the following latitudes: a) and c) south of 47°N; b) and d) north of 47°N. Note that a) and b) are colour-coded with $j_{O(^1D)}$; c) and d) are colour-coded by $NO_x$ mixing ratio. The dashed lines indicate the linear fit for visual support.

The relationship between $RO_2^*$ and $P_{RO_2^*}$ is further investigated to identify the dominant $RO_2^*$ loss process in the air masses considered in this study. As stated in section 3, $HO_2$ and $RO_2$ are not speciated but retrieved as $RO_2^*$ by the PeRCEAS instrument. So, the effect of changes in the $HO_2$ to the total $RO_2^*$ ratios, represented by δ, i.e., $[HO_2] = \delta[RO_2^*]$ and $[CH_3O_2] = (1-\delta)[RO_2^*]$, is investigated. As a first approach, $CH_3O_2$ reactions are taken as a surrogate for all $RO_2$ reactions to reduce the complexity of the calculations. Consequently, Eq. 5 is accordingly modified:

$$2j_1[O_3]\beta + j_3[HONO] + 2j_8[HCHO] + 2j_9[CH_3CHO] + 2(j_{10a} + j_{10b})[CH_3C(O)CH_3] + 2j_{11}[CHOCHO] = 2k_{15}\delta(1-\delta)[RO_2^*]^2 + 2k_{16a}\left((1-\delta)[RO_2^*]\right)^2 + 2k_{14}(\delta[RO_2^*])^2 \quad\text{(Eq. 6)}$$

where β is the effective yield of OH production in the reaction of $O(^1D)$ with $H_2O$ given by:

$$\beta = \left(\frac{k_{2a}[H_2O]}{k_{2a}[H_2O] + k_{2b}[O_2] + k_{2c}[N_2]}\right)$$

From Eq. 6, $[RO_2^*]$ can be calculated as





$$[RO_2^*] = \sqrt[2]{P_{RO_2^*}\big/2k_{RO_2^*}}$$ (Eq. 7)
where
$$k_{RO_2^*} = (k_{15}\delta(1-\delta) + k_{16a}(1-\delta)^2 + k_{14}\delta^2)$$

$$P_{RO_2^*} = 2j_1[O_3]\beta + j_3[HONO] + 2j_8[HCHO] + 2j_9[CH_3CHO] + 2(j_{10a} + j_{10b})[CH_3C(O)CH_3] + 2j_{11}[CHOCHO]$$

The second solution gives negative values for $[RO_2^*]$, therefore has no physical meaning. A more detailed derivation of Eq. 6 and
Eq. 7 are given in the supplementary information.
Figure 9 shows the measured $RO_2^*$ ($RO_2^*{}_m$) mixing ratio versus the calculated $RO_2^*$ ($RO_2^*{}_c$) mixing ratio using Eq. 7, colour-
coded with respect to the NO mixing ratios. $RO_2^*{}_m$ and $RO_2^*{}_c$ are the measured and calculated $RO_2^*$ respectively for $\delta = 1$, i.e.
$RO_2^* = HO_2$ and $\delta = 0.5$, i.e. $HO_2 = RO_2$. The eCL corresponding to $\delta = 1$ and $\delta = 0.5$ used for the $RO_2^*{}_m$ retrievals were
determined in laboratory experiments, as reported by George et al. (2020). The small circles represent 1-minute $RO_2^*{}_m$, whereas
the large circles are the mean of the $RO_2^*{}_m$ binned over 10 pptv $RO_2^*{}_c$ intervals. Despite the limited number of production and
loss processes considered, $RO_2^*{}_c$ reasonably agrees with $RO_2^*{}_m$ as indicated by the fit parameters (Table 2). $RO_2^*{}_c$ often
overestimates $RO_2^*{}_m$ for NO mixing ratios above 250 pptv. The overestimation is also evident for $RO_2^*{}_m$ below 40 pptv. This
may be due to the HO and RO losses during the radical interconversion by reacting with $NO_x$ producing HONO, $HNO_3$ and
organic nitrate.

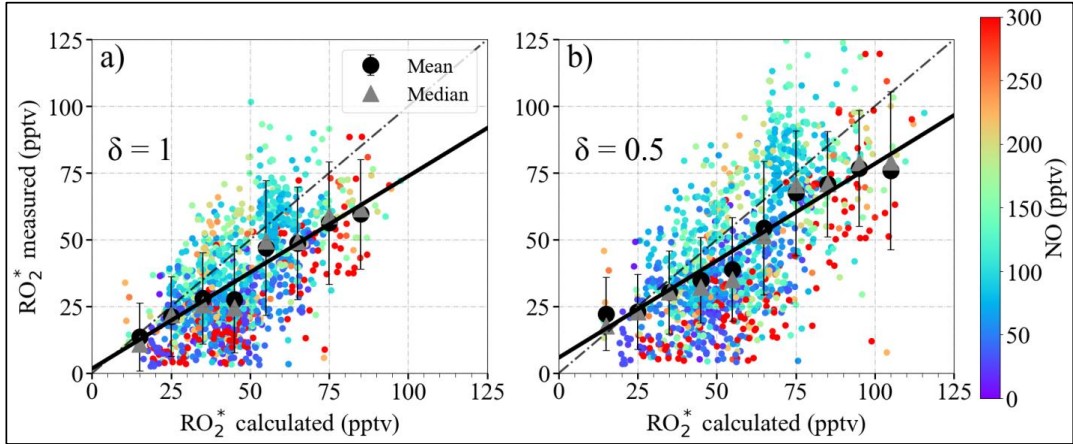


Figure 9: $RO_2^*$ measured ($RO_2^*{}_m$) versus PSS $RO_2^*$ calculated ($RO_2^*{}_c$) using Eq. 7 for a) $\delta = 1$, b) $\delta = 0.5$ by assuming only $RO_2^*$
– $RO_2^*$ loss reactions. The 1-minute (small circles), the mean of the binned $RO_2^*{}_m$ over 10 pptv $RO_2^*{}_c$ intervals (large circles),
and the median of each bin (grey triangles) are shown. The error bars indicate the standard error of each bin. The linear
regression for the binned values (solid line) and the 1:1 relation (dashed line) are also plotted for reference. The fit parameters
are given in Table 2.
The PSS data presented in Fig, 9 are calculated assuming interconversion reactions between OH, RO and $RO_2^*$ occur without
losses and the limiting case of $[OH] <<< [HO_2+ RO_2]$ in the case of a low amount of NO and $NO_2$. Furthermore, VOC oxidation
processes are not considered as a source of radicals. To identify other major loss processes, Eq. 6 is extended with radical
conversion reactions between oxy and peroxy radicals and OH and RO losses through HONO, $HNO_3$, and organic nitrate





formation. In the resulting Eq. 8, $CH_4$, HCHO, $CH_3CHO$, CHOCHO, $CH_3OH$, and $CH_3C(O)CH_3$ measured on-board HALO are
taken as surrogates for the dominant VOC acting as $RO_2^*$ precursors through oxidation:
$(2j_1[O_3]\beta + j_3[HONO])(1 - \rho) + 2j_8[HCHO] + 2j_9[CH_3CHO] + 2(j_{10a} + j_{10b})[CH_3C(O)CH_3] + 2j_{11}[CHOCHO] =$
$\delta[RO_2^*](k_{23}[NO] + k_{24}[O_3])\rho + \left(2k_{16b}((1-\delta)[RO_2^*])^2 + k_{25}(1-\delta)[RO_2^*][NO]\right)\left(\frac{k_{22}[NO]}{(k_{22}[NO]+k_{26}[O_2])}\right) + 2k_{15}\delta(1-$
$\delta)[RO_2^*]^2 + 2k_{16a}\left((1-\delta)[RO_2^*]\right)^2 + 2k_{14}(\delta[RO_2^*])^2$                     (Eq. 8)
where $\beta$ is the OH production efficiency of the $O_3$ photolysis and $\delta$ is the $HO_2$ to $RO_2^*$ ratio as in Eq. 6, $\rho$ is the OH loss during
the OH – $RO_2^*$ interconversion. As in Eq. 6, $CH_3O_2$ is taken as a surrogate for all $RO_2$. The detailed derivation of Eq. 8 is given
in the supplementary information.
During the IOP in Europe, HCHO and $CH_3CHO$ are the dominant radical precursors from OVOC oxidations. Their impact on
the radical budget is similar because their respective concentrations compensate the difference in the rate coefficients of their
reactions with OH ($k_{OH+HCHO} = 8.5 \times 10^{-12}$ cm$^3$ molecule$^{-1}$ s$^{-1}$ and $k_{OH+CH_3CHO} = 1.5 \times 10^{-11}$ cm$^3$ molecule$^{-1}$ s$^{-1}$). Despite the
high mixing ratios measured, $CH_3C(O)CH_3$ is a less important source of $RO_2^*$. This is because the rate coefficient
$k(T)_{OH+CH_3C(O)CH_3}$ is significantly slower than $k_{OH+HCHO}$ and $k_{OH+CH_3CHO}$ (see Table S1 in the supplement). Similarly, the
$RO_2^*$ production rate of CHOCHO and $CH_3OH$ through oxidation is an order of magnitude lower than that of HCHO and
$CH_3CHO$. Since $k_{HO_2+O_3}$ is almost four orders of magnitude smaller than $k_{HO_2+NO}$ and the NO concentrations remained about
three orders of magnitude smaller than the $O_3$ measured, the $HO_2$ reaction with $O_3$ had a negligible effect in Eq. 8.
The impact of the methylglyoxal ($CH_3C(O)C(O)H$) photolysis was also investigated by using the $CH_3C(O)C(O)H^*$
measurements provided by the miniDOAS instrument. The $CH_3C(O)C(O)H^*$ measured is the sum of $CH_3C(O)C(O)H$, and a
fraction of other substituted dicarbonyls (mainly 2,3-butanedione, $C_3H_6O_2$), with similar visible absorption spectra. For the
calculation, $CH_3C(O)C(O)H$ was assumed to be half of $CH_3C(O)C(O)H^*$ as recommended by Zarzana et al. (2017) and Kluge et
al. (2020). The $RO_2^*$ calculated by including $CH_3C(O)C(O)H$ photolysis systematically overestimated the measurements. Since
the adequacy of the recommended factor of 0.5 varies with the actual air mass composition, $CH_3C(O)C(O)H$ was not included in
the calculations.
The revised PSS $[RO_2^*]$ is then calculated from Eq. 8, as:
$[RO_2^*] = \frac{-(-L_{RO_2^*}) - \sqrt[2]{L_{RO_2^*}^2 - 4(-2k_{RO_2^*})P_{RO_2^*}}}{2(-2k_{RO_2^*})}$                     (Eq. 9)
where
$k_{RO_2^*} = \left(\left(k_{16b}\left(\frac{k_{22}[NO]}{(k_{22}[NO]+k_{26}[O_2])}\right)\right) + k_{16a}\right)(1-\delta)^2 + k_{15}\delta(1-\delta) + k_{14}\delta^2\right)$

       $L_{RO_2^*} = \left(\delta(k_{23}[NO] + k_{24}[O_3])\rho + \left(\frac{k_{22}[NO]}{(k_{22}[NO]+k_{26}[O_2])}\right)k_{25}(1-\delta)[NO]\right)$

       $P_{RO_2^*} = (2j_1[O_3]\beta + j_3[HONO])(1-\rho) + 2j_8[HCHO] + 2j_9[CH_3CHO] + 2(j_{10a}+j_{10b})[CH_3C(O)CH_3] + 2j_{11}[CHOCHO]$

Applying Eq. 9 to the measured dataset reduces the overestimation of $RO_{2m}^*$ by $RO_{2c}^*$ at NO mixing ratios higher than 250 pptv
(Fig. 10), especially for $RO_{2m}^* < 40$ pptv but does not introduce significant changes in the overall correlations (Table 2). As in





Fig. 9, plots of the 1-minute $RO_2^*{}_m$ and the average of $RO_2^*{}_m$ binned over 10 pptv $RO_2^*{}_c$ intervals versus $RO_2^*{}_c$ are depicted for $\delta$
= 1 and $\delta$ = 0.5 in Fig. 10. The $RO_2^*$ data are colour-coded with the on-board NO measurements. The linear regression slopes are
around 0.7 ($R^2$= 0.96), indicating an overall 25 – 30 % underestimation of the $RO_2^*{}_m$. However, the $RO_2^*{}_m$ are mostly
overestimated 4000 m under low insolation and underestimated in polluted plumes measured below 2000 m with NO mixing
ratios approximately above 50 pptv (see Fig. 11 for $\delta$ = 0.5). The y-axis intercept is below the instrumental detection limit for
most measurement conditions.
Table 2: Linear regression parameters from $RO_2^*{}_m$ versus $RO_2^*{}_c$ using Eq. 7 and Eq. 9 from Fig. 9 and Fig.10, respectively.

| Formula used to calculate $RO_2^*$ | $\delta$ | slope | y-intercept (pptv) | $R^2$ |
|---|---|---|---|---|
| Eq. 7 | 1.00 | 0.72 | 2 | 0.96 |
| | 0.50 | 0.73 | 6 | 0.96 |
| Eq. 9 | 1.00 | 0.71 | 5 | 0.96 |
| | 0.50 | 0.74 | 6 | 0.97 |

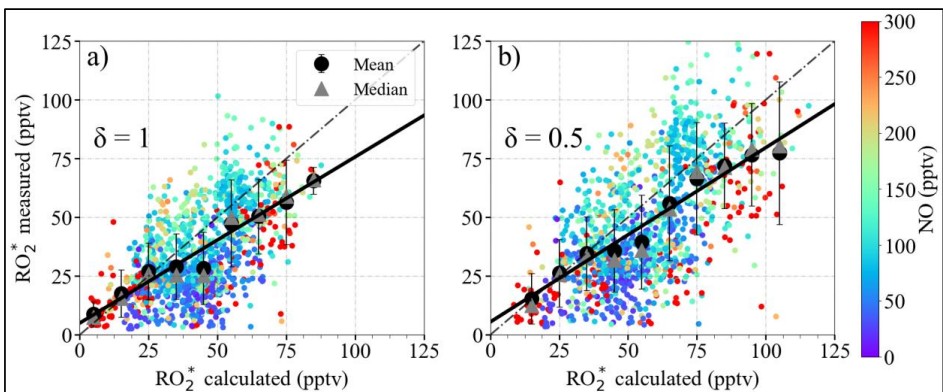


Figure 10: $RO_2^*{}_m$ versus $RO_2^*{}_c$ calculated using Eq. 9 for a) $\delta$ = 1 and b) $\delta$ = 0.5. The data are colour-coded with the measured
NO mixing ratios. The 1-minute (small circles), the mean of the binned $RO_2^*{}_m$ over 10 pptv $RO_2^*{}_c$ intervals (large circles), and
the median of each bin (grey triangles) are shown. The error bars indicate the standard error of each bin. The linear regression for
the binned values (solid line) and the 1:1 relation (dashed line) are also depicted for reference.



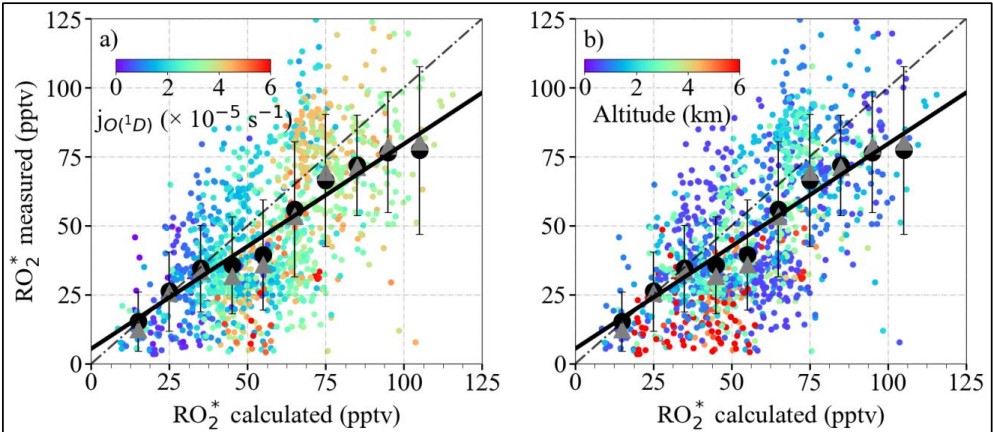


Figure 11: $RO_2^*{}_m$ versus $RO_2^*{}_c$ calculated using Eq. 9 for $\delta = 0.5$. The data points are colour-coded for a) photolysis frequency of
$O_3$; b) altitude. The 1-minute (small circles), the mean of the binned $RO_2^*{}_m$ over 10 pptv $RO_2^*{}_c$ intervals (large circles), and the
median of each bin (grey triangles) are shown. The error bars indicate the standard error of each bin. The linear regression for the
binned values (solid line) and the 1:1 relation (dashed line) are also depicted for reference.

Figure 12 shows the data for $\delta = 0.5$ colour-coded with NO, $NO_x$, the sum of HCHO, $CH_3CHO$, CHOCHO, $CH_3OH$, and
$CH_3C(O)CH_3$ (from now on referred to as $\Sigma$VOCs), as a surrogate for the amount of OVOCs acting as $RO_2^*$ precursors, and the
$\Sigma$VOCs to NO ratio. The largest differences between $RO_2^*{}_m$ and $RO_2^*{}_c$ is observed for the bins around 50 pptv. The $RO_2^*{}_m < 25$
pptv observed above 4000 m are overestimated for air masses with low insolation, i.e. $j_{O(^1D)} < 2 \times 10^{-5}$ s$^{-1}$ (Fig. 11), NO < 50
pptv, $\sum$VOCs typically below 4 ppbv, and high $\Sigma$VOCs/NO ratios ( > 50). Under these insolation conditions, the radical
production rate is expected to be low, and the $RO_2^* - RO_2^*$ reactions are expected to dominate the $RO_2^*$ loss processes. Since OH
and $H_2O_2$ were not measured during the EMeRGe campaign in Europe, Eq. 9 does not include the loss reactions R17 and R18,
which might be significant under such conditions and explain the $RO_2^*$ overestimation. This is also the case for the
overestimations observed below 40 pptv $RO_2^*{}_m$ at other altitudes, where NO < 50 pptv but the $\Sigma$VOCs/NO ratios remain low.
The overestimation may therefore be independent of the $\Sigma$VOCs/NO ratios. For NO $\leq$ 50 pptv, $NO_2 \leq$ 100 pptv, $RO_2^* \leq$ 40 pptv
and HCHO $\leq$ 1 ppbv, the rate of reaction R18, which forms $H_2O$ and $O_2$ from OH and $HO_2$, is about 4 times faster than the rate
of the OH oxidation reaction of the dominant OVOCs (R12) considered in this study or the rate of formation of HONO (R19).

$RO_2^*{}_m$ is systematically underestimated for $\sum$VOCs greater than 7 ppbv. The composition of these air masses is quite different,
as reflected by the $\Sigma$VOCs/NO ratios. This implies that Eq. 9 does not capture the peroxy radical production adequately from
VOCs in these cases. The underestimation of $RO_2^*{}_m$ may be explained in part by a) OH recycling through additional VOC
oxidation processes, which are not in Eq. 9 and/or b) $RO_2^*$ production from the photolysis of carbonyls, which were not
measured and/or c) $RO_2^*$ production from the ozonolysis of alkenes.





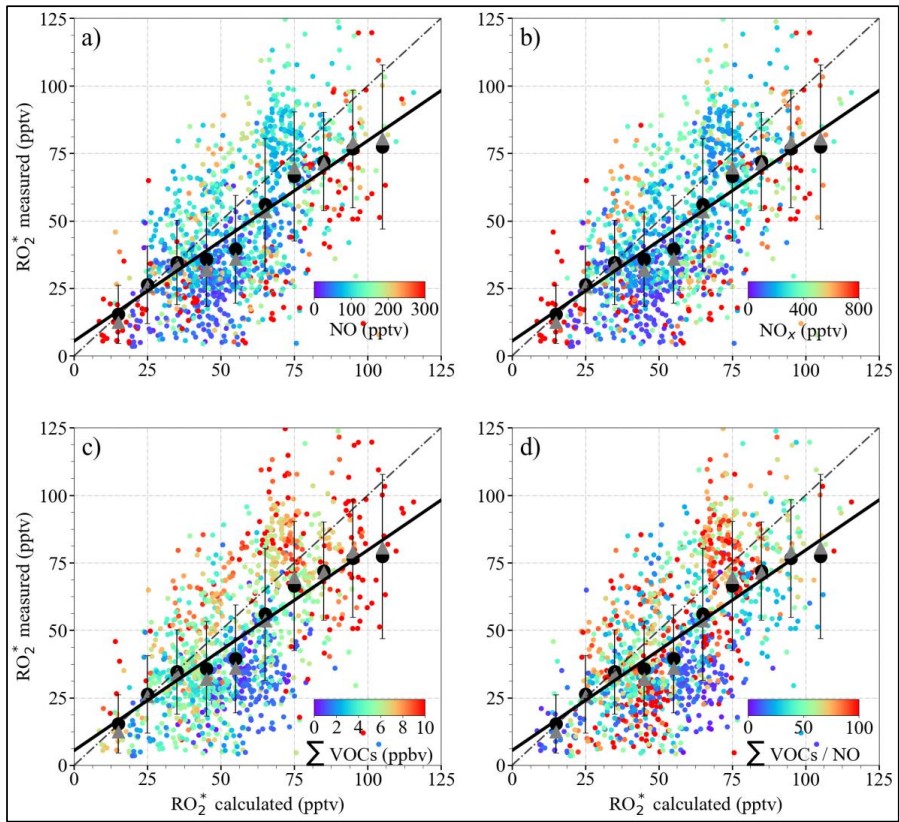

425

Figure 12: $RO_2^*{}_m$ versus $RO_2^*{}_c$ using Eq. 9 for $\delta = 0.5$ colour-coded with the measured a) NO mixing ratio, b) $NO_x$ mixing ratio, c) $\Sigma VOCs$ mixing ratio, where $\Sigma VOCs = HCHO + CH_3CHO + (CHO)_2 + CH_3OH + CH_3C(O)CH_3$, and d) $\Sigma VOCs/NO$ ratio. The 1-minute (small circles), the mean of the binned $RO_2^*{}_m$ over 10 pptv $RO_2^*{}_c$ intervals (large circles), and the median of each bin (triangles) are shown. The error bars represent the standard error of each bin. The linear regression for the binned values (solid line) and the 1:1 relationship (dashed line) are plotted for reference.

Spatial and temporal differences in the in-situ measurements of the key trace gases ($O_3$, NO, $H_2O$, CO, $CH_4$, VOCs) with respect
to remote sensing observations ($NO_2$ and HONO) used in Eq. 9 may also contribute to the overall spread observed in Fig. 12.
Although the temporal evolution and the amount of the trace gases measured using in-situ and remote sensing instruments agree
reasonably well, as shown for HCHO in Fig.13, the remote sensing instruments have, in general, larger air sampling volumes
compared to that of in-situ instruments. This may occasionally lead to significant differences depending on the location of the
pollutant layers with respect to HALO. In addition, PTR-MS measurements of HCHO might include interferences from
molecular fragments of other compounds in the sample air (Inomata et al., 2008). Further details about the accuracy and
comparability of the instrumentation on-board during the campaign can be found elsewhere (Schumann, 2020).

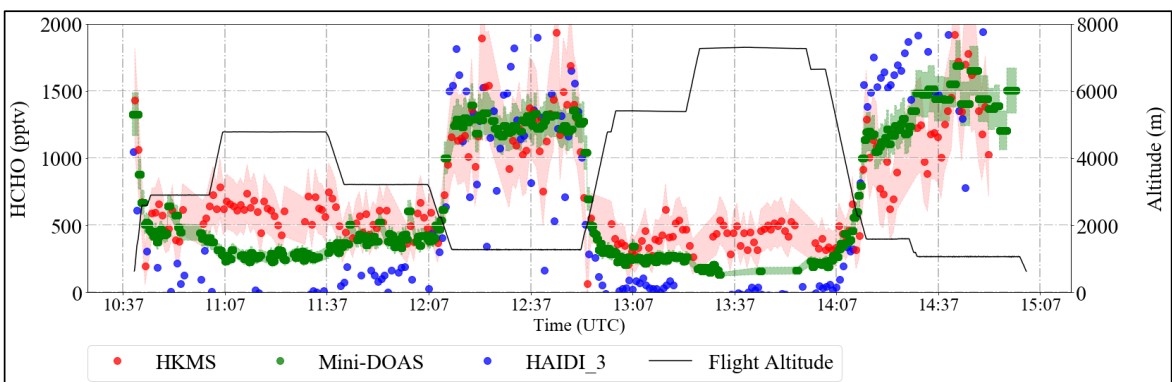


Figure 13: An example of the time series of the measured HCHO mixing ratios retrieved from the remote sensing (HAIDI in blue and miniDOAS in green) and in-situ (HKMS in red) instruments during the E-EU-04 flight on 14.07.2017. The shaded region shows ± 1σ uncertainties of the HKMS and miniDOAS instruments. The flight altitude is depicted in black.

In summary, the differences between $RO_2^*{}_m$ and $RO_2^*{}_c$ might be caused by a combined effect of the limitations of the analytical expression to simulate complex non-linear chemistry and the measurement uncertainties arising from the spatial heterogeneity of the plume for the remote sensing instruments. Consequently, individual analysis of the pollution events encountered along the flights is required to quantify limiting factors in Eq. 9.

The ratio of $RO_2^*{}_m$ to $RO_2^*{}_c$ ($RO_2^*{}_m/RO_2^*{}_c$) has been used to assess the applicability of Eq. 9 for the calculation of $RO_2^*$ in the air masses probed (Fig. 14). In Fig. 14, the data are colour-coded with respect to $RO_2^*{}_m/RO_2^*{}_c$, $H_2O$, $\Sigma$VOCs, and $NO_x$. The air masses probed at altitudes above 2000 m are close to the PSS assumptions used to develop Eq. 9, and consequently, the $RO_2^*{}_m/RO_2^*{}_c$ remains ≤ 1. In contrast, $RO_2^*{}_m/RO_2^*{}_c$ is at its highest value below 2000 m, reaching up to 3. At these altitudes, most of the flights in Europe were carried out in pollution plumes, in which both the amount of $NO_x$ and $RO_2^*$ precursors are high. The analytical expression does not capture the $RO_2^*$ variations resulting from fast non-linear photochemistry present in these pollution plumes. This is the case for the measurements made between 42°N and 46°N in the outflow of Po Valley and Rome. $\Sigma$VOCs > 7 ppbv and $NO_x$ mixing ratios > 500 pptv indicate high radical precursor loading and relatively fresh emissions. The $RO_2^*{}_m/RO_2^*{}_c$ is also > 2 in the measurements over the English Channel (between 50°N and 52°N) with $\Sigma$VOCs and $NO_x$ mixing ratio > 7 ppbv and 1000 pptv, respectively.



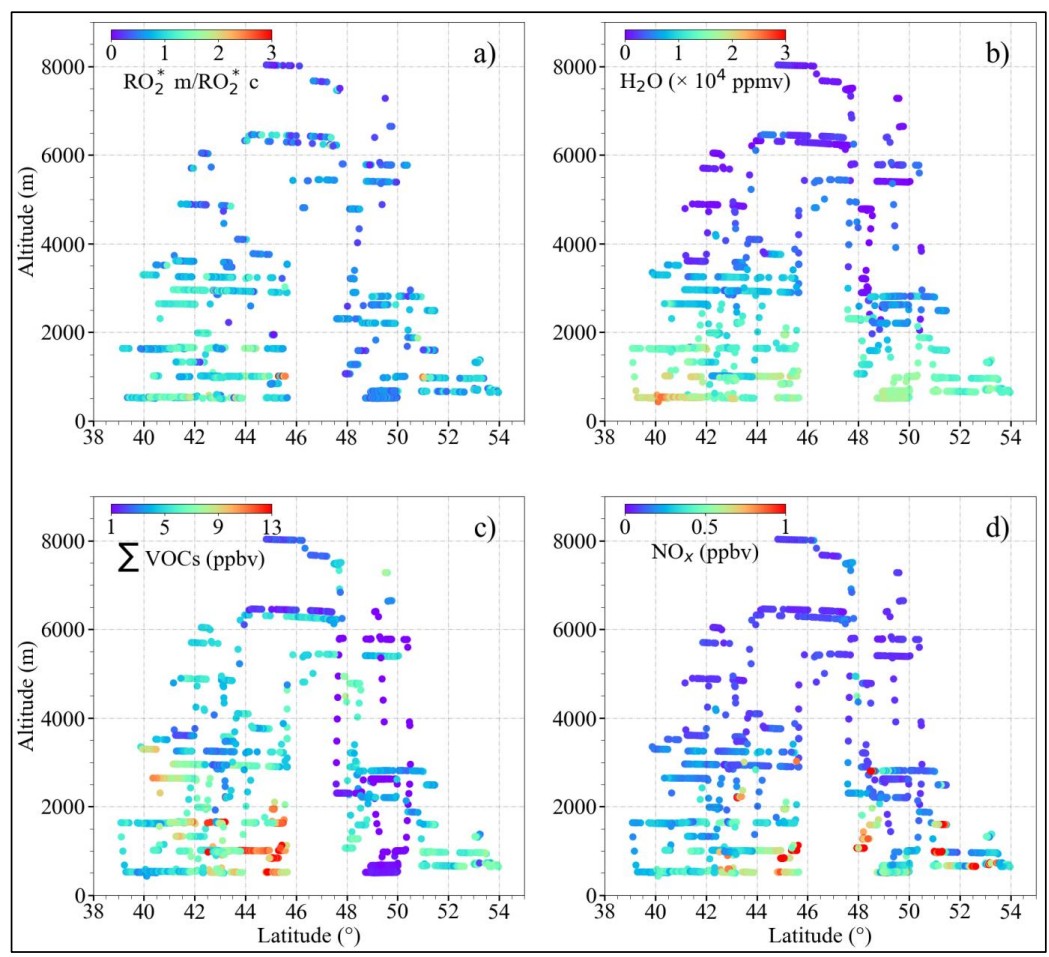

Figure 14: Plots of a) the ratio of $RO_2^*{}_m$ to $RO_2^*{}_c$ ($RO_2^*{}_m/RO_2^*{}_c$) assuming that $\delta = 0.5$; b) $H_2O$; c) $\sum$VOCs; d) $NO_x$ as a function of latitude and altitude for the EMeRGe measurements in Europe.

The applicability of Eq. 9 for calculating the in-flight measurements of $RO_2^*$ along the track of the E-EU-03 flight on 11 July 2017 is shown in Fig. 16. The E-EU-03 flight investigated the outflow of selected MPCs in Italy (i.e., Po Valley and Rome). Consequently, the flight track was routed along the western coast of Italy and included vertical profiling over the Tyrrhenian Sea upwind of Rome (Fig. 15). As can be seen in $j_{O(^1D)}$, cloudless conditions dominated throughout the flight track. The $RO_2^*{}_c$ agree reasonably well with $RO_2^*{}_m$ throughout this period except in the pollution plume measured from 12:05 to 12:25 UTC. In this plume, CO, NO, NO₂, HONO, NOᵧ, and HCHO reached 100 ppbv, 180 pptv, 150 pptv, 120 pptv, 1ppbv and 2 ppbv, respectively. The $RO_2^*$ measurements are approximately 20 % underestimated during this period. Backward trajectories calculated using FLEXTRA indicate the transport of pollution through the Mediterranean mixed with dust plumes originating from Tunisia. The NO mixing ratios observed indicate the proximity to emission sources.



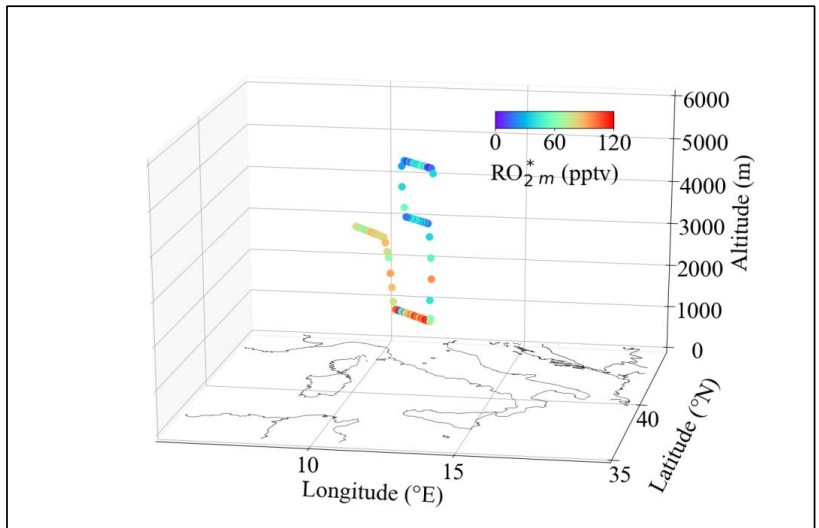


Figure 15: Maps of the flight track of E-EU-03 11 July 2017 along the western coast of Italy over the Tyrrhenian Sea colour-
coded with $RO_2^*$ measurements.
The measurements of VOCs used in Eq. 9 may not be representative of the actual complex VOC composition in the plume
measured from 12:05 to 12:25 UTC. Consequently, the $RO_2$ to $HO_2$ ratio and the RO intermediates involved in the radical
interconversion processes, the branching ratios and effective rate coefficients for $RO_2^* - RO_2^*$ reactions might not be well
represented in Eq. 9. Taking $CH_3O_2$ as a surrogate for all $RO_2$ might lead to uncertainties in the $RO_2^*$ calculations in the presence
of OVOCs with larger organic chains. On the experimental side, changes in the $HO_2$ to $RO_2$ ratio affect the accuracy of the
PeRCEAS retrieval of the total sum of radicals. As noted in section 3, in this study $RO_2^* = HO_2 + 0.65 \times RO_2$, and the eCL is
determined for a 1:1 mixture of $HO_2$:$CH_3O_2$, i.e. $\delta = 0.5$ is used for the $RO_2^*$ retrieval. However, the $HO_2$ to $CH_3O_2$ ratio is not
expected to remain constant in all the air masses probed. For a 3:1 ratio of $HO_2$:$RO_2$, the $RO_{2\,m}^*$ would decrease by 10 %.
Similarly, a $HO_2$:$RO_2$ ratio of 1:3 would lead to an increase of 10 % in the reported $RO_{2\,m}^*$. This uncertainty is well below the in-
flight uncertainty of the PeRCEAS instrument indicated by the error bars in Fig. 16 (George et al., 2020), and cannot account for
the overall 20 % underestimations. However, it might reduce the differences observed between $RO_{2\,m}^*$ and $RO_{2\,c}^*$ in particular
cases. A complete explanation of the variability of $RO_2^*$ in the pollution plumes measured within the IOP in Europe is beyond
the scope of this analysis and requires an investigation by high-resolution chemical models.


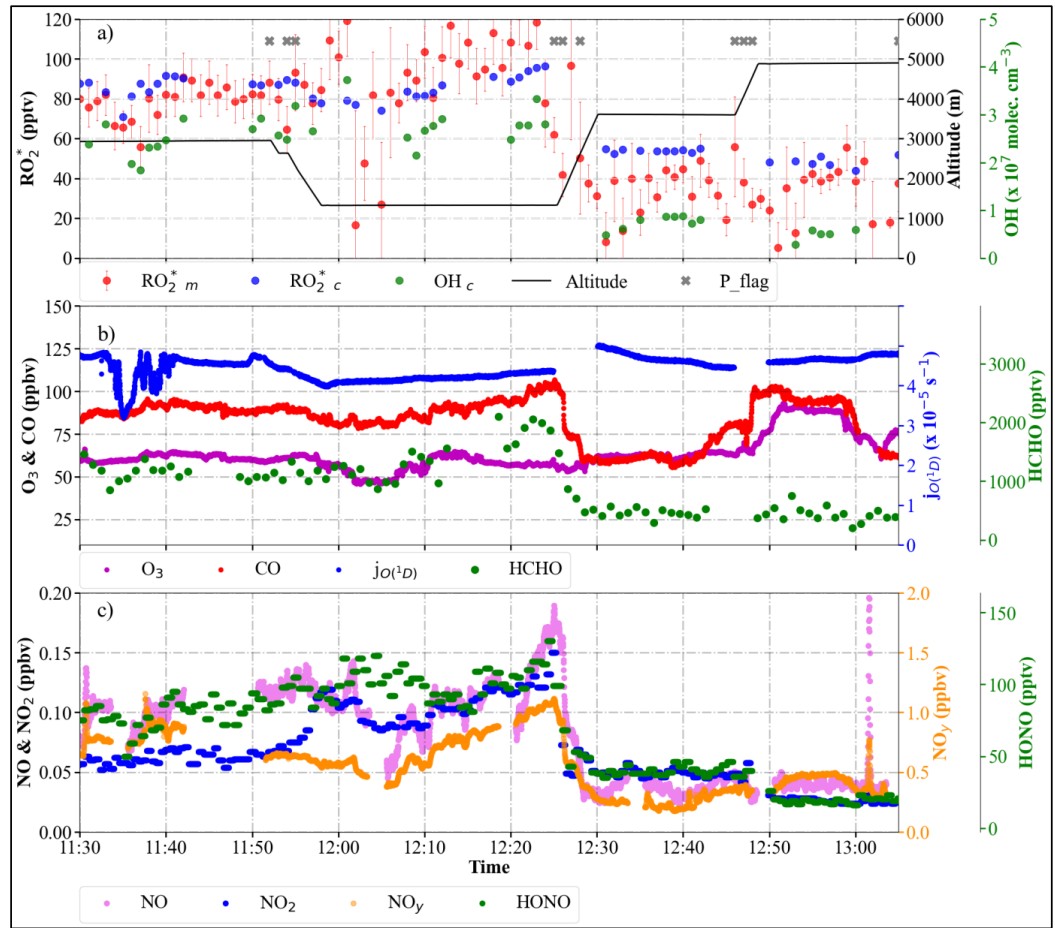

Figure 16: Temporal variation of $RO_2{}^*{}_m$ and $RO_2{}^*{}_c$, selected radical precursors and $j_{O(^1D)}$ along the E-EU-03 flight track: a) $RO_2{}^*{}_m$, $RO_2{}^*{}_c$ and calculated OH ($OH_c$) mixing ratios. The flight altitude is indicated in black. The P_flag indicates $RO_2{}^*$ measurements affected by dynamical pressure variation in the inlet; b) $O_3$, CO, HCHO mixing ratios, and $j_{O(^1D)}$; c) NO, $NO_2$, $NO_y$, and HONO mixing ratios.

The OH concentrations in Fig. 16 are upper limits calculated by assuming pseudo-steady-state for the OH production (R1- R3, R24 and R26) and loss (R4 to R6, R13 and R15 to R19) reactions as described by Eq. 10:

$$2j_1[O_3]\beta + j_3[HONO] + k_{23}[HO_2][NO] + k_{24}[HO_2][O_3] = [OH](k_5[O_3] + k_6[CO] + k_7[CH_4] + k_{12a}[HCHO] + k_{12b}[CH_3CHO] + k_{12c}[CH_3C(O)CH_3] + k_{12d}[CH_3OH] + k_{12e}[CHOCHO] + k_{17}[HO_2] + k_{19}[NO] + k_{20}[NO_2] + k_{21}[HONO]) - 2(k_{18a} + k_{18b})[OH]^2$$

(Eq. 10)

The OH calculated from Eq. 10 assuming δ = 0.5 is much higher than the OH concentration reported in the previous airborne (Crawfor et al., 1999'; Tan et al. 2001) and ground-based measurements (Mihelcic et al., 2003; Kanaya et al., 2007, 2012; Hofzumahaus et al., 2009; Elshorbany et al., 2012; Lu et al., 2012, 2013; Tan et al., 2017, 2018; Whalley et al., 2018, 2021; Michelle et al., 2020) in different urban environments. This indicated that the limited number of OVOCs measurements available for the EMeRGe data set is insufficient to calculate the OH reactivity. The overestimation of OH agrees with the underestimation of $RO_2{}^*{}_m$ in airmasses with a high amount of OVOCs (∑VOCs > 7 ppbv ) as the missing OH – OVOCs reactions in Eq. 8 should reduce ρ (the OH loss during the OH – $RO_2{}^*$ interconversion) and thereby increase the $RO_2{}^*{}_c$. Due to the direct reaction of OH





with most of the gases emitted in the atmosphere, OH budget calculations in airmasses of complex chemistry are challenging and
require the experimental determination of the OH reactivity, as described by Tan et al. 2019 and Whalley et al., 2021.

**4.4. Comparison of results with other studies**

**4.4.1 $RO_2^*$ production rate**

Cantrell et al. (2003b) proposed $P_{RO_2^*}$ to be equal to the sum of two terms representing $RO_2^* - RO_2^*$ reactions and the $RO_2^* -$
$NO_x$ reactions in the troposphere. As a result of this assumption, the relationship between $HO_2$, $RO_2$, $P_{RO_2^*}$ and $NO_x$ is described
by Eq.11:
$$P_{RO_2^*} = k_{RR} [HO_2 + RO_2]^2 + k_{RN} [HO_2 + RO_2] [NO_x] \qquad \text{(Eq. 11)}$$

where $k_{RR}$ and $k_{RN}$ refer to effective rate coefficients for $RO_2^* - RO_2^*$ and $RO_2^* - NO_x$ reactions, and are calculated as fit
parameters. Solving Eq. 11 for $[HO_2 + RO_2]^2$ leads to:
$$[HO_2 + RO_2] = \sqrt[2]{A + B^2} - B \qquad \text{(Eq. 12)}$$

where $A = \frac{P_{RO_2^*}}{k_{RR}}$ and $B = \frac{k_{RN}[NO_x]}{2 \, k_{RR}}$. For low $NO_x$ and/or high $P_{RO_2^*}$, B becomes negligible compared to A. Then $[HO_2 + RO_2]$
approaches $\sqrt[2]{A}$ and is independent of $NO_x$. For high $NO_x$ and /or low $P_{RO_2^*}$, $[HO_2 + RO_2]$ approaches zero.
The least-square fitting in Eq. 12 is applied to $RO_2^*{}_m$ and $RO_2^*{}_c$ from the EMeRGe measurements in Europe binned in 0.1 pptv s$^-$
$^1$ $P_{RO_2^*}$ intervals as shown in Fig. 17. The fit parameters for Fig. 17a and Fig. 18b are $k_{RR} = 7 \times 10^{-5}$; $k_{RN} = 9 \times 10^{-6}$. The $RO_2^*$
calculated by Eq. 9 appears to be close to the linear function of the $NO_x$ measured. Similar to the results of the study of Cantrell
et al. (2003b), a decrease of $RO_2^*$ with $NO_x$ is identified for $NO_x > 1000$ pptv, although only for $P_{RO_2^*} < 0.7$ pptv s$^{-1}$. In the study
of Cantrell et al. (2003b), $P_{RO_2^*}$ only reached values up to 0.275 pptv s$^{-1}$.
Despite the low agreement of the fitted lines with the $RO_2^*{}_m$, a decrease of the $RO_2^*{}_m$ as a function of $NO_x$ is still observed. The
disagreement between the $RO_2^*{}_m$ and the curves estimated using Eq. 12 implies that the simplified Eq. 11 is insufficient to
adequately describe the chemical and physical processes occurring in the troposphere. Part of the disagreement might arise from
missing terms in the $P_{RO_2^*}$ calculation or inaccuracies related to the NO to $NO_2$ ratio in the air mass, which are more evident at
higher $P_{RO_2^*}$. As expected, the ratio of calculated $[RO_2^*{}_c]$ to $\sqrt[2]{P_{RO_2^*}}$ has a negative linear dependency on the measured $[NO_x]$ (see
Fig. 17c). The comparable plot of $[RO_2^*{}_m]$ to $\sqrt[2]{P_{RO_2^*}}$ is not linear. This indicates that more complex non-linear processes are
involved in the air masses investigated than those considered in Eq. 11 (see Fig.17d).



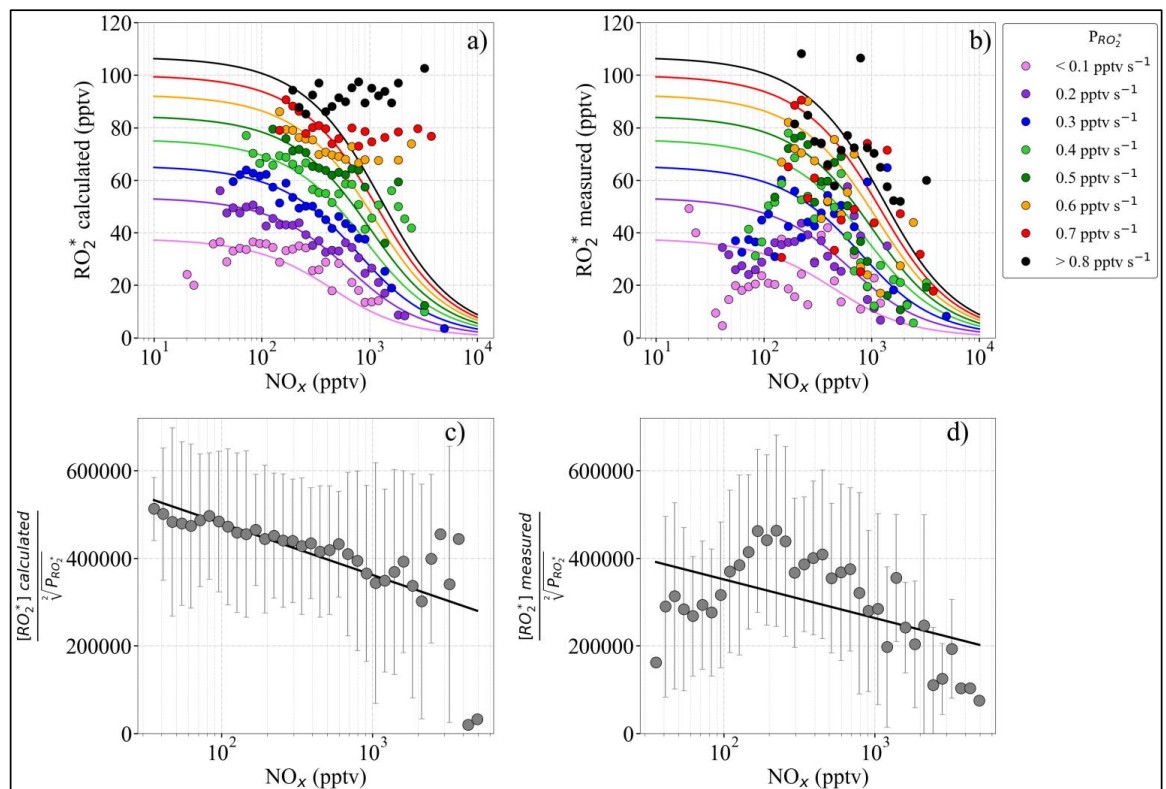


Figure 17: a) $RO_2{^*}_c$ binned with $P_{RO_2^*}$; b) $RO_2{^*}_m$ binned with $P_{RO_2^*}$; c) $[RO_2^*]_c$ to $\sqrt[2]{P_{RO_2^*}}$ ratio; and d) $[RO_2^*]_m$ to $\sqrt[2]{P_{RO_2^*}}$ ratio. The data in a) and b) are coloured with radical production rates. The $RO_2{^*}$ for different $P_{RO_2^*}$ ranges are binned into 50 bins over $NO_x$ in the logarithmic scale from 10 to 10000 pptv. The solid lines are the least square fits obtained using Eq. 11. Error bars indicate $\pm 1\sigma$ of the bins.

### 4.4.2 O$_3$ production rate

The O$_3$ production rate ($P_{O_3}$) is calculated from the EMeRGe Europe dataset using the reaction of $RO_2{^*}$ with NO in a similar manner to that used in previous studies of photochemical processes in urban environments (e.g. Kleinman et al., 1995; Volz-Thomas et al., 2003; Mihelcic et al., 2003; Cantrell et al., 2003b; and references herein).

$$P_{O_3} = k_{RO_2^* + NO}[RO_2^*][NO] \qquad \text{(Eq. 13)}$$

where $k_{RO_2^*+NO}$ is taken as the average of $k_{HO_2+NO}$ and $k_{CH_3O_2+NO}$.

Figure 18 shows plots of the mean $P_{O_3}$ calculated using Eq. 13 from the $RO_2{^*}_m$ and $RO_2{^*}_c$ as a function of NO. The measurements are binned into 50 NO mixing ratio bins. The bin size increases with NO to keep the points equidistant in the logarithmic scale. The calculated $P_{O_3}$ using the $RO_2{^*}_m$ and $RO_2{^*}_c$ agree well within the standard deviation of the bins.

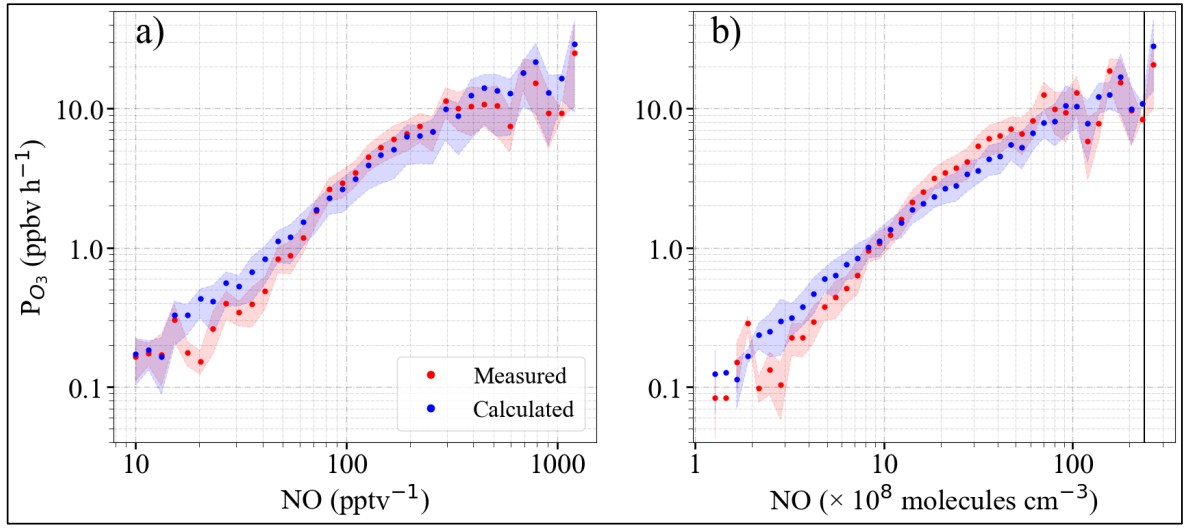

541

Figure 18: Calculated $O_3$ production ($P_{O_3}$) determined using $RO_2{}^*{}_m$ (red dots) and $RO_2{}^*{}_c$ (blue dots) as a function of: a) NO mixing ratio; b) NO number concentration. The 1-minute measurements are binned into 50 bins over NO in the logarithmic scale from 10 to 10000 pptv and from $5 \times 10^7$ to $3.5 \times 10^{10}$ molecules cm$^{-3}$ in 15a and 15b, respectively. The shaded area shows $\pm$ 1$\sigma$ standard deviation of each bin. For comparison with ground-based measurements, the number concentration corresponding to 1 ppbv NO at 1000 mbar and 25°C has been marked by the solid black line in plot b).

Similar $P_{O_3}$ values have been reported for ground-based measurements in polluted areas such as Wangdu (Tan et al., 2017) and Beijing (Whalley et al., 2021) and similar ranges of peroxy radicals and NO mixing ratios. In previous work, Whalley et al. (2018) calculated $P_{O_3}$ to be about an order of magnitude lower than that found in this study from observations in central London for about an order of magnitude lower amount of $HO_2 + RO_2$. For NO > 1 ppbv, the $P_{O_3}$ estimated from the measurement of $HO_2$ and $RO_2$, or from the assumptions of an $HO_2$ to $RO_2$ ratio were underestimated by the models in other studies in the urban atmosphere (e.g. Martinez et al., 2003; Ren et al., 2003; Kanaya et al., 2008; Mao et al., 2010; Kanaya et al., 2012; Ren et al., 2013; Brune et al., 2016; Griffith et al., 2016). This is generally attributed to underestimating large $RO_2$ concentrations, which likely undergo multiple bimolecular reactions with NO before forming an $HO_2$ radical.

During the EMeRGe IOP in Europe, the NO mixing ratios were < 1 ppbv (approximately < $3 \times 10^{10}$ molecules cm$^{-3}$). The ozone production rates obtained for both measured and calculated $RO_2{}^*$ are in reasonable agreement with other modelling studies in urban environments where the mixing ratio of NO is < 1 ppbv.

## 5.  Summary and conclusions

This study exploits the airborne measurements of various atmospheric constituents on-board the HALO research aircraft over Europe in summer 2017 to investigate radical photochemistry in the probed airmasses. $RO_2{}^*$ are calculated by assuming a **p**hotostationary **s**teady-**s**tate (PSS) of $RO_2{}^*$ and compared with the actual measurements. The calculation is constrained by the simultaneous airborne measurements of radical precursors, photolysis frequencies and reactants of $RO_2{}^*$ such as $NO_x$ and $O_3$. The significance and the importance of selected production and loss processes in the $RO_2{}^*$ chemistry are investigated by gradually increasing the complexity of the analytical expression. The agreement of the calculations with the measurements over a wide range of chemical composition and insolation conditions improves when the analytical expression is extended to account for oxy–peroxy radical interconversion reactions and loss of OH and RO during the interconversion. The $RO_2{}^*$ measured is



usually overestimated when NO is < 50 pptv in the air probed. This is attributed to $RO_2^*$ loss processes involving reactions with OH, which are not considered in the analytical expression. The reactions are excluded from the analytical expression to constrain it with on-board measurements. These reactions become significant $RO_2^*$ loss processes at low NO concentrations.

The results indicate that the steady-state calculations mostly underestimated the $RO_2^*$ measurements in polluted plumes of urban origin at altitudes below 2000 m. Changes in the $HO_2$ to $RO_2$ ratios in different plumes partly account for the disagreement in particular cases. In pollution plumes with the sum of the OVOCs measured mixing ratios > 7 ppbv, the underestimation of the measurements can reach up to 80 %. In these plumes, the oxidation and/or photolysis of non-measured VOCs and the ozonolysis of alkenes might be significant, limiting the accuracy of the analytical expression. The overestimation of the OH concentration calculated based on the measured reactants also indicates missing oxy–peroxy radical interconversion reactions in the analytical PSS expression. More information about peroxy radical speciation and VOC partitioning is required to better describe the fast photochemistry in these pollution plumes.

The analytical expression developed is robust enough to simulate the radical chemistry in most of the conditions in the free troposphere encountered during EMeRGe IOP in Europe. Speciated radical and VOC measurements in future campaigns would facilitate the estimation of radical loss reactions in air masses with NO < 50 pptv and improve radical production rates estimations in pollution plumes with a high amount of VOCs, where non-linear complex chemistry is involved. Comparing $RO_2^*$ measurements with $RO_2^*$ calculations from the analytical expression helps to identify different chemical and physical regimes, which can be used to constrain future model studies.

The calculated $O_3$ production rates for NO < 1 ppbv are in the same order of magnitude as those previously reported for urban environments. This indicates that the selected $RO_2^*$ production and loss processes and observations of the radical precursors on-board are, to a good approximation, adequate for the estimation of the $O_3$ production in the measured airmasses in the free troposphere over Europe.

*Competing interests.* The authors declare that they have no conflict of interest.

*Financial support.* The study was funded in part by the German Research Foundation (Deutsche Forschungsgemeinschaft; DFG) HALO-SPP 1294, the University and the State of Bremen, IPA, DLR, Oberpfaffenhofen, Germany. The contributions from BS, FK, and KP were supported via the DFG grants PF 384/16, PF 384/17 and PF 384/19. KB was granted funding via the DFG grant Pl 193/21-1 and acknowledges additional financial from the Heidelberg Graduate School for Physics. EF was supported via the DFG grant NE 2150/1-1 and acknowledges additional financial support from the Karlsruhe Institute of Technology. MG, YL, MDAH and JPB acknowledge financial support from the University of Bremen.

**Acknowledgements:**

MG, MDAH, YL and JPB thank Wilke Thomssen for support during the preparation and integration phases of EMeRGe.

**Author contribution:**

MG, VN, and YL undertook the $RO_2^*$ measurements, flying as key scientists on-board HALO. VN led the deployment of PeRCEAS in the HALO aircraft. MG led the analysis of the PeCEAS measurements and prepared the manuscript with contributions from all co-authors. MDAH and JPB initiated the EMeRGe research project and consortium, acted as co-principal and principal investigators, and participated in the measurement campaigns. They developed the overarching EMeRGe scientific objectives and the required measurement portfolio, directed the EMeRGe research campaigns, and participated in the data



analysis presented. AZ, BB, BS, EF, FO, FK, HS, HZ, KB, KP, and TH have contributed by providing their measurements made
on-board HALO during the campaign and participated in the discussion of results.
**Competing interests**:
The authors declare that they have no conflict of interest.

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
