# Peer review of "On the understanding of tropospheric fast photochemistry: airborne observations of peroxy radicals during the EMeRGe-Europe campaign"

_Atmospheric Chemistry and Physics, 2022_

## Author Comment (AC1)

**Comment on acp-2022-119**

Anonymous Referee #2

*This manuscript presents some rare airborne RO2 measurements over much of Europe. These are useful measurements that should be shared, especially the vertical dependence of concentrations, but the analysis is not focused enough and requires significant revisions.*

*Major comments:*

- *line 101 (in the intro) shows the photostationary state (PSS) equation for [HO2 + RO2], and describes previous studies that compared this value to measured HO2 + RO2. The introduction ends with "Consequently, this data set provides an excellent opportunity to gain a deeper insight into the source and sink reactions of RO2\* and the applicability of the PSS assumption for the different pollution regimes and related weather conditions in the free troposphere". I was looking forward to seeing what insights the authors had to provide regarding the applicability of the PSS assumption…. but it does not show up at all later in the main text! The authors make use of the equation P(RO2\*) = L(RO2\*) quite a bit, but not the above NO-NO2-O3-RO2 photostationary state assumption.*

The NO-NO$_2$-O$_3$-HO$_2$+RO$_2$ PSS radical calculation in Eq.1 assumes that NO$_2$ is in a steady state and has been shown in the past to be very sensitive to the accuracy of the NO$_2$ to NO ratio. This ratio calculated from the NO (in-situ) and NO$_2$ (miniDOAS remote) measurements during the EMeRGe campaign is considered to have a sufficiently large error for not to be a valuable approach to calculate the [HO$_2$+RO$_2$]

In the present paper, we assume that the peroxy radicals are in PSS, i.e., the production and losses of radicals are considered to a good approximation to be equal. During the campaign, a large amount of the trace species involved in the radical formation and loss mechanisms were measured. In this work an analytical expression for the PSS calculation of the total sum of peroxy radicals which takes into consideration only measured species is used. The efficiency of this analytical expression as predicting tool is investigated by comparing the calculated RO$_2$* PSS with the RO$_2$* measurements on-board.

A sentence has been included on Line 111 to emphasise this aspect.

"The PSS radical calculation made on the assumption of the NO$_2$ steady state is very sensitive to the accuracy of the NO$_2$ to NO ratio and the O$_3$ measurements."

- *Line 119: in the abstract RO2\* is defined as RO2 + HO2. On line 119, it's defined as RO2+ HO2 + OH. Quantitatively there's little difference since [OH] is much smaller than the other two terms, but conceptually this is very important. Line 24 states "RO2\* is primarily produced following the photolysis of ozone (O3), formaldehyde (HCHO), glyoxal(CHOCHO), and nitrous acid (HONO) in the airmasses investigated", which is true for (RO2+HO2+OH) but not for (RO2 + HO2). Please be consistent in terminology.*

**Answer:**

The definition of RO$_2$* has been checked for consistency. RO$_2$* is mentioned now in the abstract as:

"RO$_2$* is to a good approximation the sum of peroxy radicals reacting with NO to produce NO$_2$

And in the main text on Line 119 (Line 127 in the revised manuscript) is defined more precisely as:

"The available on-board measurements of RO$_2$* are defined as the total sum of OH, RO and peroxy radicals (i.e., RO$_2$* = OH + $\sum$RO + HO$_2$ + $\sum$RO$_2$, where RO$_2$ are the organic peroxy radicals producing NO$_2$ in their reaction with NO). As the amount of OH and RO is much smaller, RO$_2$* to a good approximation is the sum of HO$_2$ and those RO$_2$ radicals that react with NO to produce NO$_2$)"

- *later in abstract: "The dominant removal processes of RO2\* in the airmasses measured up to 2000 m are the loss of OH and RO through the reaction with NOx during the radical*

*interconversion". This is very confusing – if a reaction is a radical interconversion reaction, then no radicals are lost. Moreover, reactions of RO with NOx are rare and not discussed at all later in the manuscript.*

**Answer:**

This has not been indeed explained adequately. The abstract and the rest of the manuscript have been checked to remove sentences leading to confusion.

In the revised manuscript the sentence mentioned by the referee has been re-written as:

"The dominant terminating processes for $RO_2^*$ in the pollution plumes measured up to 2000 m are the formation of nitrites and nitrates from radical reactions with $NO_x$."

- *It appears that one of the main challenges the authors are facing when interpreting their dataset is that their instrument is much less sensitive to CH3O2 than HO2, and so in the limit that all RO2 are CH3O2, then RO2\* = HO2 + 0.65 × RO2. This relates to the high NO mixing ratios used in their instrument, as commented on by both reviewers for George et al. 2020. This limits how much can be gleaned from the comparison of calculated RO2\*with measured RO2\*, since a priori the CH3O2/RO2\* ratio is not known.*

**Answer:**

The ratio $CH_3O_2/HO_2$ is definitely a challenge for the interpretation of PeRCEAS data in complex environments. The PeRCEAS instrument is calibrated in the laboratory for a 1:1 $HO_2$ to $CH_3O_2$ mixture at the NO mixing ratios used for the measurement on the field. The results are interpreted by assuming a ratio of 1:1 and investigating the impact of systematic deviations from this ratio in the composition of the air masses observed. Thus, in line 483 in original manuscript (see below) the impact of having ratios of $HO_2/RO_2$ different than 1:1 in the disagreement between $RO_2^*$ measured and calculated has been estimated for measurements during the E-EU-03 flight. A 3:1 and a 1:3 ratio of $HO_2:RO_2$, would lead to either a 10% decrease or a 10% increase in the $RO_2^*_m$ retrieved and this uncertainty is well below the in-flight uncertainty of the PeRCEAS instrument.

Line 507 in the revised manuscript:

"Taking $CH_3O_2$ as a surrogate for all $RO_2$ might lead to uncertainties in the $RO_2^*$ calculations in the presence of OVOCs with larger organic chains. On the experimental side, changes in the $HO_2$ to $RO_2$ ratio affect the accuracy of the PeRCEAS retrieval of the total sum of radicals. As noted in section 3, in this study $RO_2^* = HO_2 + 0.65 \times RO_2$, and the eCL is determined for a 1:1 mixture of $HO_2:CH_3O_2$, i.e. $\delta = 0.5$ is used for the $RO_2^*$ retrieval. However, the $HO_2$ to $CH_3O_2$ ratio is not expected to remain constant in all the air masses probed. For a 3:1 ratio of $HO_2:RO_2$, the $RO_2^*_m$ would decrease by 10 %. Similarly, a $HO_2:RO_2$ ratio of 1:3 would lead to an increase of 10 % in the reported $RO_2^*_m$. This uncertainty is well below the in-flight uncertainty of the PeRCEAS instrument indicated by the error bars in Fig. 14 (George et al., 2020), and cannot account for the overall 20 % underestimations. However, it might reduce the differences observed between $RO_2^*_m$ and $RO_2^*_c$ in particular cases. A complete explanation of the variability of $RO_2^*$ in the pollution plumes measured within the IOP in Europe is beyond the scope of this analysis and requires an investigation by high-resolution chemical models."

- *line 231: there is a major error in Equation 2. The penultimate term – the sum of OH +VOC reactions – should not be part of this equation as it reflects radical cycling rather than a primary radical source.*

**Answer:**

The radical propagating reactions in Eq. 2 have been removed.

- *Figures 8-12 and accompanying text: These show variations of the same plot (RO2 measured vs RO2 calculated). I would recommend keeping just 2 of these and moving the rest to the SI as not much is revealed by the 3rd – 5th set of plots. This would help shorten the paper and improve the readability.*

**Answer:**

The text has been shortened and the figures 11 and 14 in the original manuscript have been moved to the supplementary information.

- *Line 40 – remove comma*

**Answer:**

done

- *Line 90 – should say "R23 and R25 are two of the most…"*

**Answer:**

done

- *Section 3: the description of perceas is confusing. Nowhere does it even mention that the sampled air is mixed with NO and CO in amplification mode – the basics of PERCA operation. I do recognize that the instrument has been described in the referenced papers, but just a few more details would be helpful.*

**Answer:**

The description of PeRCEAS has been extended in Line 161 to account for the criticisms of the referee:

"Each channel has a separate chemical reactor and detector, which operate alternatively in both background and amplification modes to account for the rapid background variations during airborne measurements. In both modes NO is continuously added to the air sampled at the reactor, while CO is only added in the amplification mode to initiate the chain conversion of $RO_2^*$ into $NO_2$. In the amplification mode, the sum of the $NO_2$ produced from ambient $RO_2^*$ through the chain reaction, the ambient $NO_2$, the $NO_2$ produced from the ambient $O_3 - NO$ reagent gas reaction and the $NO_2$ produced in the inlet from any other sources (e.g. thermal decomposition of PAN) is measured."

- *line 174: the relative sensitivity to CH3O2 vs. HO2 (a) – was this just based on their previous study, or was it experimentally determined again in between flights? Similarly, please provide more information on calibrations – how many were done? Were the eCL values stable (and their dependence on humidity) or were different values used for each flight? That's great that glyoxal and methyl glyoxal, in addition to the other OVOCs, were measured.*

**Answer:**

The relative sensitivity to $CH_3O_2$ vs. $HO_2$ has been determined for the conditions applied during the measurements with a series of laboratory measurements in a previous study. The eCL calibrations are not possible once the instrument is installed in the aircraft and are therefore made before and after the campaigns. There was no significant difference (within errors) in sensitivity or dependence on humidity) observed in those experiments.

This information has been included in the revised text:

"The $HO_2$ and $RO_2$ detection sensitivity depends on the reagent gas NO concentration due to the rate coefficient of reaction R22 being larger than that for R19. The average eCL for a 1:1 $HO_2$ to $CH_3O_2$ mixture under the DUALER conditions during the campaign in Europe was determined to be $50 \pm 8$ from laboratory calibrations, where the error is the standard deviation estimated from the reproducibility of the experimental determinations. Likewise, the ratio $\alpha = eCL_{CH_3O_2}/eCL_{HO_2}$ was determined to be 65% for the measurement conditions (George et

al., 2020). The values obtained from calibrations before and after the campaign agreed within the experimental uncertainties."

- *Line 202 "Typically, the highest RO2\* mixing ratios were observed below 3000 m over Southern Europe. This is attributed to the higher insolation and temperatures favouring the rapid production of RO2\* from the photochemical oxidations of CO and VOCs" I question the inclusion of temperature in that sentence. If the authors are simply presenting a \*correlation\* between highest RO2 mixing ratios and temperature that is fine, but to \*attribute\* the high mixing ratios to elevated temperature requires some discussion. Are they inferring that the reaction rate constants are faster at higher temperatures? This is certainly not true for all of the reactions. Or are they referring to the increased emissions of biogenic VOC emissions at higher temperatures, leading to higher bVOC concentrations? This by itself won't necessarily lead to higher RO2\* mixing ratios.*

**Answer:**

We agree with the referee in the speculative character of this sentence which referred to a potential combination of higher insolation and temperatures favouring not only higher emissions of bVOC but also stagnant conditions with high photochemical processing which could affect the $RO_2$* levels. Following also the comment of reviewer 1, the sentence has been removed.

- *For HONO photolysis, do the calculated numbers reflect the gross OH formation from HONO photolysis or the net amount (subtracting out the reverse reaction OH + NO)?*

**Answer:**

The calculations in this work are made using the Eq. 6 which considers the net amount of radical formation from HONO photolysis through R3 and R19. In the revised version of the manuscript a text has been included to explain in detail the terms of the Eq. 6 in the revised manuscript:

Line 359:

"$(2j_1[O_3]\beta + j_3[HONO])(1-\rho) + 2j_8[HCHO] + 2j_9[CH_3CHO] + 2(j_{10a} + j_{10b})[CH_3C(O)CH_3] + 2j_{11}[CHOCHO] = \delta[RO_2^*](k_{23}[NO] + k_{24}[O_3])\rho + 2k_{15}\delta(1-\delta)[RO_2^*]^2 + 2k_{16a}((1-\delta)[RO_2^*])^2 + 2k_{14}(\delta[RO_2^*])^2$        (Eq. 6)

where $\beta$ is the effective yield of OH in the reaction of $O(^1D)$ with $H_2O$ given by:

$\beta = \left(\frac{k_{2a}[H_2O]}{k_{2a}[H_2O] + k_{2b}[O_2] + k_{2c}[N_2]}\right)$,

On the left hand side of Eq. 6, $1-\rho$ accounts for the effective yield of $HO_2+RO_2$ through the radical initiation reactions R2a and R3 and reactions R5 to R7 and R12. As the calculation is constrained with on-board measurements, only the reactions of measured VOCs were considered in R12. Similarly, $\rho$ accounts for the effective yield of HONO, $HNO_3$ and $H_2O$ formation through reactions R19 to R21 and the $HO_2 + NO$ and $HO_2 + O_3$ reactions (R23 and R24 respectively) on the right hand side of Eq. 6.

Consequently, $\rho$ is given by:

$$\rho = \frac{(k_{19}[NO] + k_{20}[NO_2] + k_{21}[HONO])}{(k_5[O_3] + k_6[CO] + k_7[CH_4] + k_{12a}[HCHO] + k_{12b}[CH_3CHO] + k_{12c}[CH_3C(O)CH_3] + k_{12d}[CH_3OH] + k_{12e}[CHOCHO] + k_{17}[HO_2] + k_{19}[NO] + k_{20}[NO_2] + k_{21}[HONO])}$$

"

- *Line 244 and 304: "calculated", not "estimated"*

**Answer:**

done

- *Figure 4: I would have found it more useful to see a plot of altitude vs. P(RO2*) rather than [RO2*] colored by P(RO2*).*

**Answer:**

The vertical profile and the altitude latitude distribution of $P(RO_2^*)$ are now included as proposed by the referee, as Fig. S2 and S3 in the supplementary information.

Line 301: "(for the vertical profile and the latitudinal distribution of $P_{RO_2^*}$ see Fig. S2 and S3 in the supplementary information).

[Figure]

Figure S2: Composite average vertical profiles of $P_{RO_2^*}$. The measurements are binned over 500 m altitude. The error bars are the ± 1σ standard deviation of each bin. Median values (red triangles) and the number of individual measurements, n, for each bin (in green) are additionally plotted.

[Figure]

Figure S3: $P_{RO_2^*}$ as a function of latitude and altitude for the EMeRGe measurements in Europe.

The composition and photochemical activity of the air masses sampled during the investigated 7 flights change significantly during the tracks and the production of radicals is expected to present a complex pattern with the altitude. The idea of the figure 4 is to identify relations between the $P(RO_2^*)$ calculated and the $RO_2^*$ measured, under the assumption that the terms included in the $P(RO_2^*)$ are sufficient. In addition, Figure 4 helps to identify more and less frequent $P(RO_2^*)$ regimes, as a basis for the analysis in Figures 5 and 6.

- *line 266- needs some re-wording. "…the high amount of H2O in the air masses probed results in the O3 photolysis and subsequent reaction of O1D with H2O (R1-R2a) and is the highest RO2\* radical production rate". The H2O itself does not cause O3 photolysis…rather the high H2O leads to the reaction O(1D) + H2O being the most important RO2\*source.*

**Answer:**

The text has been accordingly reworded:

"Typically, the high amount of $H_2O$ in the air masses probed leads to the reaction of $O^1D$ with $H_2O$ (R1-R2a) being the highest $RO_2^*$ radical production rate ($\geq 50$ %) below 4000 m"

- *line 305: The sentence "The [RO2\*] < 0.5…." is awkward, change to something like "Measurements in which [RO2\*] were less than xyz…"*

**Answer:**

The sentence has been accordingly re-worded as:

"Measurements in which $[RO_2^*]$ were less than $0.5 \times 10^{12}$ molecules $cm^{-3}$ , $\sqrt[2]{P_{RO_2^*}}$ less than 1000 and with $j_{O(^1D)}$ $> 5 \times 10^{-5}$ were made above 6000 m, where the amount of $RO_2^*$ precursors is low."

- *Figure 7 and text: it's good that someone has done this analysis! I think the correlations observed in figure 8 are about as good as could be expected, though it's interesting that the impact of NOx is not so clear.*

**Answer:**

Thanks. The correlations have indeed a significant spread and the impact of NOx is difficult to be identified. However, a closer analysis reveals that for similar NOx concentrations and $J_{O1D}$ in North

and South of 47°N, the concentrations of the radicals measured are much higher in the South, indicating the differences in VOC sources and that the radical yields coming from the oxidation of VOC are probably responsible for the increase in the amount of the radicals measured.

- *line 338: "The second solution gives…" I don't see any solutions…. please clarify.*

**Answer:**

This sentence refers to the solutions of Eq. 6 which can be solved as a quadratic equation respect to $[RO_2^*]$. One solution is negative and is therefore assumed not to have any physical meaning. The $[RO_2^*]$ calculated are taken from the second solution of the quadratic equation:

$$[RO_2^*] = \frac{-\left(-L_{RO_2^*}\right) - \sqrt[2]{L_{RO_2^*}^2 - 4\left(-2k_{RO_2^*}\right)P_{RO_2^*}}}{2\left(-2k_{RO_2^*}\right)}$$

In Line 406, the text has been modified for clarification:

"Since Eq. 6 is quadratic in $[RO_2^*]$ it can be solved for $[RO_2^*]_c$ where c stands for calculated, as:

$$[RO_2^*]_c = \frac{-\left(-L_{RO_2^*}\right) - \sqrt[2]{L_{RO_2^*}^2 - 4\left(-2k_{RO_2^*}\right)P_{RO_2^*}}}{2\left(-2k_{RO_2^*}\right)} \qquad \text{(Eq. 7)"}$$

- *line 340 – "the measured RO2* (RO2*m) mixing ratio" is confusing – what is RO2*(RO2*m)? Is this a product of RO2* and RO2*m? Or should it just be RO2*m?*

**Answer:**

$RO_2^*{}_m$ stands for $RO_2^*$ measured and $RO_2^*{}_c$ for $RO_2^*$ calculated. $RO_2^*{}_c$ is defined in Line 406 and $RO_2^*{}_m$ in Line 416 of the revised text. From this line, only the abbreviations $RO_2^*{}_m$ and $RO_2^*{}_c$ are used in the text.

- *line 341: "RO2*m and RO2*c are the measured and calculated RO2* respectively for $\delta$ =1, i.e. RO2* = HO2 and d = 0.5, i.e. HO2 = RO2." Confusing. RO2*m is measured RO2, and RO2*c is calculated, but for which case – $\delta$ = 1 or 0.5? This section should be prefaced with some text along the lines of "because not all peroxy radicals are detected equally by the instrument, the comparison of measured and calculated RO2* values is complicated. To investigate this, we …"*

**Answer:**

The introductory sentence in Line 324 (now Line 352 in the revised text) has been extended for clarifying this aspect:

"As stated in section 3, $HO_2$ and $RO_2$ are not speciated but retrieved as $RO_2^*$ by the PeRCEAS instrument. Because not all peroxy radicals are detected equally by the instrument, the comparison of measured and calculated $RO_2^*$ values is complicated. To investigate this, changes in the $HO_2$ to the total $RO_2^*$ ratios, have been taken into consideration by $\delta$, i.e., $[HO_2] = \delta[RO_2^*]$ and $[CH_3O_2] = (1-\delta)[RO_2^*]$ in the analysis. As a first approach, $RO_2$ is assumed to consist only of $CH_3O_2$ to reduce the complexity of the calculations by considering only $CH_3O_2$ reaction rate constants. Moreover, in a previous study the ratio $\alpha = eCL_{CH_3O_2}/eCL_{HO_2}$ was determined to be 65% for the measurement conditions (George et al., 2020)."

- *eq 11: the first term on the right hand side of the equation refers to the RO2* loss reactions HO2 + HO2, RO2 + RO2, and HO2 + RO2. The 2nd term should represent the RO2*loss reactions RO2 + NO2 and HO2 + NO2 and OH + NO2, but not RO2 + NO or HO2 + NO as those are radical interconversion reactions. I recommend simply writing out the full equation as it is confusing to always deal with "HO2 + RO2" and "NOx" in these rate equations.*

**Answer:**

Eq. 11 (now Eq 9 in the revised manuscript) refers to the analysis made by Cantrell et al. (2003b) and the terms of their analysis are just transcribed. The first term on the right hand refers to radical-radical reactions and the second term to $RO_2^*$-NOx reactions where $RO_2^*$ is considered to be the sum of $HO_2$+ $RO_2$ and $K_{RR}$ and $K_{RN}$ are effective rate coefficients, whose value is retrieved from the fitting of the curves obtained from real data. Except for the production rates there is no calculation made. As explained in the text, the production rates calculated by Eq.2 in the manuscript and the EMeRGe measurements in Europe are used to obtain the Figure 16. The text of the Sect. 4.4.1 has been partly reworded for clarification.

- *line 524: dependence, not dependency*

**Answer:**

corrected

---

## Author Comment (AC2)

*The manuscript presents airborne RO2\* observations from the EMeRGe-Europe campaign which was designed to study the chemistry in the outflows from major population centres. The concentration and variability of RO2\* with altitude, latitude and inside and outside of the urban plumes is of interest to the community. The authors compare the observations to a couple of steady state calculations for RO2\*. Some details on the % breakdown of the primary sources of RO2\* are provided, but there is little discussion on the main sinks for RO2\*, which I think should be added to the manuscript. There are some major problems with the manuscript currently: The steady state calculations used are flawed; see my major comments below. Many of the figures are extremely 'busy', and I believe some of the axes have been labelled incorrectly, and so it becomes very difficult to follow the discussion related to these plots.*
*It is difficult for a reader to draw any solid conclusions on the observations and comparison to calculated RO2\* because, not only are there unknowns relating to VOCs present, which impact the calculated RO2\*. The VOCs present also affect the ambient HO2:RO2 ratio, so the absolute sensitivity of the instrument becomes uncertain. I do think that the results from this study should be published, but major revisions to the analyses performed are needed before publication.*

*Major comments*

- *Equation 2: The sum of OH+VOC reactions do not constitute a primary source of radicals and should be removed from this equation*

**Answer:**

The Eq. 2 has been corrected and the rest of the text has been modified to draw a clear line between the radical primary production and initiating and terminating processes.

- *Equation 3 & 5: The authors need to be clear that the photolysis rates for HCHO and CHOCHO account only for the radical forming channels.*

**Answer:**

The text introducing the Eq. 3 and Eq.5 has been modified to prevent misunderstanding.

- *Figure 4, 5 and 6: I don't see the value in binning p(RO2\*) as a function of production rate. I think total production rate as a function of altitude and observed [RO2\*] would be easier to visualise and take information from. I suggest p(RO2\*) is broken down into %precursor contribution in figure 4 (so figure 5 wouldn't be needed). I also find figure 6 extremely difficult to read. From this figure it is impossible to see the HONO concentration profile for example. I suggest showing the altitude profiles of the key RO2\* precursor species as shown in figure 2 (there is no need to break these profiles down into p(RO2\*)production rate. In figure 6, I would focus on HONO, OVOC and O3 altitude profiles as H2O (v) and j(O1D) profiles are provided in figure 2.*

**Answer:**

By binning as a function of production rates it is possible to distinguish between different regimes of the production rates. This enables in a further step a separate analysis of the extreme and most common conditions during the EMeRGe campaign in Europe. The Fig. 6 has been modified for improving readability of the HONO profiles.

- *Section 4.3: This section is difficult to follow and take away any clear conclusions. I don't think anything is gained from gradually increasing the analytical expression. I suggest just beginning with the most comprehensive expression and discussing the components of the expression that have the biggest impact on [RO2\*c]. Although additional terms have been added to equation 8*

*and 9, a number of these terms represent propagation of one radical species to another and so should not be considered. The P(ROx) and D(ROx) expressions given in Tan et al., (2019) and Whalley et al., (2021) with the additional photolytic sources from acetaldehyde, acetone and glyoxal available from EMERGE would seem to me like the most robust expression to use.*

**Answer:**

In the revised version of the manuscript the changes in the analytical expression used for the analysis have been minimised. The analytical expression Eq. 5 is extended only once to Eq. 6. This enables to analyse first qualitatively the relationship between the $RO_2^*$ measured and the square of the $P_{RO2*}$ in the two latitude windows of the campaign. In Eq. 6 $RO_2^*$ the radical yields from VOC oxidations and formation of nitrates and nitrites are included (by introducing 1-$\rho$), and the $HO_2$ to $RO_2^*$ ratio ($\delta$).

- *Section 4.3: This section begins with a series of correlation plots of RO2\* measured vs the square root of the primary production of RO2\*. I wondered if these figures could be put into context, by drawing on previous research by Ehhalt and Rohrer (JGR-Atmos., 105,3565–3571, 2000) and Vaughan et al. (ACP, 2149-2172, 2012)? These papers demonstrate the linear dependence of OH on p(OH) (or jO1D) and square root dependence of HO2 on p(OH) (or jO1D) using the following expression: [OH] or [HO2] or[RO2\*] = (a x JO1Db + c). Where 'a' represents the influence of all chemical sources and sinks, 'b' accounts for the effect of combining all photolytic processes that produce OH,HO2 or RO2 into a single power function of J(O1D) and 'c' is the contribution from all light independent processes. This sort of analyses may be revealing in highlighting differences between different regions that could, for example, be related to differences in VOCs. It may highlight times when, light-independent processes, such as ozonolysis reactions are of significance.*

**Answer:**

The parametrisation of $[HO_2] = a \times jO(1D)^b + c$ has mostly been applied to conditions with low NO (< 20 pptv), in which the $HO_2$ + NO reaction can be taken as negligible in the calculation of the $HO_2$ rate except in Kanaya et al., 2001. The low NO condition is not applicable to EMeRGe measurements in Europe since NO > 50 pptv in 75 % of the air masses probed. In addition, the power relation between $HO_2$ and $jO(^1D)$ requires constant CO and $\tau_{OH}$, for a constant NO value or range (Vaughan et al. 2012). The EMeRGe data set does not satisfy this condition as the air masses sampled have different composition and origin. Furthermore, as mentioned in Kanaya et al (2001) the steady state equation cannot be simplified for $RO_2^*$ to a form such as the Eq. 7 in Vaughan et al. 2012 as the chemistry is more complex when $HO_2$ and $RO_2$ are taken into consideration together.

References:

Penkett, S. A., P.S. Monks, L. J. Carpenter, K. C. Clemitshaw, G. P. Ayers, R. W. Gillett, I. E. Galbally, and C. P. Meyer, Relationship between ozone photolysis rates and peroxy radical concentrations in clean marine air over the Southern Ocean, J. Geophys. Res.,102, 12,805-12,817, 1997.

Carpenter L. J, P. S. Monks, B. J. Bandy S. A. Penkett, I. E. Galbally, and C. P. Meyer, A study of peroxy radicals and ozone photochemistry coastal sites in the Northern and Southern Henri sphere J. Geophys. Res.,102,2 5,417-25,4271, 997.

Kanaya, Y., Sadanaga, Y, Nakamura, K. and Akimoto, H., Behaviour of OH and HO2 radicals during the observations at a remote island of Okinawa (ORION99) field campaign. 1. Observation using a laser induced fluorescence instrument, JGR, Vol 106, NoD20, pp 24,197-24,208, 2001.

- *Section 4.3: It would be useful to show the breakdown of the termination pathways of RO2\* as a function of altitude in this section; similar to the primary production pathways presented earlier.*

**Answer:**

The plot has been introduced as proposed by the RC3 as Fig.9 in Line 392 of the revised manuscript:

"Figure 9 shows the fractional contribution of the destruction rate ($D_{RO_2^*}$) calculated for a 1:1 mixture of $HO_2$ and $CH_3O_2$ using the reactions included in Eq. 6 as a function of altitude. The data are classified into three groups according to the rate of destruction of $RO_2^*$ mixing ratio $D_{RO_2^*} < 0.01$ pptv s$^{-1}$ (a), $0.01 < D_{RO_2^*} < 0.9$ pptv s$^{-1}$ (b), and $D_{RO_2^*} > 0.9$ pptv s$^{-1}$ (c) to show the lowest, most common, and highest ranges, respectively, encountered during the EMeRGe campaign. For 90 % of the measurements, $0.01 < D_{RO_2^*} < 0.9$ pptv s$^{-1}$ applies, while the rest of the data are equally distributed in the other two $D_{RO_2^*}$ ranges. The data in each group are always binned over 500 m when available.

As can be seen in Fig. 9, the ±1σ standard deviation of the obtained bins is very high. In spite of this, the $HO_2$ – $CH_3O_2$ and $HO_2$ – $HO_2$ reactions seem to dominate the radical destruction processes in the air masses probed. Their combined contribution is > 70 % in all the cases except in the 1000 m bin of $D_{RO_2^*} > 0.9$ pptv s$^{-1}$. Other significant radical losses occur through the HONO and $HNO_3$ formation. The contribution of the $CH_3O_2 + CH_3O_2$ reaction to the total $RO_2^*$ destruction rate is < 5 %."

[Figure]

Figure 9: $RO_2^*$ destruction rate $D_{RO_2^*}$ and fractional contributions from loss reactions in Eq.6 as a function of altitude, for: a) $D_{RO_2^*} < 0.01$ pptv s$^{-1}$, b) $0.01$ pptv s$^{-1} < D_{RO_2^*} < 0.9$ pptv s$^{-1}$, and c) $D_{RO_2^*} > 0.9$ pptv s$^{-1}$. Note the different scales in the number of measurements."

- *Figures 10, 11, & 12: The colour coding for the different parameters considered, other than for the sum of VOCs (fig. 12c), don't show a clear trend and I suggest the majority of these figures are moved to the SI, so as not to detract focus from the main discussion.*

**Answer:**

New figures have been introduced In the revised manuscript in order to answer different issues raised by the referees, and the Figures 11 and 14 in the original manuscript have been moved to the supplementary information.

- *Looking at fig. 12c, the calculated RO2\* over-predicts the measured RO2\* under high VOC loading. This, however, directly contradicts what is written on line 420, where the authors state 'RO2\*m is systematically underestimated for ΣVOCs greater than 7 ppbv' and, therefore, the subsequent discussion surrounding missing VOCs in the calculation becomes moot. Unless this is a plotting error (axes labelled incorrectly), this mistake means that the conclusions drawn on ln 573, 574 are wrong. Figure 14a, which looks at the ratio of RO2\*m/RO2\*c, does seem to suggest that the axes have been labelled incorrectly in the earlier figures, but the authors need to confirm this is the case and correct these figures.*

**Answer:**

The axes of the plots mentioned are labelled correctly. The text was misleading and has been modified for clarification (Line 453):

"$RO_2{}^*{}_m$ is both underestimated and overestimated for $\sum$VOCs mixing ratios greater than 7 ppbv. The composition of these air masses is very different, as reflected by the $\Sigma$VOCs/NO ratios. This implies that Eq. 7 does not capture the peroxy radical yields adequately from the measured VOCs and OVOC in these cases. The differences between $RO_2{}^*{}_m$ and $RO_2{}^*{}_c$ may be explained in part by a) changes in OH yields due to additional VOC oxidation processes, which are not in Eq. 7 and/or b) $RO_2{}^*$ production from the photolysis of carbonyls, which were not measured and/or c) $RO_2{}^*$ production from the ozonolysis of alkenes or unidentified biogenic terpene emissions and/or d) overestimation of the loss processes. For VOC < 2ppb and $\sum$VOCs/NO < 20, $RO_2{}^*{}_m$ is systematically overestimated, what might indicate underestimation in the radical losses through nitrite and nitrate formation. "

- *Ln 375 – 381: What were the range of concentrations of methylglyoxal during EMERGE? It seems a little surprising to me that, including the production of RO2\* from methylglyoxal photolysis, leads to RO2\*c systematically overestimating the measured RO2\*m. Assuming RO2\*m is underestimated by RO2\*c in regions of high VOC loading (see comment above on fig 12c), it would be useful to gauge how much additional p(RO2\*) is needed in the calculation to bring RO2\*c into agreement with the observations.*

**Answer:**

The figures below show the measured mixing ratios of methylglyoxal\*, i.e. , $CH_3C(O)C(O)H^*$ in the air masses probed during the EMeRGe flights. As explained in the text, the $CH_3C(O)C(O)H^*$ measured is the sum of methylglyoxal, and a fraction of other substituted dicarbonyls (mainly 2,3-butanedione), with similar visible absorption spectra. For the calculation, methylglyoxal was assumed to be half of the measured $CH_3C(O)C(O)H^*$ as recommended by Zarzana et al. (2017) and Kluge et al. (2020) from other experiments. However, this fraction might change substantially in the different air masses measured, depending on the amount of the dicarbonlys present. Therefore, the calculation of the concentration of methylglyoxal is subject to a high uncertainty and was not considered in the final analysis.

The figures are colour coded in two ranges of mixing ratios in order to visualize the highest and the most common mixing ratios observed.

[Figure]

[Figure]

The highest value of methyl glyoxal* was around 15 ppbv (plot left), and the highest value of methyl glyoxal was therefore calculated to be 7.5 ppbv. The plot on the right indicates that methyl glyoxal* was ≤ 25 ppbv in 80 % of the measurements. Model calculations beyond the scope of this work would be required to investigate the effect of methylglyoxal accounting for the high level of uncertainty in the calculation of the methyl glyoxal mixing ratios from the observations of methylglyoxal*.

*Specific comments:*

- *Abstract: 'measurements of the sum of hydroperoxyl (HO2) and organic peroxy (RO2)radicals that react with NO to produce NO2, i.e. RO2 *' to measurements of the sum of hydroperoxyl (HO2) and organic peroxy (RO2) radicals (i.e. RO2 *) that react with NO to produce NO2'*

**Answer:**

Actually, PeRCEAS and the PERCA technique only measures those $RO_2$ that react with NO to produce $NO_2$ while $HO_2$ always react with NO to produce $NO_2$.

- *Ln 103: 'kNO+(HO2+RO2) is the weighted average rate coefficient assumed for the reactions of peroxy radicals with NO' the authors should state the rate that has been used.*

**Answer:**

The text has been extended for clarification (Line 107):

".. and $k_{NO+(HO_2+RO_2)}$ is usually estimated for the most abundant $HO_2$ and $CH_3O_2$ by assuming a 1:1 $HO_2$ to $CH_3O_2$ ratio and averaging the $k_{NO+HO2}$ ($8.2{\times}10^{-12}$ $cm^3$ molecules$^{-1}$ s$^{-1}$ at 298K and 1 atm.) and $k_{NO+CH3O2}$ ($7.7{\times}10^{-12}$ $cm^3$ molecules$^{-1}$ s$^{-1}$ at 298K and 1 atm.) rate coefficients for the reaction with NO."

- *Ln 202: From figure 3, j(O1D) increases with altitude, so the highest RO2* concentrations (below 3000m) cannot be attributed to higher insolation alone. Rather the net j(O1D)*[H2O] leads to the greatest primary production of OH below 3000 m.*

**Answer:**

This sentence intended pointing out in a qualitative manner the latitudinal differences in $RO_2^*$concentrations below 3000 m. Following the comments of other referees, the sentence has been changed and moved to the discussion to prevent confusion.

---

## Author Comment (AC3)

**Comment on acp-2022-119**

Anonymous Referee #4

*This paper presents aircraft measurements of peroxy radicals during the EMeRGe-Europe campaign. The authors compare their measurements to predictions from several iterations of a photostationary state analysis. The authors find that the predicted peroxy radical concentrations were lower than the measured concentrations and suggest that photolysis and oxidation of OVOCs not included in the steady-state expression were responsible for the discrepancies.*
*While the measurements likely provide important new information, the paper is difficult to read. In addition to an analysis of the ability of the photostationary state expression to reproduce the measured peroxy radical concentrations, the authors also provide an analysis of the rates and sources of radical production and the estimated rate of ozone production. Unfortunately, the main conclusions of the paper are lost in the extended discussion. There are also problems with their chemical mechanism and the form of the steady-state equations that they are using to estimate the peroxy radical concentrations.*
*Overall, this paper presents some interesting and valuable measurements of peroxy radical concentrations. The paper may be suitable for publication after correcting their photostationary state expressions and re-analyzing their results. The paper would also benefit from moving much of this analysis and the discussion of the rates of radical and ozone production to a supplement and focus the main discussion on their primary conclusions as outlined in the abstract and the summary.*

*Major comments*

- *The authors need to correct and clarify their conclusions stated in the abstract and the text regarding loss of RO radicals (lines 120, 358 and 566 for example). I'm surprised that they are considering the RO + NO reaction an important loss mechanism for alkoxy radicals in the troposphere when the traditional understanding of the fate of these reactions in the atmosphere is reaction with O2 or isomerization and/or decomposition.*
  *While the RO + NO termination reaction (reaction 22) may be important in laboratory studies, it is unlikely that this termination reaction for alkoxy radicals larger than methoxy or ethoxy could compete with reaction with O2 or isomerization/decomposition under atmospheric conditions (see Orlando et al., Chem. Rev. 103, 4657-4689, 2003). This would likely become apparent if they had included the rate of isomerization/decomposition of alkoxy radicals in their photostationary state expressions in addition to reaction with NO and O2 in their attempt to calculate the fraction of RO termination vs propagation (equations S12 and others).*

**Answer:**

This is a misunderstanding. We have not explained role of the reaction of RO with NO adequately. In the atmosphere the rate of the reaction of RO with NO does not compete with the rate of reaction of RO with $O_2$. The ratio $k_{22}[RO]NO]$ to $k_{26}[RO][O_2]$ is typically $< 1 \times 10^{-4}$. In the introduction the reaction R22 was included for completeness.

The text has been revised to correct any potential misunderstanding. In line 96 the sentence

"The rate of R22 in the atmosphere compared to that of R26 is negligible"

has been included. In addition, a new figure (Fig. 9) has ben included to show the fractional contribution of loss reactions considered in this study.

- *Instead, termination of peroxy radicals through reactions with NOx leading to the formation organic nitrates such as the RO2 + NO -> RONO2 reaction are likely more important. Unfortunately, it appears that the authors are not including these reactions in their chemical mechanism.*
  *As a result, their steady-state equations that attempt to incorporate the formation of organic nitrates as radical termination reactions are incorrect (equations 8 and 9). The authors should incorporate an average organic nitrate yield from the RO2 + NO reaction instead of incorrectly attempting to account for the formation of RONO relative to reaction with O2 using rate*

*constants for methoxy radical with NO and O2. It is not clear how this correction would impact their calculated peroxy radical concentrations, but their results should be recalculated and reanalyzed in a revision of their manuscript.*

**Answer:**

The terminating reactions of peroxy radicals with $NO_x$ were considered in the manuscript in the former Eq. 8 ( Eq. 6 in the revised paper). In the old Eq. 8 the term :

$$\left( \left( \frac{k_{22}[NO]}{(k_{22}[NO] + k_{26}[O_2])} \right) k_{25}(1 - \delta)[NO] \right)$$

was correctly calculated. However, this term is negligible as the referee pointed out above and therefore it is not included in Eq 6 in the revised manuscript. This does not change the calculated peroxy radical concentrations.

Concerning the formation of nitrates, as stated in the text, $CH_3O_2$ is the only $RO_2$ considered in the analysis:

Line 355: "As a first approach, $RO_2$ is assumed to consist only of $CH_3O_2$ to reduce the complexity of the calculations by considering only $CH_3O_2$ reaction rate constants"

We agree with the referee that larger $RO_2$ react with NO to make $RONO_2$ as is well documented. However, our $RO_2^*$ is the sum of $HO_2$ and those $RO_2$ which react with NO to form $NO_2$. We use $CH_3O_2$ to represent the $RO_2$, as we wrote. According to JPL, the yield of the formation $CH_3ONO_2$ is <0.5% while 99.5% leads to $CH_3O + NO_2$ . Therefore, this formation path has not been further considered in the analysis (see table note (l) in the Table S1 in the supplementary information).

In order to reduce the complexity and the assumptions about the unknown $RO_2$ composition, the formation of organic nitrates for longer $RO_2$ than $CH_3O_2$ ( as e.g. investigated by Atkinson and Carter and Butkovskaya et al, see references below) has not been considered in the calculations. These studies show that generally the pressure dependent yield of alkylnitrates for the reaction of alkylperoxy radicals with NO increases from ca. 4% for C3 to ca. 30% for C>8. For the case that the mixing ratios of long alkylperoxyradicals is higher than the $CH_3O_2$ mixing ratio, the $RO_2^*$ calculated in this study will overestimate the ambient $RO_2^*$.

[Figure]

**Figure 4.** Comparison of the branching ratios $k_b/k_a$ of alkyl nitrate formation in $RO_2$ + NO reactions for small alkylperoxy radicals: ▽, ref 14; ■, this work; ●, ref 18; ▲, ref 17; ◆, ref 16; atmospheric pressure data for propyl nitrate (○) and butyl nitrate (□) are from ref 10, and those for n-pentyl (△) nitrate are from ref 43. The open circle at 100 Torr is the branching ratio for isopropyl nitrate from ref 44.

(Butkovskaya et al., 2015)

Atkinson, R.; Carter, W. P. L.; Winer, A. M. Effects of Temperature and Pressure on Alkyl Nitrate Yields in the NOx Photooxidations of n-Pentane and n-Heptane. J. Phys. Chem., 87,2012-2018, 1983.

Carter, W. P. L.; Atkinson, R. Alkyl Nitrate Formation from the Atmospheric Photooxidation of Alkanes; A Revised Estimation Method., J. Atmos. Chem., 8, 165-173, 1989

Butkovskaya, N.; Kukui, A.; Le Bras, G. Pressure and Temperature Dependence of Ethyl Nitrate Formation in the $C_2H_5O_2$ + NO Reaction. J. Phys. Chem. A 2010, 114, 956-964.

Butkovskaya, N.; Kukui, A.; Le Bras, G. Pressure Dependence of Iso-Propyl Nitrate Formation in the i-$C_3H_7O_2$ + NO Reaction. Z. Phys. Chem. 2010, 224, 1025-1038

Butkovskaya, N. I., Kukui, ⊥ A., Le Bras, G., Rayez, M.-T., and Rayez J.-C., Pressure Dependence of Butyl Nitrate Formation in the Reaction of Butylperoxy Radicals with Nitrogen Oxide, JPCA, 119, 4408-4417,2015.

*Specific comments*

- *The authors seem to confuse radical initiation and termination processes with radical production and loss through propagation in several places in the manuscript. For example, it appears that the authors intended to calculate the rate of OH, HO2, and RO2 radical initiation using equation 2, but the equation incorrectly includes the rate of radical propagation by the OH + VOC reaction. Even though they neglect this term in their analysis, they should remove it from the equation and clarify their use of radical production vs. initiation throughout the manuscript and supplement.*

**Answer:**

This has been identified and corrected all over the text and the supplementary information.

- *In their revision, the authors should consider only including the results of their overall photostationary state calculations (after correction) in the main text and include the incremental analysis in the supplement (Figures 9-11). This would reduce the length of this discussion and the number of similar plots, making the discussion easier to follow.*

**Answer:**

The text has been shortened and modified for clarification following the comments of all referees. Some equations have been simplified or removed and a few figures have been moved to the supplementary information.

- *In addition to the correlation plots shown in Figure 12, it would be useful to include the calculated RO2\* concentrations in the plots of the measured RO2\* concentrations as a function of altitude (Figure 3 and perhaps Figure 4), illustrating the agreement/disagreement as a function of height.*

**Answer:**

The figure 11 in the revised manuscript has been included as proposed by the referee.

[Figure]

Figure 11: Vertical distribution of the mean $RO_2{}^*_m$ and mean $RO_2{}^*_c$ using Eq. 7 for $\delta = 0.5$ for the EMeRGe data set in Europe. The measurements are binned over 500 m altitude. The error bars are the $\pm 1\sigma$ standard deviation of each bin. Median values (red and green triangles) the interquartile 25-75% range (red and blue shaded areas) and the number of individual measurements, n, for each bin (in green) are additionally plotted.

The vertical profiles comprise measurements from different flights on different days and different geographical areas over Europe, which are taken under very heterogeneous conditions and only have in common the altitude. Please also note that the number of measurements decrease with the altitude due to the own nature of the EMeRGe campaign.

In order to analyse the agreement/disagreement between $RO_2{}^*_m$ and $RO_2{}^*_c$ with the altitude in more detail, the Fig. S4 showing the mixing ratios as a function of latitude and altitude has additionally included in the supplementary information

[Figure]

Figure S4: Latitudinal and altitudinal distribution of a) $RO_2^{*}{}_{m}$ and b) $RO_2^{*}{}_{c}$ mixing ratios calculated using Eq. 7 for $\delta = 0.5$.

Further information about the agreement/disagreement between $RO_2^{*}{}_{m}$ and $RO_2^{*}{}_{c}$ can be seen in the former Fig. 11 (in the revised manuscript moved to the supplementary information as Fig. S5) where $RO_2^{*}{}_{m}$ is plotted versus $RO_2^{*}{}_{c}$ and the data points are coloured-coded for altitude:

[Figure]

Figure S5: $RO_2^{*}{}_{m}$ versus $RO_2^{*}{}_{c}$ calculated using Eq. 7 for $\delta = 0.5$. The data points are colour-coded for a) photolysis frequency of $O_3$; b) altitude. The 1-minute (small circles), the mean of the binned $RO_2^{*}{}_{m}$ over 10 pptv $RO_2^{*}{}_{c}$ intervals (large circles), and the median of each bin (grey triangles) are shown. The error bars indicate the standard error of each bin. The linear regression for the binned values (solid line) and the 1:1 relation (dashed line) are also depicted for reference

as well as in the figure 14 by the $RO_2^{*}{}_{m}/RO_2^{*}{}_{c}$ ratio as a function of altitude and latitude for all the EMeRGe flights over Europe.

Generally, there is no obvious relation only with the flight altitude except for the measurements above around 4000 m, which are all overestimated, as states in the text:

The text has been modified as follows (from Line 429 in the revised manuscript):

"Figure 11 shows the vertical profiles of $RO_2^{*}{}_{m}$ and $RO_2^{*}{}_{c}$ mixing ratios calculated for $\delta = 0.5$, averaged for the EMeRGe flights over Europe in 500 m altitude bins. $RO_2^{*}{}_{c}$ seems to overestimate $RO_2^{*}{}_{m}$ for altitudes above 4000 m. As mentioned in Sect. 4.1, the vertical profiles are a composite from averaging flights with legs carried out at different longitude and latitudes. Therefore, the differences between $RO_2^{*}{}_{m}$ and $RO_2^{*}{}_{c}$ have been studied in more detail respect to the composition of the individual air masses (see the $RO_2^{*}{}_{m}$ and $RO_2^{*}{}_{c}$ mixing ratios as a function of latitude and altitude in Fig. S4 in the supplementary information).

Figure 12 shows the data for δ = 0.5 colour-coded with NO, NO$_x$, the sum of HCHO, CH$_3$CHO, CHOCHO, CH$_3$OH, and CH$_3$C(O)CH$_3$ (from now on referred to as ΣVOCs), as a surrogate for the amount of OVOCs acting as RO$_2^*$ precursors, and the ΣVOCs to NO ratio. The largest differences between RO$_2^*{}_m$ and RO$_2^*{}_c$ are observed for the bins around 50 pptv. The RO$_2^*{}_c$ overestimate the RO$_2^*{}_m$ mostly for RO$_2^*{}_m$ < 25 pptv observed above ≈4000 m. These air masses are characterised by NO < 50 pptv, ΣVOCs typically below 4 ppbv, high ΣVOCs/NO ratios ( > 50), and low insolation conditions , i.e. j$_{O(^1D)}$ < 2 × 10$^{-5}$ s$^{-1}$ (see Fig. S5 in the supplementary information). Under these insolation conditions, the radical production rate is expected to be low, and the RO$_2^*$ – RO$_2^*$ reactions are expected to dominate the RO$_2^*$ loss processes. As OH and H$_2$O$_2$ were not measured during the EMeRGe campaign in Europe, Eq. 7 does not include the loss reactions R17 and R18, which might be significant under such conditions (Tan et al., 2001) and explain the overestimation of RO$_2^*{}_m$."

- *The data shown in Figure 4 is not consistent with their reported binning as there appears to be a point below 500 m even though there are no reported measurements at this altitude.*

**Answer:**

The point indicated by the referee below 500 m was measured in a stable flight leg just after the take-off of one of the flights (as indicated by the number of points) and has been removed in the revised figure.

[Figure]

Figure 4: a) Composite averaged vertical distribution of measured RO$_2^*$ colour-coded according to the value of P$_{RO_2^*}$, b) the number of measurements in each altitude bin. Small circles are 1-minute individual measurements binned with P$_{RO_2^*}$ values in 0.1 pptv s$^{-1}$ intervals. Larger circles result from a further binning over 500 m altitude steps. All the production rates below 0.1 pptv s$^{-1}$ and above 0.8 pptv s$^{-1}$ are binned to 0.1 pptv s$^{-1}$ and 0.8 pptv s$^{-1}$, respectively. The error bars are the standard deviation for each altitude bin.

---

## Author Comment (AC4)

**Comment on acp-2022-119**
Anonymous Referee #1

*The authors report and discuss peroxy radical measurements performed during flights with the aircraft HALO across Europe. Because there are only few flight measurements of radicals over Europe, these measurements are valuable. However, it is not very clear, what the improvement in the understanding of tropospheric fast photochemistry really is from the manuscript. The author mainly compare measurements with different approaches of steady state calculations. Results are mainly descriptive, but there is little discussion about the meaning for the understanding of photochemistry. The presentation quality needs to be improved. It is partly unclear, how equations for steady state calculations are derived and what the meaning is. This manuscript needs major improvements to be suitable for publication in ACP.*

- *Abstract: The definition of RO2* is unclear. In the first sentence it sounds as if this is the sum of RO2+HO2, but later it looks as if also OH is included. Please clarify and be precise and accurate with definitions.*

**Answer:**

The meaning of $RO_2^*$ is now clarified. The $RO_2^*$ is mentioned in the abstract in Line 19 of the revised manuscript:

"The measurements of $RO_2^*$ on HALO were made using the in-situ instrument **P**eroxy **R**adical **C**hemical **E**nhancement and **A**bsorption **S**pectrometer (PeRCEAS). ). $RO_2^*$ is to a good approximation the sum of peroxy radicals reacting with NO to produce $NO_2$."

In Line 127 in the main text, the meaning of $RO_2^*$ is explained in more detail:

"The available on-board measurements of $RO_2^*$ are defined as the total sum of OH, RO and peroxy radicals (i.e., $RO_2^* = OH + \sum RO + HO_2 + \sum RO_2$, where $RO_2$ are the organic peroxy radicals producing $NO_2$ in their reaction with NO). As the amount of OH and RO is much smaller, $RO_2^*$ to a good approximation is the sum of $HO_2$ and those $RO_2$ radicals that react with NO to produce $NO_2$. "

- *Abstract L22: How can a production rate agree with a concentration?*

**Answer:**

The sentence in Line 23 has been rewritten to prevent misunderstanding, as follows:

"Radical production rates were estimated using knowledge of the photolysis frequencies and the $RO_2^*$ precursor concentrations measured on-board HALO, as well as the relevant rate coefficients. Generally, high $RO_2^*$ were measured in air masses with high production rates."

- *Abstract L23: RO2 is not directly produced from the photolysis of ozone and HONO, but OH is that then further reacts to produce RO2* species. Please be accurate how you phrase this.*

**Answer:**

The sentence has been rephrased as follows:

"In the airmasses investigated, $RO_2^*$ is primarily produced by the reaction of $O^1D$ with water vapour and the photolysis of nitrous acid (HONO), and of oxygenated volatile organic compounds (OVOC, e.g. formaldehyde (HCHO), and glyoxal (CHOCHO))."

- *Abstract L25: For an abstract the statement about the PSS is vague and not well-defined. Please expand here, which processes are considered in the PSS and what quantity is calculated.*

**Answer:**

The sentence in Line 27 in the revised manuscript has been extended for clarification:

" Due to their short lifetime in most environments, the $RO_2^*$ concentrations are expected to be in a photostationary steady state (PSS) i.e., it is assumed a balance between production and loss rates. The $RO_2^*$ production and loss rates and the suitability of PSS assumptions to estimate the $RO_2^*$ mixing ratios and variability during the airborne observations are discussed. The PSS assumption for $RO_2^*$ is considered robust enough to calculate $RO_2^*$ mixing ratios for most conditions encountered in the air masses measured."

- *Abstract L30: Really RO+NOx ? If RO2\* is the sum of RO2+HO2+OH, it is not clear to me, why this statement is about radical interconversion, because radical interconversion reactions cancel out. Please rephrase and clarify.*

**Answer**

The text has been rephrased for clarification:

"The dominant terminating processes for $RO_2^*$ in the pollution plumes measured up to 2000 m are the formation of nitrites and nitrates from radical reactions with $NO_x$. Above 2000 m, $HO_2 – HO_2$ and $HO_2 – RO_2$ reactions dominate the $RO_2^*$ removal."

- *L90: Reaction R25 should be mentioned as well.*

**Answer:**

The R25 has been included in Line 95 as suggested:

"R23 and R25 are two of the most important reactions in the troposphere as they lead to $O_3$ formation via the reactions R27 and R28."

- *L91: The first half of the sentence is not clear. What do you mean with insolation? Do you mean PSS? This would not be required to ensure rapid photochemical processes. Please rephrase and clarify.*

**Answer:**

The text from Line 99 has been modified for clarification:

"The sum of $HO_2$ and $RO_2$ that react with NO to produce $NO_2$ can be estimated by assuming that the interconversion of NO to $NO_2$ reaches a **p**hotostationary **s**teady-**s**tate (PSS), in which production and loss are to a good approximation equal.

The PSS assumption for $[NO_2]$ in the following mechanism (R23 to R29) leads to Eq. 1

$$HO_2 + NO \rightarrow OH + NO_2 \qquad\qquad\qquad\qquad\qquad\qquad\qquad (R23)$$

$$RO_2 + NO + O_2 \rightarrow R_{H-1}O + NO_2 + HO_2 \qquad\qquad\qquad\qquad (R25 + R26)$$

$$NO_2 + h\nu\ (\lambda < 400\ nm) \rightarrow NO + O \qquad\qquad\qquad\qquad\qquad (R27)$$

$$O + O_2 \overset{M}{\rightarrow} O_3 \qquad\qquad\qquad\qquad\qquad\qquad\qquad\qquad\qquad (R28)$$

$$NO + O_3 \rightarrow NO_2 + O_2 \qquad\qquad\qquad\qquad\qquad\qquad\qquad\qquad (R29)$$

$$[HO_2 + RO_2]_{PSS} = \frac{k_{NO+O_3}}{k_{NO+(HO_2+RO_2)}}\left(\frac{j_{NO_2}[NO_2]}{k_{NO+O_3}[NO]} - [O_3]\right) \qquad (Eq.1)$$

where $j_{NO_2}$ is the photolysis frequency of $NO_2$; $k_{NO+O_3}$ ($1.9\times10^{-14}$ $cm^3$ molecules$^{-1}$ s$^{-1}$ at 298K and 1 atm.) is the rate coefficient of the reaction of NO with $O_3$ and $k_{NO+(HO_2+RO_2)}$ is usually estimated for the most abundant peroxy radicals $HO_2$ and $CH_3O_2$ by assuming a 1:1 $HO_2$ to $CH_3O_2$ ratio and averaging the $k_{NO+HO2}$ ($8.2\times10^{-12}$ $cm^3$

molecules$^{-1}$ s$^{-1}$ at 298K and 1 atm.) and k$_{NO+CH3O2}$ (7.7×10$^{-12}$ cm$^3$ molecules$^{-1}$ s$^{-1}$ at 298K and 1 atm.) rate coefficients for the reaction with NO. As noted by Parrish et al (1986), the PSS assumption requires conditions with sufficient and stable solar irradiation, ensuring NO$_2$ stable photolysis rates (j$_{NO2}$) "

- *L102: Specifically since the manuscript is about airborne measurements, the temperature and if necessary also the pressure should be given, if values for reaction rate constants are mentioned.*

**Answer:**

The text in Line 106 has been accordingly modified:

" where j$_{NO_2}$ is the photolysis frequency of NO$_2$ ; k$_{NO+O_3}$ (1.9×10$^{-14}$ cm$^3$ molecules$^{-1}$ s$^{-1}$ at 298K and 1 atm.) is the rate coefficient of the reaction of NO with O$_3$ (…). "

As stated now in S1 of the supplementary information, the reaction rate constants used for the RO$_2$$^*$ calculations along the flights were calculated for the ambient temperature and pressure measured on-board HALO.

- *L103: The typical reader may not know, what exactly is meant with "weighted average rate coefficient" and why this is required. Please clarify and rephrase.*

**Answer:**

In Eq. 1 the k$_{NO}$(HO$_2$+RO$_2$) is usually estimated for the most abundant HO$_2$ and CH$_3$O$_2$ peroxy radicals by averaging the k$_{NO+HO2}$ (8.2×10$^{-12}$ cm$^3$ molecules$^{-1}$ s$^{-1}$ at 298K and 1 atm.) and k$_{NO+CH3O2}$ (7.7×10$^{-12}$ cm$^3$ molecules$^{-1}$ s$^{-1}$ at 298K and 1 atm.) rate coefficients assuming a 1:1 HO$_2$:CH$_3$O$_2$ ratio. The text in Line 107 has been accordingly modified:

"… and k$_{NO+(HO_2+RO_2)}$ is usually estimated for the most abundant HO$_2$ and CH$_3$O$_2$ by assuming a 1:1 HO$_2$ to CH$_3$O$_2$ ratio and averaging the k$_{NO+HO2}$ (8.2×10$^{-12}$ cm$^3$ molecules$^{-1}$ s$^{-1}$ at 298K and 1 atm.) and k$_{NO+CH3O2}$ (7.7×10$^{-12}$ cm$^3$ molecules$^{-1}$ s$^{-1}$ at 298K and 1 atm.) rate coefficients for the reaction with NO."

- *L126: It is not obvious, why the measurements of trace gases in Reactions R1 to R26 other than required in Equation 1 minimizes the number of assumptions for calculating RO2*. My expectation would have been that this would allow to perform also full model calculations of RO2* concentrations, which could be compare PSS calculations. Please explain in more detail.*

**Answer:**

The PSS radical calculated by Eq. 1 assumes that the NO$_2$ is in a steady state and has been shown in the past to be very sensitive to the accuracy of the NO$_2$ to NO ratio. The NO (in-situ) and NO$_2$ (miniDOAS remote) measurements during the EMeRGe campaign would not enable a suitable calculation of PSS peroxy radicals with Eq. 1.

In the present paper, we assume that the peroxy radicals are in PSS, i.e., the production and losses of radicals are considered to a good approximation to be equal. During the campaign, a large amount of the trace species involved in the radical formation and loss mechanisms were measured. We agree with the referee that this calculation is a good task for a model if available. In this study an analytical expression for the PSS calculation of the total sum of peroxy radicals, which takes into consideration only measured species, is used. The efficiency of this analytical expression as predicting tool is investigated by comparing the calculated RO$_2$* PSS with the RO$_2$* measurements on-board.

The text starting with "In contrast to other experimental deployments" has been modified for clarification:

"In contrast to other experimental deployments, the concentrations and/or mixing ratios of the majority of the key species involved in reactions R1 to R26 were continuously measured on-board HALO during the EMeRGe campaign. This enables the use of a large number of measurements to constrain the PSS calculation of $RO_2^*$. Consequently, this data set provides an excellent opportunity to gain deeper insight into the source and sink reactions of $RO_2^*$ and the applicability of the PSS assumption for the different pollution regimes and related weather conditions in the free troposphere."

- *L135: Please avoid to define and use abbreviations like IOP and MPC and others that are not common. The typical reader will forget them, while reading the manuscript. It only makes it difficult to follow the line of arguments.*

**Answer:**

The use of abbreviations has been kept to a minimum in the revised version, as proposed by the referee. However, a few abbreviations are required for the terms or definition which are often repeated, in order to facilitate the reading.

- *L143: What do you mean with "stable flight layers"?*

**Answer:**

"stable flight layers" refer to the fact that the altitude of the HALO aircraft was kept constant for the time probing at each level of the vertical profiles.

The text has been modified:

"Vertical profiles of trace constituents were typically made by keeping the HALO altitude constant at different flight levels upwind and downwind of the target MPCs."

- *L168: Please add also the pressure, for which you calculated the concentrations.*

**Answer:**

The values have been calculated for 200 mbar which was the PeRCEAS inlet pressure during the EMeRGe campaign in Europe. This information has been added in Line 174.

- *L172: Why do you only refer to CH3O2 as RO2? Earlier you mention "weighted average rate coefficient" implying that you not only have CH3O2.*

**Answer:**

The calibration of PeRCEAS in the laboratory is only made for $HO_2$ and for a 1:1 mixture of $HO_2$ + $CH_3O_2$ as described in George et al. (2020). The text from Line 176 has been extended for clarification and in order to include the details requested by RC2:

"The $HO_2$ and $RO_2$ detection sensitivity depends on the rates of loss of $HO_2$ and $RO_2$ by the R19 and R22 reactions. The latter depends on the concentration of the reagent gas NO added and the reaction rate coefficients, where $k_{22}$ is larger than $k_{19}$. The average eCL for a 1:1 $HO_2$ to $CH_3O_2$ mixture under the DUALER conditions during the campaign in Europe was determined to be $50 \pm 8$ from laboratory calibrations, where the error is the $\pm 1\sigma$ standard deviation estimated from the reproducibility of the experimental determinations. Likewise, the ratio $\alpha = eCL_{CH_3O_2}/eCL_{HO_2}$ was determined to be 65% for the measurement conditions (George et al., 2020). The values obtained from calibrations before and after the campaign agreed within their experimental errors."

- *L180: I would recommend to give a number how large the humidity effect was for measurements in this work.*

**Answer:**

The following text has been included in Line 189 of the revised manuscript, to provide quantitative information about the humidity effect on the eCL of PeRCEAS, as suggested by the referee:

"The $[H_2O]$ in the DUALER inlet was lower than $1 \times 10^{17}$ molecules cm$^{-3}$ for 60 % of measurements during EMeRGe in Europe, for which the eCL$_{wet}$ = 76 % of eCL$_{dry}$. At the highest humidity observed during the campaign, i.e., $[H_2O]_{inlet} = 2 \times 10^{17}$ molecules cm$^{-3}$, the eCL$_{wet}$ is 55 % of eCL$_{dry}$ (see Fig. S1 in the supplementary information)."

In addition, the Fig. S1 showing the ambient $[H_2O]$ versus $[H_2O]$ during the EMeRGe campaign in Europe has been included in the supplement together with an explanatory text:

[Figure]

Figure S1: Ambient $[H_2O]$ versus $[H_2O]$ measured in DUALER during EMeRGe campaign in Europe, colour-coded with altitude.

"Figure S1 shows the humidity measured in the DUALER during the EMeRGe campaign in Europe. As the pressure in the DUALER inlet is lower than the ambient, $[H_2O]$inlet < $[H_2O]$ambient. However, the humidity is still significant and affects the eCL in the DUALER. Therefore, the eCL was corrected using the equation eCL$_{wet}$ = eCL$_{dry}$ $\times$ A$^{([H_2O] \times 10^{-16})}$ obtained from the laboratory characterisation of the eCL water dependence, where A = 0.973 for the NO number concentration added to the DUALER inlet during EMeRGe campaign in Europe (George, 2022, PhD thesis).

The $[H_2O]$ in the inlet was lower than $1 \times 10^{17}$ molecules cm$^{-3}$ for 60 % of measurements during EMeRGe in Europe, for which the eCL$_{wet}$ = 76 % of eCL$_{dry}$. At the highest humidity observed during the campaign, i.e., $[H_2O]_{inlet} = 2 \times 10^{17}$ molecules cm$^{-3}$, the eCL$_{wet}$ is 55 % of eCL$_{dry}$."

- *L183 ff: The short description of miniDOAS data / data evaluation is hard to understand for the non-expert. Please rephrase. It is also not clear at this point, why this instrument is explain in more detail, whereas other instruments more obvious useful to determine the PSS are not explained.*

**Answer:**

Table 1 includes references for detailed descriptions of the instruments on-board HALO used in this study. In addition to adding further important details of PeRCEAS, the objective of the section 3 is on the remote-sensing instruments to understand the shortcomings of combining results of in-situ and remote sensing instruments. With this purpose further details on the conversion of column densities measured by remote sensing instruments into mixing ratios or number concentrations as provided by in-situ instruments are given. To improve clarity the following sentences have been rephrased:

"The remote sensing instruments used on HALO during EMeRGe were the mini Differential Optical Absorption (minDOAS) and the Heidelberg Airborne Imaging DOAS Instrument (HAIDI). The miniDOAS observes the atmosphere using six telescopes: two being optimised for the ultraviolet, two for the visible, and two for the near infrared. Three telescopes observe in nadir viewing and three in limb viewing. The three limb scanning telescopes point to the starboard side perpendicular to the aircraft fuselage axis. They are rotated to compensate for roll relative to the horizon. A variant of the DOAS retrieval technique uses least square fitting of the measured and radiative transfer modelled absorption along the line of sight to retrieve the differential salt column density, dSCD, of the target gas and a scaling reference gas. The latter is the dimer of molecular oxygen $(O_4)$. As the vertical profile of the concentrations of $O_2$ and thus $O_4$ are known then the mixing ratios of the target gas at the flight altitude obtained from the target gas and $O_4$ dSCDs (for more details see Stutz et al., 2017; Hüneke et al., 2017; Kluge et al., 2020; Rotermund et al., 2021). The HAIDI nadir observations are used to retrieve the dSCDs below the aircraft. The dSCDs from HAIDI are then converted to the mixing ratios using knowledge of the aircraft altitude and the corresponding geometric Air Mass Factor (AMF), calculated by a radiative transfer model under a well-mixed $NO_2$ layer assumption. As a result of this assumption, the calculated mixing ratios for HAIDI target gases are lower limits and similar to the actual values observed while flying within and close to a well-mixed boundary layer. In spite of the differences in the sampling volume, temporal sampling and spatial resolution between the in situ and remote sensing techniques, the concentrations of the gas HCHO measured by both techniques were in good agreement and the concentrations of the $NO_2$ (remote sensing) and $NO_y$ (in situ) were consistent (for more details see Schumann, 2020)."

- *L186: Please explain RT modelling.*

**Answer:**

RT stands for radiative transfer and has now so been included in the rephrased paragraph beginning at the Line 192

- *L187: Please explain the abbreviation HAIDI.*

**Answer:**

HAIDI is the abbreviation for the name of the instrument: The Heidelberg Airborne Imaging DOAS Instrument (HAIDI).This has been included in the text.

- *L191: Please explain what you mean with "common and related species".*

**Answer:**

The text has been modified for clarification:

"In spite of the differences in the sampling volume, temporal sampling and spatial resolution between the in situ and remote sensing techniques, the concentrations of the gas HCHO measured by both techniques were in good agreement and the concentrations of the $NO_2$ (remote sensing) and $NO_y$ (in situ) were consistent (for more details see Schumann, 2020)."

- *L202: I would avoid a conclusion about the reason for high RO2 in specific regions before doing the analysis. Your arguments are plausible but there are also other plausible explanations giving the contrary conclusion.*

**Answer:**

The sentence in Line 202 "This is attributed to the higher insolation and temperatures favouring the rapid production of $RO_2^*$ from the photochemical oxidations of CO and VOCs." has been modified and moved to Line 335, as suggested by the referee.

"Photochemical processing is expected to be enhanced over Southern Europe due to the prevailing conditions of high insolation and temperatures during the EMeRGe flights, which might lead to the rapid production of $RO_2^*$ from the photochemical oxidations of CO and VOCs."

- *L210: I do not understand the argument "comparable". What is exactly compared here? Calculating RO2\* from PSS can always been done as long as the time required to reach PSS is short enough that concentrations of species do not significantly change. Please explain and rephrase.*

**Answer:**

The emphasis of this sentence in Line 210 in the original manuscript (*"Provided that insolation conditions and a sufficient number of key participating precursors are comparable, the air mass origin is irrelevant for calculating $RO_2^*$ concentrations and mixing ratios."*) is on the fact that the $RO_2^*$ are dominated by fast photochemistry. The local conditions and chemical composition of an airmass rather than its origin should determine the variability and variations of $RO_2^*$ concentrations. Therefore, the air trajectories are not considered in this analysis.

The text from line 222 has been modified for clarification:

"The origin and thus the composition of the air sampled during the seven flights over Europe were different and heterogeneous. Typically, the air masses measured were influenced by emissions from MPCs and their surroundings, and sometimes by biomass burning transported over short or long distances. The concentration and mixing ratio of $RO_2^*$ rather depends on the insolation and the chemical composition of the air probed, particularly on the abundance of $RO_2^*$ precursors, than on the origin of the air masses. Since $RO_2^*$ are controlled by fast chemical and photochemical processes, the air mass origin and trajectory are not used in the calculation of $RO_2^*$ concentrations and mixing ratios, but are of interest as the source of the $RO_2^*$ precursors. Thus, the $RO_2^*$ variability and its production rates provide valuable insight into the photochemical activity of the air masses probed."

- *Figure 3: Wouldn't make more sense to show percentiles instead of standard deviations to be independent from outliers?*

**Answer:**

The figure 3 has been modified as proposed by the referee.

[Figure]

Figure 3: Composite average vertical profiles of a) $RO_2^*$, b) $j_{O(^1D)}$ and c) $[H_2O]$ observations. The measurements are binned over 500 m altitude. The error bars are the $\pm 1\sigma$ standard deviation of each bin. Median values (red triangles) the interquartile 25-75% range (red-shaded area) and the number of individual measurements, n, for each bin (in green) are additionally plotted.

- *L224: I cannot follow the argument that differences between mean and median values indicate more or less variability. Median and mean values could be exactly the same, if the distribution*

*of values is symmetric independent on how big the range of values is. It is also not obvious, if you want to say that there is a change how similar median and mean values are. I do not see that the similarity depends on the height.*

**Answer:**

In agreement with the last two comments of the referee the Figure 3 has been changed and the sentence in Line 224 has been removed.

- *Line 235: "becomes" instead of "become"*

**Answer:**

This has been corrected

- *L235: Please clarify what you mean with "low NOx conditions" and why this impacts the significance of H2O2 photolysis.*

**Answer:**

The results of Tan et al., (2001) during the PEM-Tropics B campaign show that in the case of low concentrations of $NO_x$, the rate of the reactions $HO_x$ with $NO_x$ decreases. All other conditions being the same, this increases the concentration of $HO_2$ and $RO_2$ and thus the rates of production of the concentrations of the peroxides $H_2O_2$ and $ROOH$. These peroxides photolyse and react with OH to form $HO_x$. At higher mixing ratios of $NO_x$, the reactions of $HO_x$ with $NO_x$ dominate. Tan et al., 2001 investigated clean chemical conditions, which are defined as "low $NO_x$ conditions", with 95% of the measured NO values below 50 pptv and 76% below 20 pptv, and median $O_3$ values below 20 ppbv.

As can be seen in the figure below, in most of the air masses measured during EMeRGe in Europe, NO were higher than 50 pptv, in ~75 % of the airmasses probed. Therefore, the conclusions from Tan et al.(2001) are applicable to the present work.

[Figure]

The text has been modified to clarify this aspect (starting at Line 251 ):

"In this study, Eq. 2 has been applied to the measurements taken within the EMeRGe campaign in Europe. There were no $H_2O_2$ measurements available for EMeRGe. However, the measurements reported by Tan et al. (2001) indicate that the rate of OH production from the $H_2O_2$ photolysis is not significant except when the $NO_x$ is low. To be more precise, for conditions having NO < ~50 ppt, the partitioning of $HO_x$ is strongly shifted to $HO_2$, which then predominantly reacts with itself or $RO_2$ to form peroxides, which can in turn photolyse. For conditions with NO > 50 pptv, the rates of reactions of $HO_2$ or $RO_2$ with $NO_x$ are faster than those of $HO_2$ with $HO_2$ and $RO_2$. As the NO mixing ratio was higher than 50 pptv in 75 % of the air masses probed in Europe, the rate of the photolysis of $H_2O_2$ was, as a first approximation, assumed not to be significant source of OH for the EMeRGe dataset considered in this study."

- *L238: Please define OVOC before using it in Eq 2*

**Answer:**

OVOC has been defined with the Eq. 2 in Line 250 as :

"where OVOC stands for oxygenated volatile organic compounds."

- *L237. This statement needs explanation. Why can you assume that photolysis of OVOCs is more important compared to reaction with OH? This is not obvious. Which were the most important OVOCs and VOCs and can you quantitatively show that your assumption is valid? Can you also show this for ozonolysis reactions? If you want to calculate the RO2\*production rate you may not need to consider OH reaction, because this is a radical conversion reaction and not a primary production, which you may want to calculate. This should be clarified, if you talk about production. Please explain and extend your description.*

**Answer:**

The text has been modified to clearly distinguish between primary radical production and propagation reactions in Eq. 2. Furthermore, it has also been clarified, that the terms without measurements are not considered in the analysis. This is the case of the ozonolysis of alkenes and the photolysis of $H_2O_2$.

"The rate of production of $RO_2^*$from the reactions R1 to R13 is given by:

$$P_{RO2*} = 2jO_D^1[O_3] \frac{k_{O_D^1+H_2O}[H_2O]}{k_{O_D^1+H_2O}[H_2O]+k_{O_D^1+O_2}[O_2]+k_{O_D^1+N_2}[N_2]} + j_{HONO}[HONO] + 2j_{H_2O_2}[H_2O_2] + 2\sum_i j_i[OVOC_i] + {}$$
$$+\sum k_{O_3+alkenes_k}[O_3][alkenes_k] \qquad (Eq.\ 2)$$

where OVOC stands for oxygenated volatile organic compounds.

In this study, Eq. 2 has been applied to the measurements taken within the EMeRGe campaign in Europe. There were no $H_2O_2$ measurements available for EMeRGe. However, the results reported by Tan et al. (2001), indicate that the rate of OH production from the $H_2O_2$ photolysis is not significant except when $NO_x$ is low. To be more precise, for conditions having NO < 50 ppt, the partitioning of $HO_x$ is strongly shifted to $HO_2$. $HO_2$then predominantly reacts with itself or $RO_2$ to form peroxides, which can in turn photolyse. For conditions with NO > 50 pptv the rates of reactions of $HO_x$ with $NO_x$ are faster than those of $HO_2$ with $HO_2$ and $RO_2$. As the NO mixing ratio was higher than 50 pptv in 75 % of the air masses probed in Europe, the rate of the photolysis of $H_2O_2$ was as a first approximation assumed not to be a significant source of OH for the EMeRGe dataset considered in this study.

Formaldehyde (HCHO), acetaldehyde ($CH_3CHO$), acetone ($CH_3C(O)CH_3$), and glyoxal (CHOCHO) were the OVOCs measured in EMeRGe forming directly radicals through photolysis. They are produced in the photolysis and oxidation of VOCs and are likely the most abundant and reactive OVOCs present. In this study they were assumed to be the dominant VOCs in the air masses probed.

There were no measurements of alkenes provided in EMeRGe. Consequently, the ozonolyis term in Eq. 2 was not included in the analysis."

- *L238: How large were the concentrations of these OVOCs? What do you mean concretely, if you take this as "surrogate"? Equation 3 only considers 4 OVOC species, which rather indicates that you neglect others.*

**Answer:**

The present study uses only the measurements on-board as input for the $RO_2^*$ calculations. The 4 OVOCs species mentioned were the only VOCs measured in most of the flight legs. OVOCs are the most abundant and reactive VOCs and are therefore considered to be representative for the sum of VOCs. There were taken as a surrogate because it cannot be ruled out that the air masses contain other non-measured OVOC species. As mentioned in the answer above, the text has been modified to clarify this:

"Formaldehyde (HCHO), acetaldehyde ($CH_3CHO$), acetone ($CH_3C(O)CH_3$), and glyoxal (CHOCHO) were the OVOCs measured in EMeRGe forming directly radicals through photolysis. They are produced in the photolysis and oxidation of VOCs and are likely the most abundant and reactive OVOCs present. In this study they were assumed to be the dominant VOCs in the air masses probed."

- *L244: I assume that measurements allowed a calculation of the air concentration density rather than an estimate.*

**Answer:**

This is true. The word "estimated" has been replaced by "calculated".

- *L245ff: Avoid explaining details of a figure that is explained in the legend and / or caption of the figure.*

**Answer:**

The text has accordingly been modified to avoid redundancy.

- *L291: The section header referring to PSS. From what is written earlier, one would expect calculations using Equation 1, but then you start with calculations using Eq 5. Also later in this Section Eq 5 is stated as PSS calculation instead of Equation 1 and not used at all in the end. This is confusing. Please be consistent. It is not clear, why Equation 1 is introduced earlier at all.*

**Answer:**

As explained above in the answers for Line 91 and Line 126, the PSS radical calculation in Eq. 1 assumes the $NO_2$ steady state and in the present paper, the approach bases on the assumption of PSS for peroxy radicals. As the $NO_2$ steady state has usually been used in the past for the calculation of peroxy radicals, the Eq.1 is written in the introduction as a reference and to emphasise the difference with this work. The text at these positions has already been modified and potential confusion is hopefully by now clarified.

- *L297: It is a bit contradictory to state "interconversion reactions occur without losses", because interconversion implies that the radical nature is not lost.*

**Answer:**

We agree with the referee about this intrinsic redundancy. The initial idea was to emphasise the "competition between reactions". The sentence has been reworded as:

"The R5 to R7, R12, R16b, and R23 to R26 are interconversion reactions between OH, RO, HO$_2$ and RO$_2$ and do consequently occur without radical losses. Solving Eq. 4 leads to Eq. 5 if RO$_2^*$ – RO$_2^*$ reactions are assumed to be the dominant radical terminating processes."

- *L298 ff: Please justify that you can calculate the loss of RO2* -RO2* reaction by an weighted average rate coefficient? What do you use as weights? Without knowing the distribution between HO2 and RO2 it is hard to imagine how this loss rate can be accurately calculated. It is not obvious how this is mathematically done, if you expand the right side of Eq. 5 using [HO2] and [RO2] concentrations. If you assumed e.g. [HO2] = [RO2] = 0.5 [RO2*], this should be clearly said and written down what this means for the equation. The assumption of [HO2]=[RO2] would be expected if the loss of [HO2] and [RO2] is dominated by reaction with NO. Please expand, if this is the case for measurements in your work. In this case, it would be also essential to show and discuss NO measurements and peroxy radical loss rates with NO. What about the loss of RO2*due to the reaction of NO2+OH? Could this have been significantly contributed to the RO2* loss? Your analysis between differences, if you divide data sets between North and South may hint that this loss process was relevant.*

**Answer:**

Figure 7 does not involve any kind of mathematical calculations but rather provides a first qualitative glance on the relationship between the RO$_2^*$ measured and the square root of the production rate of RO$_2^*$ calculated using Eq.3. In spite of the spread around the 1:1 line, the obtained plot indicates a linear relationship. According to Eq. 5, the slope of this linear relationship should represent a total loss rate coefficient (called in the text "an effective RO$_2^*$ self-reaction rate coefficient"), i.e., a kind of weighted rate coefficient including the effect of HO$_2$ – HO$_2$, HO$_2$ – RO$_2$ and RO$_2$ – RO$_2$ reactions. As mentioned by the referee, it is challenging to calculate this rate mathematically without further details about the distribution between HO$_2$ and RO$_2$ and the RO$_2$ speciation. This was also included in the text on Line 306 (now Line 333):

"Apart from this, the spread in the diagram confirms that the effective RO$_2^*$ self-reaction rate $k_{RO_2^*+RO_2^*} [RO_2^*]^2$ varies widely in the air masses probed likely due to the effect of changes in HO$_2$ and $\sum$RO$_2$ concentrations in the individual loss reaction rate coefficients"

From this point in the analysis, the Eq. 5 is modified to include stepwise different mechanisms and assumptions aiming at the calculation of realistic RO$_2^*$. This is the case e.g. of radical reactions with NO$_x$ as mentioned by the referee.

The text has been modified for clarification:

In Line 326: "where $k_{RO_2^*+RO_2^*}$ represents an effective RO$_2^*$ self-reaction rate coefficient comprising HO$_2$ – HO$_2$, HO$_2$ – RO$_2$ and RO$_2$ – RO$_2$ reaction rates."

- *L318: I do not understand the statement about the validity of results. Please explain and rephrase.*

**Answer:**

The sentence "*Note that these results are only valid for the data set acquired over Europe during EMeRGe and do not yield a relationship between [RO$_2^*$] and $\sqrt[2]{P_{RO_2^*}}$, which is generally applicable for these two latitude windows.*" points out that the data used are only a snapshot of the atmosphere during the flights and flight legs and under the particular encountered conditions of EMeRGe. Therefore, the concentration levels and radical production rates calculated in this work for Northern and Southern Europe are only an example but are not meant to be applicable under all conditions to these latitude windows.

The sentence has been revised.

"Please note that these results are only valid for the data set acquired over Europe during EMeRGe flights and do not yield a relationship between $[RO_2^*]$ and $\sqrt[2]{P_{RO_2^*}}$, which is generally applicable under all conditions for these two latitude windows."

- *L330: It would be good, if names of e.g. photolysis frequencies in Equation 5 and 6 were consistent. It should be emphasized that the point of assuming that RO2 consist only of CH3O2 is only, in order to have one RO2 species and therefore not considering differences in RO2+RO2 and RO2+HO2 reaction rate constants. In general, I would recommend to start with Equation 6 and then you easily derive Equation 5. By doing this, you also will be able to explain what you mean with average weighted reaction rate constant in Equation 5.*

**Answer:**

The names of the photolysis rates in Eq. 5 and Eq. 6 are consistent with the list of reactions in the introduction. Since Eq. 5 is the first equation in the text for the balance between radical production and losses in the PSS analysis, the name of the photolysis rates specifies the species being photolysed to emphasise visually the species and processes considered. As the Eq. 6 gets more complicated, the names of the photolysis rates are shortened by numbering them according to the reaction list in the introduction and in the Table 1 in the supplement.

A sentence has been included to explain this in Eq.5:

"where $j_{HCHO}$, $j_{CH_3CHO}$, $j_{CH_3C(O)CH_3}$ $j_{CHOCHO}$ are respectively $j_8$, $j_9$, $j_{10a,b}$ and $j_{11}$, as in Table 1 in the supplementary information"

Starting by the simplified Eq.5 enables first showing the spread in the linear relationship between measured data and photolysis rates before proceeding with the calculation of $RO_2^*$ by taking into account the dominant individual processes involved in the formation and destruction of radicals.

- *L338: It is rather confusing that the negative solution is mentioned at this point, but not when you discuss Eq. 5, where the form of the quadratic equation is identical.*

**Answer:**

The Eq.5 is not used to calculate $[RO_2^*]$ as it is done with Eq.6. With the help of Eq. 5 the relation between $RO_2^*$ measured and the square root of $P_{RO_2^*}$ is investigated. In contrast, Eq. 6 is solved as a quadratic equation to obtain the $[RO_2{}^*{}_c]$, i.e., $RO_2^*$ calculated, that are subsequently compared with the $[RO_2{}^*{}_m]$, i.e., $RO_2^*$ measured.

The text in the revised version has been reworded for clarification.

- *L342: The effect that RO2* measurements can be affected by differences in the detection sensitivity of RO2 and HO2 should have been discussed for the results with delta=0.5 (Equation 5).*

**Answer:**

The text has been modified to improve clarity. The Eq. 8 and 9 in the original manuscript are now Eq.6 and Eq.7 and the information in Section 4 has been improved.

The text has been extended in Line 352 for the explanation of the detection sensitivity of $RO_2$ and $HO_2$:

"As stated in section 3, $HO_2$ and $RO_2$ are not speciated but retrieved as $RO_2^*$ by the PeRCEAS instrument. Because not all peroxy radicals are detected equally by the instrument, the comparison of measured and calculated $RO_2^*$ values is complicated. To investigate this, changes in the $HO_2$ to the total $RO_2^*$ ratios, have been taken into consideration by $\delta$, i.e., $[HO_2] = \delta[RO_2^*]$ and $[CH_3O_2] = (1-\delta)[RO_2^*]$, in the analysis. As a first

approach, $RO_2$ is assumed to consist only of $CH_3O_2$ to reduce the complexity of the calculations by considering only $CH_3O_2$ reaction rate constants. Moreover, in a previous study the ratio $\alpha = eCL_{CH_3O_2}/eCL_{HO_2}$ was determined to be 65% for the measurement conditions (George et al., 2020)."

- *L344: Please make rather quantitative than qualitative statements about the level of agreement. What effect do you expect from differences in reaction rate constants among RO2, if you do not assume that all RO2 is CH3O2?*

**Answer:**

If we do not assume that all $RO_2$ is $CH_3O_2$ in the calculations, we need to consider the impact of:

a) Differences in the rate for the reaction of other $RO_2$ with NO. With the exception of $CH_3CO.O_2$ the $k(RO_2+NO)$ (listed in JPL) range within ≤15% and are <10% of the $k(HO_2+NO)$. Only in the case of abundances in the air mass of $CH_3CO.O_2$ or other $RO_2$ of similar reactivity with NO higher than $CH_3O_2$, the effect of a) would be significant.

b) The branching ratio of the pressure dependent channel of the reaction of $RO_2$ with NO to form alkylnitrates $RONO_2$, R= $CH_3$, $C_2H_5$, etc. that increases as R becomes larger. The studies of Carter and Atkinson, and more recently Butkovskaya et al.(see references below). These studies show that generally the yield of alkylnitrates for the reaction of alkylperoxy radicals with NO increases from ca. 4% for C<3 to ca. 30% for C>8. If we assume that 10-20% of $RO_2$ have R larger than $CH_3$, the effect in the calculation of the pressure dependent channel forming alkyl nitrates is estimated to be within the experimental error of the measurements. For higher amounts, the $RO_2^*$ ambient would be underestimated. However, this quantification is not possible without knowledge of the $RO_2$ composition of the air mass.

c) Differences in the rates of $HO_2+RO_2$ and $RO_2+RO_2$ reactions. The quantification of this effect is also challenging without knowledge of the $RO_2$ composition of the air mass. There is a limited number of reaction coefficients available for these reactions. Modelling studies such as Carter, (2010), use active peroxy radical operators such as MEO2 + HO2 = HCHO + O2 + H2; MEO2 + MEO2 = MEOH + HCHO + O2 and MEO2 + MEO2 = #2 {HCHO + HO2} for their simulation.

A quantitative estimation of the overall effect of a) b) and c) for the calculations is beyond the scope of this study.

Atkinson, R.; Carter, W. P. L.; Winer, A. M. Effects of Temperature and Pressure on Alkyl Nitrate Yields in the NOx Photooxidations of n-Pentane and n-Heptane. J. Phys. Chem., 87,2012-2018, 1983.

Carter, W. P. L.; Atkinson, R. Alkyl Nitrate Formation from the Atmospheric Photooxidation of Alkanes; A Revised Estimation Method., J. Atmos. Chem., 8, 165-173, 1989

Carter, W.P.L., Development of the SAPRC-07 chemical mechanism, Atmos.Environ., 44, 5324-5335, 2010

Butkovskaya, N.; Kukui, A.; Le Bras, G. Pressure and Temperature Dependence of Ethyl Nitrate Formation in the $C_2H_5O_2$ + NO Reaction. J. Phys. Chem. A 2010, 114, 956-964.

Butkovskaya, N.; Kukui, A.; Le Bras, G. Pressure Dependence of Iso-Propyl Nitrate Formation in the i-$C_3H_7O_2$ + NO Reaction. Z. Phys. Chem. 2010, 224, 1025-1038

Butkovskaya, N. I., Kukui, ⊥ A., Le Bras, G., Rayez, M.-T., and Rayez J.-C., Pressure Dependence of Butyl Nitrate Formation in the Reaction of Butylperoxy Radicals with Nitrogen Oxide, JPCA, 119, 4408-4417,2015.

[Figure]

**Figure 4.** Comparison of the branching ratios $k_b/k_a$ of alkyl nitrate formation in RO$_2$ + NO reactions for small alkylperoxy radicals: $\triangledown$, ref 14; $\blacksquare$, this work; $\bullet$, ref 18; $\blacktriangle$, ref 17; $\blacklozenge$, ref 16; atmospheric pressure data for propyl nitrate ($\circ$) and butyl nitrate ($\square$) are from ref 10, and those for $n$-pentyl ($\triangle$) nitrate are from ref 43. The open circle at 100 Torr is the branching ratio for isopropyl nitrate from ref 44.

(from Butkovskaya et al., 2015)

- *L356: It is not clear, which processes you are referring to, if you mention VOC oxidation processes. OH + VOCs would be a radical interconversion process and ozonolysis reactions and Cl chemistry may be not of importance for conditions of the campaign.*

**Answer:**

As mentioned above whole text in section 4 has been shortened and modified for clarification.

- *L358: Again it is confusing, if you talk about radical conversion reactions, but in fact you mean radical termination reactions. Please rephrase and be clear with the definition throughout the manuscript.*

**Answer:**

As for the previous comment, this part of the text has been thoroughly modified to improve clarity and the whole manuscript has been checked for consistency.

- *Equation 8 / 9: Similar it is confusing that you name reaction rate constants referring to radical conversion reactions and move the loss into a loss factor. What is the value of the loss factor? It would be easier, if you added more explanation, which loss reactions (products) you include. I read the first loss term as non-radical products from HO2 + NO and HO2 +O3, it is not clear to me, what for example the product of HO2+O3 would be. The factor rho associated with this term is explained as OH loss during the OH-RO2\*interconversion, which does not fit, what I read from the equation. There is more explanation needed, what is meant with this term. It is also not clear to me, if the second loss term (organic nitrate formation k25 and k22) is correct and why this is connected to RO2+RO2 reactions (k16b). This needs to be explained in more detail. It would be much easier to understand, if you introduced yields of products produced from radical termination reactions.*

**Answer:**

The equations 5 to 9 have been modified to improve clarity and in the revised manuscript the Eq.5 is only modified to Eq. 6. The Eq. 6 is solved as Eq.7 and used for all the calculations. Some more text has been introduced to describe and clarify the terms of Eq. 6 (Line 358 onwards):

"The Eq. 5 is additionally extended to include $RO_2^*$ effective yields from VOC oxidation and radical losses through HONO, $HNO_3$, and organic nitrate formation:

$$(2j_1[O_3]\beta + j_3[HONO])(1 - \rho) + 2j_8[HCHO] + 2j_9[CH_3CHO] + 2(j_{10a} + j_{10b})[CH_3C(O)CH_3] +$$

$$2j_{11}[CHOCHO] = \delta[RO_2^*](k_{23}[NO] + k_{24}[O_3])\rho + 2k_{15}\delta(1 - \delta)[RO_2^*]^2 + 2k_{16a}((1 - \delta)[RO_2^*])^2 +$$

$$2k_{14}(\delta[RO_2^*])^2 \qquad \text{(Eq. 6)}$$

where $\beta$ is the effective yield of OH in the reaction of $O(^1D)$ with $H_2O$ given by:

$$\beta = \left(\frac{k_{2a}[H_2O]}{k_{2a}[H_2O] + k_{2b}[O_2] + k_{2c}[N_2]}\right),$$

On the left hand side of Eq. 6, 1-$\rho$ accounts for the effective yield of $HO_2 + RO_2$ through the radical initiation reactions R2a and R3 and reactions R5 to R7 and R12. As the calculation is constrained with on-board measurements, only the reactions of measured VOCs were considered in R12. Similarly, $\rho$ accounts for the effective yield of HONO, $HNO_3$ and $H_2O$ formation through reactions R19 to R21 and the $HO_2 + NO$ and $HO_2 + O_3$ reactions (R23 and R24 respectively) on the right hand side of Eq. 6.

Consequently, $\rho$ is given by:

$$\rho = \frac{(k_{19}[NO] + k_{20}[NO_2] + k_{21}[HONO])}{(k_5[O_3] + k_6[CO] + k_7[CH_4] + k_{12a}[HCHO] + k_{12b}[CH_3CHO] + k_{12c}[CH_3C(O)CH_3] + k_{12d}[CH_3OH] + k_{12e}[CHOCHO] + k_{17}[HO_2] + k_{19}[NO] + k_{20}[NO_2] + k_{21}[HONO])}$$

Measurements of $CH_4$, HCHO, $CH_3CHO$, CHOCHO, $CH_3OH$, and $CH_3C(O)CH_3$ on-board HALO are available and implemented in Eq. 6. These comprise the most abundant and reactive OVOCs and are considered to be a representative surrogate for the VOCs that act as $RO_2^*$ precursors through oxidation and photolysis. During the EMeRGe campaign in Europe, $k_{12a} \times HCHO$ and $k_{12b} \times CH_3CHO$ have the highest contribution to the $1 - \rho$ from all the OVOC measured. Their impact onthe $RO_2^*$ budget is found to be similar because their respective concentrations compensate the difference in the rate coefficients of their reactions with OH ($k_{12a} = 8.5 \times 10^{-12}$ $cm^3$ molecule$^{-1}$ s$^{-1}$ and $k_{12b} = 1.5 \times 10^{-11}$ $cm^3$ molecule$^{-1}$ s$^{-1}$ at 298K and 1 atm.). Despite its high mixing ratios measured, $CH_3C(O)CH_3$ is less important in the $1 - \rho$ term. This is because the rate coefficient $k(T)_{12c}$ is significantly slower than $k_{12a}$ and $k_{12b}$ (see Table S1 in the supplement). Similarly, the contribution of CHOCHO and $CH_3OH$ is an order of magnitude lower than that of HCHO and $CH_3CHO$.

Concerning the term $\delta[RO_2^*](k_{23}[NO] + k_{24}[O_3])\rho$ on the right hand side of Eq.6, the $HO_2$ reaction with $O_3$ has a negligible effect as $k_{24}$ is almost four orders of magnitude smaller than $k_{23}$ and the NO concentrations remained about three orders of magnitude smaller than the $O_3$ measured during the campaign."

- *L367 ff: It sounds as if you state that the reaction of OH+ HCHO and OH+ CH3CHO are the dominant radical precursor reactions, though so far you only discuss photolysis of them. OH reactions would also not be primary sources, but radical conversion reactions. In this context and for the same reason, it is also not clear, what you mean with RO2\*production from CHOCHO and CH3OH oxidation. Please clarify and rephrase.*

**Answer:**

This has been addressed with the answer above.

- *L373: The context of the statement about the importance of HO2+NO and HO2+O3 is not clear and seems displaced at this point.*

**Answer:**

The sentence has been reworded and extended for clarification:

Line 382: "Concerning the term $\delta[RO_2^*](k_{23}[NO] + k_{24}[O_3])\rho$ , the $HO_2$ reaction with $O_3$ has a negligible effect in Eq. 6 as $k_{HO_2+O_3}$ is almost four orders of magnitude smaller than $k_{HO_2+NO}$ and the NO concentrations remained about three orders of magnitude smaller than the $O_3$ measured during the campaign."

- *L421: How can you exclude that there is no over-estimation of loss processes instead of an under-estimation of production processes? What is the impact in the uncertainty of the HO2/RO2 ratio in the case, when VOCs concentrations were high?*

**Answer:**

The overestimation of loss processes cannot be excluded. The text starting in Line 452 has been reworded and extended for clarification:

"$RO_2^*{}_m$ is both underestimated and overestimated for $\sum$VOCs mixing ratios greater than 7 ppbv. The composition of these air masses is very different, as reflected by the $\Sigma$VOCs/NO ratios. This implies that Eq. 7 does not capture the peroxy radical yields adequately from the measured VOCs and OVOC in these cases. The differences between $RO_2^*{}_m$ and $RO_2^*{}_c$ may be explained in part by a) changes in OH yields due to additional VOC oxidation processes, which are not in Eq. 7 and/or b) $RO_2^*$ production from the photolysis of carbonyls, which were not measured and/or c) $RO_2^*$ production from the ozonolysis of alkenes or unidentified biogenic terpene emissions and/or d) overestimation of the loss processes. For VOC < 2ppb and $\sum$VOCs/NO < 20, $RO_2^*{}_m$ is systematically overestimated. This might indicate underestimation in the radical losses through nitrite and nitrate formation."

The uncertainties related to the presence of RO2 other than CH3O2 has been discussed in the answer to L344. In the text starting in Line 506, the effect of changes in the $HO_2$ to $RO_2$ ratio on the PeRCEAS accuracy during the E-EU-03 flight is discussed.

"Taking $CH_3O_2$ as a surrogate for all $RO_2$ might lead to uncertainties in the $RO_2^*$ calculations in the presence of OVOCs with larger organic chains. On the experimental side, changes in the $HO_2$ to $RO_2$ ratio affect the accuracy of the PeRCEAS retrieval of the total sum of radicals. As noted in section 3, in this study $RO_2^* = HO_2 + 0.65 \times RO_2$, and the eCL is determined for a 1:1 mixture of $HO_2:CH_3O_2$, i.e. $\delta = 0.5$ is used for the $RO_2^*$ retrieval. However, the $HO_2$ to $CH_3O_2$ ratio is not expected to remain constant in all the air masses probed. For a 3:1 ratio of $HO_2:RO_2$, the $RO_2^*{}_m$ would decrease by 10 %. Similarly, a $HO_2:RO_2$ ratio of 1:3 would lead to an increase of 10 % in the reported $RO_2^*{}_m$. This uncertainty is well below the in-flight uncertainty of the PeRCEAS instrument indicated by the error bars in Fig. 14 (George et al., 2020), and cannot account for the overall 20 % underestimations. However, it might reduce the differences observed between $RO_2^*{}_m$ and $RO_2^*{}_c$ in particular cases. A complete explanation of the variability of $RO_2^*$ in the pollution plumes measured within the IOP in Europe is beyond the scope of this analysis and requires an investigation by high-resolution chemical models"

- *L422: Why would OH recycling processes increase the calculated RO2*, if radical regeneration terms cancel out in the calculations for the sum measurement of RO2*?*

**Answer:**

The use of "recycling processes" was not adequate. This part of the text has been rephrased to prevent confusion.

- *L436: It would be interesting to see a more quantitative analysis of the impact of the uncertainty in HCHO measurements on the results.*

**Answer:**

The uncertainty of the HCHO measurements is ≤ 40 % in 95 % of the measurements considered in this study. The impact of this HCHO uncertainty on the $RO_2^*{}_c$ depends on the relative contribution of the HCHO photolysis to the total $RO_2^*$ production rates. For the measurements below 4000 m, where the HCHO photolysis contribution is < 40 %, the HCHO uncertainty contributes ≤ 15 % of the total uncertainty of $RO_2^*{}_c$. For the measurements above 6000 m with 80 % of the $RO_2^*{}_c$ formed from HCHO photolysis, the HCHO measurement uncertainty contributes up to 35 % of the total uncertainty of $RO_2^*{}_c$.

- *L490 ff: The calculation of OH concentrations does not really fit this manuscript and would require a much deeper description that currently done. The statement that the OH calculated from Eq 5 is higher than reported OH concentration means that OH reactivity is underestimated cannot easily be justified. I would recommend to cancel this entire paragraph. It does not add anything to the content of the manuscript and may even be rather misleading as it is now.*

**Answer:**

The whole paragraph has been deleted as proposed by the referee.

- *Section 4.4.1. / Equation 11: Again the definitions of the effective rate coefficients is not clear. Also the use of NOx makes it hard to see, what exactly is calculated. This makes it very difficult to follow any of the subsequent quantitative statements. The connection to previous Equations is also not clear. What is the difference to Equation 9, which should consider radical loss in NOx reactions? What is used for the production rate for example? The authors should make much clearer what is calculated and what the meaning of the calculation is. As it is written now, it is not clear, what the authors want to discuss in this section.*

**Answer:**

Eq. 11 (now Eq. 9 in the revised manuscript) refers to the analysis made by Cantrell et al. (2003b) and the terms of their analysis are just transcribed. The first term on the right hand refers to radical-radical reactions and the second term to $RO_2^*$-NOx reactions where $RO_2^*$ is considered to be the sum of $HO_2$+ $RO_2$ and $K_{RR}$ and $K_{RN}$ are effective rate coefficients, whose value is retrieved from the fitting of the curves obtained from real data. Except for the production rates there is no calculation made. As explained in the text, the production rates calculated by Eq.2 in the manuscript and the EMeRGe measurements in Europe are used to obtain the Figure 16. The results indicate that the simplified approach of Cantrell et al. (2003b) is not applicable to the more complex non-linear processes involved in the air masses investigated within EMeRGe.

The text has been partly reworded for clarification.

---

## Referee Report (RR1)

1. In my (anonymous ref #2) original comments I pointed out that there is a considerable amount of text in the introduction regarding the NO-$NO_2$-$O_3$ photostationary state equation (Eq. 1), and yet it does not appear later in the text. In their response document they note "This ratio calculated from the NO (in-situ) and $NO_2$ (miniDOAS remote) measurements during the EMeRGe campaign is considered to have a sufficiently large error for not to be a valuable approach to calculate the [$HO_2$+$RO_2$]", and in their revision have only added ""The PSS radical calculation made on the assumption of the NO2 steady state is very sensitive to the accuracy of the NO2 to NO ratio and the O3 measurements." Yet, the considerable amount of text and equation 1 remain in the revision. This is very confusing for the reader. If the authors insist on keeping it then they should be much more explicit that this approach is not used (e.g., at least they could add that same sentence from their response document into the manuscript). Furthermore it would be useful to the reader if they clarified that there are two HOx radical calculation methods discussed here: 1. The NO-NO2 interconversion photostationary state, described by equation 1, discussed from lines 96 – 120, and 2. The pseudo-steady state analysis presented in section 4.3 "**PSS estimation of the $RO_{2*}$ mixing ratios**" , including equation 4: P($RO2^*$) = D($RO2^*$)

2. Another comment of mine that was not adequately addressed:

The original manuscript contains ""The dominant removal processes of RO2* in the airmasses measured up to

2000 m are the loss of OH and RO through the reaction with NOx during the radical". In my original review I noted "reactions of RO with NOx are rare and not discussed at all later in the manuscript". The revision now includes this text: "The dominant terminating processes for RO2* in the pollution plumes measured up to 2000 m are the formation of nitrites and nitrates from radical reactions with NOx."

They have not understood my main point: formation of nitrites is not thought to be a major $RO_2$* removal process. I am unaware of supporting data from their study or from the literature that indicates that formation of **nitrites** can be a major $RO_2$* sink. In their *instrument* a large amount of $CH3O_2$ is lost by the reaction CH3O + NO + M → CH3ONO + M, i.e. formation of methyl nitrite, but these types of reactions are thought to be quite minor in the atmosphere. If the statement that formation of nitrites is a major $RO_2$* sink is to be kept in the manuscript it needs to be supported (which would be difficult). Moreover, it is not clear if they are including $HNO_3$ as a nitrate. I suspect the correct course of action is to simply change the sentence to "The dominant terminating processes for RO2* in the pollution plumes measured up to 2000 m are the formation of nitric acid and organic nitrates"

3. Section 4.4.1. In my initial review I objected to the formulation of eq 11 (now eq 9 in revised manuscript). While I understand that the authors are referring to the analysis by Cantrell et al. (2003b), it is still very confusing:

$$P_{RO2*} = k_{RR} [HO_2 + RO_2]_2 + k_{RN} [HO_2 + RO_2] [NO_x] \qquad \text{(Eq. 9)}$$

"The first term on the right hand refers to radical radical reactions and the second term to RO2*-NOx reactions where RO2* is considered to be the sum of HO2+ RO2 and $K_{RR}$ and $K_{RN}$ are effective rate coefficients"

The authors (and Cantrell) are calculating P($RO_2$*) by equating it to L($RO_2$*) (very safe assumption), and then calculating the L($RO_2$*) terms. For the last term, the relevant $RO_2$* loss reactions that need to be be summed in that term are the following:

OH + $NO_2$ → HNO3

$RO_2$ + $NO_2$ → RO2NO2

$HO_2$ + $NO_2$ → HO2NO$_2$

$RO_2$ + NO → RONO2

While the 2[nd] and third reactions can be combined with an effective rate constant, I simply do not see how all four terms can be combined into the single term $k_{RN}$ [$HO_2$ + $RO_2$] [$NO_x$]. The OH concentration does not necessarily scale with the [$HO_2$ + $RO_2$] concentration.

Wouldn't it be simpler to calculate P($RO_2$*) more directly based on $RO_2$* formation reactions rather than the loss processes? ie, from photolysis of HCHO, $O_3$/$H_2O$, etc.

Figure 16 is a nice figure btw: I am not trying to suggest this entire section is bad, but its formulation is still problematic.

Equation 2, last term is $\sum k_{O3+alkenes k}$[$O_3$][alkenes$_k$] – the ROx yield from this reaction should be included (i.e., 2 × the OH yield).

---

## Author Response (AR2)

**Additional Comment on acp-2022-119**

Anonymous Referee #1

The paper has been improved. The presentation quality is partly still not satisfying. I recommend that the authors go through the manuscript once more to reduce the number of abbreviations that are used, to format references of reactions as "Reaction RXX", to make sure that all font sizes in figures are appropriate and lab jargon such as "HALO flight", "HALO base", is avoided.

**Answer:**

As suggested by the referee, the references of reactions have been changed as "Reaction RXX". The figure sizes have been optimised for adequate font size, and the words such as "HALO flight" and "HALO base" are change to 'measurement flight' and 'aircraft hangar' respectively. The number of abbreviations is already at its minimum.

Anonymous Referee #2

1. In my original comments I pointed out that there is a considerable amount of text in the introduction regarding the NO-$NO_2$-$O_3$ photostationary state equation (Eq. 1), and yet it does not appear later in the text. In their response document they note "This ratio calculated from the NO (in-situ) and $NO_2$ (miniDOAS remote) measurements during the EMeRGe campaign is considered to have a sufficiently large error for not to be a valuable approach to calculate the [$HO_2$+$RO_2$]", and in their revision have only added ""The PSS radical calculation made on the assumption of the $NO_2$ steady state is very sensitive to the accuracy of the $NO_2$ to NO ratio and the $O_3$ measurements." Yet, the considerable amount of text and equation 1 remain in the revision. This is very confusing for the reader. If the authors insist on keeping it then they should be much more explicit that this approach is not used (e.g., at least they could add that same sentence from their response document into the manuscript). Furthermore it would be useful to the reader if they clarified that there are two $HO_x$ radical calculation methods discussed here: 1. The NO-$NO_2$ interconversion photostationary state, described by equation 1, discussed from lines 96 – 120, and 2. The pseudo-steady state analysis presented in section 4.3 "PSS estimation of the $RO_2$* mixing ratios" , including equation 4: P($RO_2$*) = D($RO_2$*)

**Answer:**

The authors prefer to keep Eq. 1 and the discussion about $NO_2$ steady sate approach from previous publications as this was the main approach used in filed measurements to calculate the sum of peroxy radicals. So, the text has been modified as suggested by the referee to help the reader understand there are two steady state approaches, i.e., considering $NO_2$ under study state and sum of $HO_2$ and $RO_2$ under steady state. For these lines 117 to 136 are modified as:

"The radical calculation made on the assumption of the $NO_2$ steady state is very sensitive to the accuracy of the $NO_2$ to NO ratio and the $O_3$ measurements. The comparison of $[HO_2 + RO_2]_{PSS}$ calculated using Eq.1 with ground-based (e.g., Ridley et al., 1992; Cantrell et al., 1997; Carpenter et al., 1998; Volz-Thomas et al., 2003), and airborne measurements, has shown in the past different degrees of agreement. The underestimations and overestimations found in air masses with different chemical compositions are not well understood. For the case of airborne measurements, the $NO_2$ steady state calculation generally overestimates the measured peroxy radicals (Cantrell et al., 2003a, 2003b). The differences observed could not be attributed to systematic changes in NO, altitude, water vapour and temperature, although these variables are often correlated. The NO to $NO_2$ ratio

calculated from NO measured using in-situ technique and $NO_2$ measured using remote sensing (more detail about the measurement techniques is given in Table 1) used in this study is considered to have a sufficiently large error. So, the $NO_2$ steady state approach is not accurate enough to calculate $[HO_2+RO_2]$ for the measurements considered in this study.

Ground-based (Mihelcic et al., 2003; Kanaya et al., 2007, 2012; Elshorbany et al., 2012; Lu et al., 2012, 2013; Tan et al., 2017, 2018; Whalley et al., 2018, 2021; Lew et al., 2020) and airborne (Crawford et al., 1999; Tan et al., 2001; Cantrell et al., 2003b) measurements have also been compared with model simulations of $HO_2$ and $RO_2$. The discrepancies encountered depend upon the chemical composition of the air mass and the chemical mechanisms and constraints used in the model simulations. Recently, Tan et al., 2019 and Whalley et al., 2021 reported experimental radical budget calculations using PSS assumption for OH, $HO_2$ and $RO_2$ together with the published reaction rate coefficients of the reactions (R1 to R26), which control OH, $HO_2$ and $RO_2$ in the lower troposphere, and the ground-based measurements of all relevant reactants and photolysis frequencies. In this study, a similar approach has been used, i.e., the sum of $HO_2$ and $RO_2$ is assumed to be in PSS, to calculate the amount of peroxy radicals in the air masses measured on-board of the **H**igh **A**ltitude **Lo**ng range (HALO) research aircraft over Europe during the first campaign of the EMeRGe (**E**ffect of **Me**gacities on the transport and transformation of pollutants on the Regional to Global scal**e**s) project."

2. Another comment of mine that was not adequately addressed:

The original manuscript contains ""The dominant removal processes of $RO_2$* in the airmasses measured up to 2000 m are the loss of OH and RO through the reaction with $NO_x$ during the radical". In my original review I noted "reactions of RO with $NO_x$ are rare and not discussed at all later in the manuscript". The revision now includes this text: "The dominant terminating processes for $RO_2$* in the pollution plumes measured up to 2000 m are the formation of nitrites and nitrates from radical reactions with NOx."

They have not understood my main point: formation of nitrites is not thought to be a major $RO_2^*$ removal process. I am unaware of supporting data from their study or from the literature that indicates that formation of nitrites can be a major $RO_2$* sink. In their instrument a large amount of $CH_3O_2$ is lost by the reaction $CH_3O + NO + M \rightarrow CH_3ONO + M$, i.e. formation of methyl nitrite, but these types of reactions are thought to be quite minor in the atmosphere. If the statement that formation of nitrites is a major $RO_2$* sink is to be kept in the manuscript it needs to be supported (which would be difficult). Moreover, it is not clear if they are including $HNO_3$ as a nitrate. I suspect the correct course of action is to simply change the sentence to "The dominant terminating processes for $RO_2$* in the pollution plumes measured up to 2000 m are the formation of nitric acid and organic nitrates"

**Answer:**

The sentence has been changed on line 32 as the referee suggested:

"The dominant terminating processes for $RO_2^*$ in the pollution plumes measured up to 2000 m are the formation of nitrous acid, nitric acid and organic nitrates."

3. Section 4.4.1. In my initial review I objected to the formulation of eq 11 (now eq 9 in revised manuscript). While I understand that the authors are referring to the analysis by Cantrell et al. (2003b), it is still very confusing:

$$P_{RO_2^*} = k_{RR} [HO_2 + RO_2]^2 + k_{RN} [HO_2 + RO_2] [NO_x] \text{ (Eq. 9)}$$

"The first term on the right hand refers to radical radical reactions and the second term to $RO_2^*$-$NO_x$ reactions where $RO_2^*$ is considered to be the sum of $HO_2$+ $RO_2$ and $K_{RR}$ and $K_{RN}$ are effective rate coefficients"

The authors (and Cantrell) are calculating $P(RO_2^*)$ by equating it to $L(RO_2^*)$ (very safe assumption), and then calculating the $L(RO_2^*)$ terms. For the last term, the relevant $RO_2^*$ loss reactions that need to be summed in that term are the following:

$OH + NO_2 \rightarrow HNO_3$

$RO_2 + NO_2 \rightarrow RO_2NO_2$

$HO_2 + NO_2 \rightarrow HO_2NO_2$

$RO_2 + NO \rightarrow RONO_2$

While the 2nd and third reactions can be combined with an effective rate constant, I simply do not see how all four terms can be combined into the single term $k_{RN} [HO_2 + RO_2] [NO_x]$. The OH concentration does not necessarily scale with the $[HO_2 + RO_2]$ concentration.

Wouldn't it be simpler to calculate $P(RO_2^*)$ more directly based on $RO_2^*$ formation reactions rather than the loss processes? ie, from photolysis of HCHO, $O_3/H_2O$, etc.

Figure 16 is a nice figure btw: I am not trying to suggest this entire section is bad, but its formulation is still problematic.

**Answer:**

In this study and in that of Cantrell et al. 2003b, the $P(RO_2^*)$ is calculated form the measured values radical precursors like $O_3$, HCHO etc. and their photolysis frequencies. The Eq. 9 uses this calculated $P(RO_2^*)$ and $RO_2^*$ (both calculated and measured) to fit Eq.9 and find $k_{RR}$ and $k_{RN}$. Cantrell eta al. 2003b showed this relatively simple expression can reproduce the $RO_2^*$ to $NO_x$ relation fairly well for both measured and model $RO_2^*$ from TRACE-P campaign. So, a similar approach was also made in this study were, the $RO_2^*_m$ and $RO_2^*_c$ binned by $P(RO_2^*)$ values calculated using Eq. 3 and the corresponding $P(RO_2^*)$ values are substituted in Eq. 9 to find the $k_{RR}$ and $k_{RN}$ as fit parameters. The weak agreement between $RO_2^*m$ and the fit profile (Figure 16 b) shows the $k_{RR}$ and $k_{RN}$ terms are not adequate to express the $RO_2^*$ reaction rate as suggested by the referee. This is already written in line 554.

In addition, figure 16 a) shows that neglecting the $RO_2 + NO \rightarrow RONO_2$ in Eq. 7 results in very weak $NO_x$ dependency for $RO_2^*_c$ with $P_{RO_2^*} \geq 0.7$ pptv s$^{-1}$ as pointed by referee #4.

Text has been modified in section 4.4.1 for better clarity and to address the point raised by referee #4.

4. Equation 2, last term is $\Sigma kO_3 + alkenes_k[O_3][alkenes_k]$ – the ROx yield from this reaction should be included (i.e., 2 × the OH yield).

**Answer:**

The equation 2 has been corrected as the referee pointed out with an effective $RO_2^*$ yield, denoted by γ, from ozonolysis of alkenes.

Anonymous Referee #4

In this revision, the authors have addressed many of my concerns outlined in my original review. Specifically, they have corrected equation 8 (now equation 6) by excluding radical termination by the RO + NO reaction, which may be important in their chemical amplifier, but not in the atmosphere.

However, they still have not adequately addressed potential peroxy radical loss by the $RO_2 + NO_x$ reactions. While they are assuming that all $RO_2$ radicals are $CH_3O_2$ and as a result the termination by this reaction is negligible, they should include the $RO_2 + NO \rightarrow RONO_2$ reaction in their generic reaction mechanism in the Introduction. They should also include it in their steady-state equation for completeness and can then make it clear that by assuming all $RO_2$ is $CH_3O_2$ that this term is negligible. They should also clarify that ignoring organic nitrate formation from these reactions may result in an overestimation of the peroxy radical concentrations in their calculations and could explain the some of the model overestimation highlighted in Figures 10 and 12.

**Answer:**

The reaction $RO_2 + NO \rightarrow RONO_2$ has been included in the generic reaction mechanism in the Introduction as R25b. Additional text has been included in line 364 as:

"The reaction channel R25b is not considered in the calculation since the yield of this channel is < 5 % (Burkholder et al., 2020) for $CH_3O_2 + NO$ reaction."

for the explanation of Eq. 6.

An extra paragraph has been added in line 464 to address the overestimation in Figures 10 and 12 as pointed out by the referee.

"In addition, Eq. 7 does not consider the loss of $RO_2$ through the organic nitrate formation (reaction R25b) which results in an underestimation of radical loss in the presence of $RO_2$ with higher organic group. Tan et al., 2019 reported changing the yields for organic nitrate formation channel in reaction R25 from 5% to 20% has a small but notable influence on their experimental budget analysis. Similarly, the $RO_2$ loss through organic nitrate formation which are not included in Eq. 7 might explain the $RO_2^*{}_m$ overestimations for $\sum VOC < 2ppb$ and $\sum VOCs/NO < 20$, and for $NO > 200$ pptv."

Summary and conclusion section is also extended on line 603 as:

"Similarly, the $RO_2$ loss through organic nitrate are also excluded from the analytical expression. These reactions may become significant $RO_2^*$ loss processes in the presence of $RO_2$ with higher organic groups. This might explain some of the $RO_2^*$ overestimations by the analytical expression observed for $NO > 200$ pptv."

The authors should also comment on whether ignoring radical termination by $RO_2+NO_x$ reactions their calculated $RO_2$ leads to the apparent discrepancy with that predicted by equation 10 for PROx > 0.7 ppt s$^{-1}$ as shown in Figure 16a and discussed on lines 536-538. As the authors note, the measured $RO_2$ does show a decrease with increasing $NO_x$ (Figure 16b, line 539), suggesting greater $RO_2$ radical termination by $NO_x$ than accounted for by their model.

**Answer:**

To address this point, the text in line 545 has been modified as:

"$RO_2{}^*{}_c$ does not show the decrease with increase in $NO_x$ for $P_{RO_2^*} \geq 0.7$ pptv s$^{-1}$. This might be explained by the under estimation of radical losses through organic nitrate formation in Eq. 7 as explained in section 4.3."

The revised manuscript also adds some confusing language related to their steady-state calculations. In particular, the new description of the terms in Equation 6 on page 17 (lines 365-369) is confusing. The authors state "ρ accounts for the effective yield of HONO, HNO3 and H2O formation through reactions R19 to R21 and the $HO_2$ + NO and $HO_2$ + $O_3$ reactions (R23 and R24 respectively) on the right hand side of Eq. 6." Rather than stating that they are account for the effective yield of $H_2O$ (which is a product of many OH + VOC reactions in addition to R21), the authors should clarify that they are attempting to account for the fraction of OH radical termination through the OH+NO, OH+$NO_2$, and OH+HONO reactions relative to OH radical loss by the OH+VOC, OH+$HO_2$, and other OH loss reactions. In this term, it also appears that they are assuming that OH production by reactions 23 and 24 is much greater than OH initiation. More details on how they have derived this equation should be provided for clarification.

**Answer:**

The explanation of terms in Eq. 6 has been modified in line 374 onwards as:

"On the left-hand side of Eq. 6, 1-ρ accounts for the effective yield of $HO_2+RO_2$ through the radical initiation reactions R2a and R3 and reactions R5 to R7 and reaction R12. As the calculation is constrained with on-board measurements, only the reactions of measured VOCs were considered in reaction R12. Similarly, on the right-hand side of Eq. 6, ρ accounts for the radical termination through the OH + NO, OH + $NO_2$, and OH + HONO reactions (reactions R19 to R21) relative to the radical undergoing OH to peroxy radical conversion."

As now explained, the ρ account for radical termination relative to radical undergoing OH to peroxy radical conversion. The detailed derivation of Eq. 6 and Eq. 7 are already given in the supplementary information.

---

## Author Response (AR3)

**Author's Response**

**Technical corrections:**

- The equations for rho and beta after Equation 6 should be also numbered; the numbering of the subsequent equations should be adjusted accordingly.

The equation numbering is modified as suggested by the handling editor.

- Figure 9: Spell out 'num. of meas.' . There is enough space to have the legend in two lines if needed

num. of meas. in all the figures (fig. 3, 5, 9 and 11) has been spell out as suggested by the handling editor.

- Table 2: Make sure that the Table is separated by a line break from the rest of the text. The first column seems redundant since this information is included in the Table caption. As an alternative you may want to consider adding the resulting linear equations to the panels in Figure 10 and skip the table, e.g.

RO2,m = 5 + 0.71 RO2,c (R^2 = 0.96)

The table 2 is now removed and the fit parameters are now given in Figure 10 and 12a as suggested by the handling editor.

- l. 465 & l. 605: 'with higher organic group' is not clear. Please specify what you mean by 'higher' (larger than CH3?)

The text has been modified as (now line 466 and 607):

"$RO_2$ with organic group larger than $CH_3$."

l. 467: ...which is not included ... (replace 'are' by 'is')

Text modified as suggested by the handling editor (now line 468).

l. 604: '...is also included' (replace 'are' by 'is')

Text modified as suggested by the handling editor (now line 606).